# Function and phylogeny support the independent evolution of an ASIC-like Deg/ENaC channel in the Placozoa

Wassim Elkhatib [1,2], Luis A. Yanez-Guerra [3], Tatiana D. Mayorova[4], Mark A. Currie[1,2], Anhadvir Singh[1,2], Maria Perera[1,2], Julia Gauberg [1,2] & Adriano Senatore [1,2✉]

ASIC channels are bilaterian proton-gated sodium channels belonging to the large and functionally-diverse Deg/ENaC family that also includes peptide- and mechanically-gated channels. Here, we report that the non-bilaterian invertebrate *Trichoplax adhaerens* possesses a proton-activated Deg/ENaC channel, *Tad*NaC2, with a unique combination of biophysical features including tachyphylaxis like ASIC1a, reduced proton sensitivity like ASIC2a, biphasic macroscopic currents like ASIC3, as well as low sensitivity to the Deg/ENaC channel blocker amiloride and $Ca^{2+}$ ions. Structural modeling and mutation analyses reveal that *Tad*NaC2 proton gating is different from ASIC channels, lacking key molecular determinants, and involving unique residues within the palm and finger regions. Phylogenetic analysis reveals that a monophyletic clade of *T. adhaerens* Deg/ENaC channels, which includes *Tad*NaC2, is phylogenetically distinct from ASIC channels, instead forming a clade with BASIC channels. Altogether, this work suggests that ASIC-like channels evolved independently in *T. adhaerens* and its phylum Placozoa. Our phylogenetic analysis also identifies several clades of uncharacterized metazoan Deg/ENaC channels, and provides phylogenetic evidence for the existence of Deg/ENaC channels outside of Metazoa, present in the gene data of select unicellular heterokont and filasterea-related species.

[1] Department of Biology, University of Toronto Mississauga, 3359 Mississauga Road, Mississauga, ON L5L 1C6, Canada. [2] Department of Cell and Systems Biology, University of Toronto, 25 Harbord Street, Toronto, ON M5S 3G5, Canada. [3] Living Systems Institute, University of Exeter, Stocker Road, Exeter EX4 4QD, England. [4] NINDS, National Institutes of Health, Bethesda, MD 20892, USA. ✉email: adriano.senatore@utoronto.ca

Degenerin/Epithelial Na$^+$ Channels (Deg/ENaC channels) are a large family of metazoan cation channels that exhibit a remarkable diversity in their mechanisms for activation, gating, and physiological functions. Mammals for example possess three major types of Deg/ENaC channels: Acid Sensing Ion Channels (i.e., ASIC channels), which are activated by extracellular protons and serve as major extracellular pH sensors in the central and peripheral nervous system[1], Bile-Acid Sensitive Ion Channels (BASIC channels), which are expressed in the brain, liver, and intestinal epithelium and are activated by bile acids[2], and Epithelial Na$^+$ Channels (ENaC channels), which conduct Na$^+$ leak currents in epithelial cells of the lung and kidney important for Na$^+$ reabsorption and homeostasis[3]. Despite these distinctions, ASIC and ENaC channels also share some overlapping functional properties, in that both are modulated by mechanical stimuli[4,5], and the ENaC-δ subunit, which is expressed in the brain, is proton-activated similar to ASIC channels albeit with different gating kinetics[6]. All Deg/ENaC channels are thought to form hetero- and/or homotrimeric holochannels, with each subunit comprised of two membrane spanning helices and a large extracellular domain[7]. Nonetheless, during evolution these channels underwent extensive sequence divergence and genetic expansion/loss in different animal lineages, in several cases obscuring their phylogenetic relationships. Although several recent phylogenetic studies have provided some important insights[8–10], there are still unanswered questions about the evolutionary origins ASIC, BASIC, and ENaC channels, and their relationships with the many divergent channels identified in invertebrates.

Like in mammals, invertebrate Deg/ENaC channels show striking diversity in their mechanisms for activation and physiological functions. In the nematode worm *C.elegans* for example, the subunits MEC-4 and MEC-10 form mechanically-gated heterotrimeric channels that are required for sensory mechanotransduction[11], while multiple homologs including ACD-1, ACD-5, and the heteromeric channel FLR-1/ACD-3/DEL-5 are all proton-inhibited channels that, respectively, are involved in acidic pH avoidance behavior, oscillations in intestinal lumen pH, and regulation of intracellular Ca$^{2+}$ waves[12–14]. On the other hand, the *C. elegans* channels ACD-2, DEL-9, and ASIC-1 are activated by external protons and are thought to play roles in neurotransmission[14], making them similar to ASIC channels in both their mode of activation and physiological function. Insects like the vinegar fly *Drosophila melanogaster* also possess a diverse set of Deg/ENaC channels[15], including the proton-activated channel Pickpocket 1 (PPK1) which is expressed in proprioceptive and nociceptive sensory neurons where it is also thought to respond to mechanical stimuli[16,17]. In molluscs and annelids, Deg/ENaC channels act as neurotransmitter receptors activated by the secreted neuropeptides FMRFamide and Wamide[9,18–20]. Peptide-gated channels are also found in the cnidarian species *Hydra magnipapilatta* (a hydrozoan), in the form of *Hy*NaC channels that are activated by RFamide neuropeptides[21,22]. In the cnidarian *Nematostella vectensis*, an anthozoan, Deg/ENaC channels are unresponsive to neuropeptides, but rather, are proton sensitive with the channel *Ne*NaC8 being blocked by protons, and the channels *Ne*NaC2 and *Ne*NaC14 being proton-activated[8]. In this species, the ASIC-like channel *Ne*NaC2 contributes to cnidocyte discharge and expulsion of a venom-laced barb for defense and prey capture[8,23]. To date, the most early-diverging Deg/ENaC channel to be functionally characterized in vitro is *Tad*NaC6 from the placozoan species *Trichoplax adhaerens*, which forms a Na$^+$ leak channel that is blocked by external protons and Ca$^{2+}$ ions[24]. Placozoans are an intriguing group of animals that lack nervous systems, and yet possess a large complement of genes involved in neural and synaptic signaling. Combined, the unclear phylogenetic relationships of Deg/ENaC channels, coupled with their diverse but sometimes similar modalities for gating (e.g., mechanical force, peptides, and protons), makes it challenging to infer whether these functional features are of common descent, or evolved independently.

Here, we report the functional properties of a second Deg/ENaC channel from *T. adhaerens*, *Tad*NaC2, which forms a proton-activated channel in vitro with biophysical properties that encompass the unique features of the three mammalian ASIC channels, ASIC 1 to 3. Specifically, *Tad*NaC2 macroscopic currents exhibit reduced sensitivity to proton activation like ASIC2a, are biphasic with distinct early and late components like ASIC3, and exhibit rundown or tachyphylaxis like ASIC1a. Through a combined cluster and phylogenetic analysis, we generated a phylogenetic tree that corroborates a recent report of two ancient and distinct subfamilies of metazoan Deg/ENaC channels, clades A and B, with clade A bearing the mammalian ASIC and BASIC channels and clade B the ENaC channels[8]. We also identified several uncharacterized subclades of Deg/ENaC channels as well as bona fide Deg/ENaC channel homologs from *Tunicaraptor unikontum* (a filasterean-related species) and heterokont single-celled eukaryotes. Lastly, we find that *Tad*NaC2 and most other *T. adhaerens* Deg/ENaC channels form a strongly-supported clade with bilaterian BASIC channels, within clade A but separate from ASIC channels, while the singleton *Tad*NaC10 channel falls within clade B.

To better understand the functional similarities and differences of *Tad*NaC2 relative to ASIC channels, we conducted structural and functional analyses, focusing on key extracellular regions of ASIC channels that bear molecular determinants for proton activation. We find that *Tad*NaC2 lacks all major determinants for proton activation of ASIC channels, including the critical H73 and K211 residues that are common to ASIC channels[10,25], and the acid pocket, a region also important for proton activation of ASIC channels[26]. Instead, our mutation analysis revealed that two histidine residues, H80 and H109, within the palm and finger regions, respectively, contribute to *Tad*NaC2 proton gating. Altogether, our phylogenetic and functional analyses suggest that proton-activated (ASIC-like) Deg/ENaC channels evolved independently in the early-diverging phylum Placozoa.

## Results

**Phylogenetic properties of Deg/ENaC channels from metazoans and non-metazoans.** To better understand the relationships of *T. adhaerens TadNaC* channels to other Deg/ENaC channels, including those from the fellow placozoan *Hoiliungia hongkongensis*, we used CLuster ANalysis of Sequences (CLANS)[27] on a set of 1074 Deg/ENaC channel protein sequences extracted from high-quality gene datasets of representative species spanning the major animal groupings, followed by phylogenetic inference. We tested a range of P value cut-offs for the CLANS analysis (i.e., 1E−10, 1E−20, 1E−30, 1E−40, and 1E−50), finding in all cases that the sequences formed one major cluster comprised of two inter-connected sub-clusters (Fig. 1, Supplementary Data 3 to 7). One of these sub-clusters contained the chordate ASIC and BASIC channels, along with the *T. adhaerens* channels *Tad*NaC1 to 9 and *Tad*NaC11 (and corresponding *H. hongkongensis* homologs), and the other the chordate ENaC channels with the singleton placozoan homologs *Tad*NaC10 and *Hho*NaC10. Our analysis is altogether consistent with a previous study[19], both also finding the peptide-gated FaNaC and WaNaC channels from lophotrochozoans to associate with the ENaC sub-cluster, and the peptide-gated *Hy*NaC channels from *Hydra magnipapilata* to associate with the ASIC/BASIC sub-cluster.

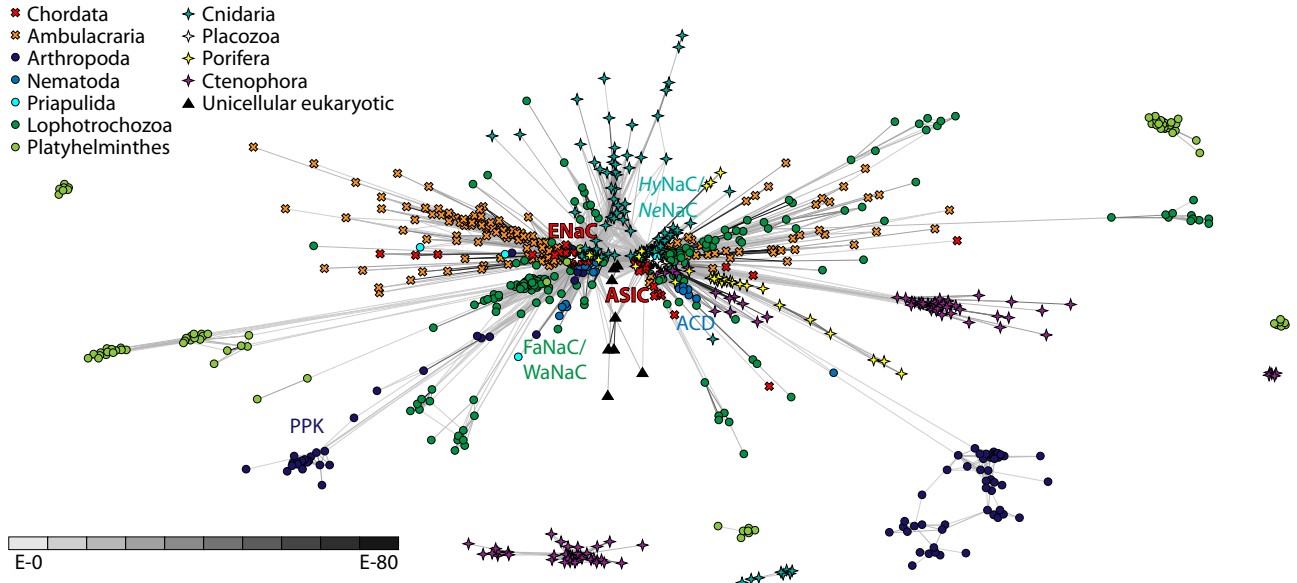

**Fig. 1 BLOSUM62 cluster map revealing two major sub-clusters of metazoan Deg/ENaC channels.** Nodes correspond to individual channel sequences and are colored by taxon as indicated by the legend. Edges correspond to BLAST connections with *P*-values < 1E−30. The general locations of the chordate ASIC and ENaC channels, the cnidarian *Hy*NaC and *Ne*NaC channels, the lophotrochozoan FaNaC and WaNaC channels, and the *C. elegans* ACD channels are indicated. Singletons and non-connected clusters with less than five sequences are masked but available in the corresponding CLANS file (Supplementary Data 5).

Comparing the various CLANS analyses at different thresholds, we found that decreasing the *P* value from 1E−20 to 1E−30 caused numerous sequences to no longer associate with the main cluster, including a large group of ctenophore sequences (Fig. 1, Supplementary Data 4 and 5). Decreasing it further to 1E−40 caused a large group of *D. melanogaster* PPK channels to no longer associate with the ENaC sub-cluster, and a set of *C. elegans* ACD channels to lose their relatively strong connectivity with the ASIC sub-cluster (Fig. 1), instead forming a single connection with the ENaC sub-cluster (Supplementary Data 6). We therefore selected a P value cut-off of 1E−30 to isolate a central cluster of sequences for phylogenetic inference, reasoning that this cut-off struck a balance between strategically removing divergent and/or truncated sequences that would interfere with phylogenetic analysis, while being permissive enough to include most PPK channels. In agreement, a previous study that employed a similar CLANS pre-filtering approach prior to phylogenetic analysis but with a P value of 1E−50 excluded the *D. melanogaster* PPK channels[19]. In our analysis, pre-filtering the sequences at 1E−30 resulted in the removal of 200 sequences, which in addition to the noted cluster of ctenophore channels, included numerous singletons and smaller clusters from platyhelminths and cnidarians (Fig. 1). Lastly, our clustering analysis revealed that several Deg/ENaC homologs present in the gene data for unicellular eukaryotic species from the clades Heterokonta (i.e., from the SAR supergroup, for Stramenopila, Alveolata, and Rhizaria) and Filasterea, clustered the ASIC and ENaC sub-clusters (Fig. 1), corroborating a report that Deg/ENaC channels are present outside of animals, in select unicellular organisms[28].

A maximum likelihood phylogeny inferred from the aligned protein sequences, rooted on the Deg/ENaC channel homologs from the unicellular filasterea-related species *Tunicaraptor unikontum*, reveals strong phylogenetic support for two distinct clades, termed Clades A and B, corresponding to the ASIC and ENaC sub-clusters (Fig. 2), which is consistent with another recent phylogenetic analysis[8]. In both analyses, most *Tad*NaC channels fall within Clade A (*Tad*NaC1 to 9 and 11), forming a

sister relationship with chordate BASIC channels. Instead, the singleton channel *Tad*NaC10, along with its orthologue from fellow placozoan *Hoilungia hongkongensis*, falls within Clade B. Our analysis also identifies several groups of uncharacterized channels that are positioned between the *Tad*NaC and BASIC channels in Clade A, with representatives from chordates (i.e., cephalochordate and urochordate), ambulacrarians (i.e., echinoderm and hemichordate), and lophotrochozoans (i.e., annelid and brachiopod). Our tree also expands the group of *C. elegans* channels that form a sister relationship with BASIC channels by including the channels ACD-1, ACD-5, and FLR-1, which notably, resemble *Tad*NaC6 and BASIC channels in being inhibited/blocked by external protons[13,14,24,29], and ACD-2 which is proton-activated[14]. Between *T. adhaerens* and *H. hongkongensis*, most Clade A Deg/ENaC channel sequences exhibit one-to-one orthology, except for *Tad*NaC4, 6, and 7, for which *H. hongkongensis* only bears the single homolog, *Hho*NaC4/6/7. Also consistent with previous reports[10,25], ASIC channels within our phylogenetic tree form two distinct subgroups, Groups A and B (not to be confused with Clades A and B), with chordates (vertebrates, urochordates, and cephalochordates) possessing only Group A orthologues, cephalochordates also possessing Group B orthologues, and ambulacrarians and lophotrochozoans only possessing Group B orthologues. Together, these various described channels form a well-supported subclade within Clade A (i.e., subclade I), which is distinct from subclade II which bears representatives from a broad range of bilaterian and non-bilaterian animals. This includes a clade of *C. elegans* channels bearing ACD-1, ACD-5, and FLR-1, which resemble *Tad*NaC6 and BASIC channels in being inhibited/blocked by external protons[13,14,24,29], and ACD-2 which is proton-activated[14], and a large clade of arthropod channels including the *D. melanogaster* PPK channels, of which PPK1 is also proton-activated[16]. In addition, Clade A subclade II includes two groups of cnidarian channels, one bearing the neuropeptide-gated *Hy*NaC channels from *Hydra magnipapillata*[21] and the proton-activated channel *Ne*NaC2 from *Nematostella vectensis*[8],

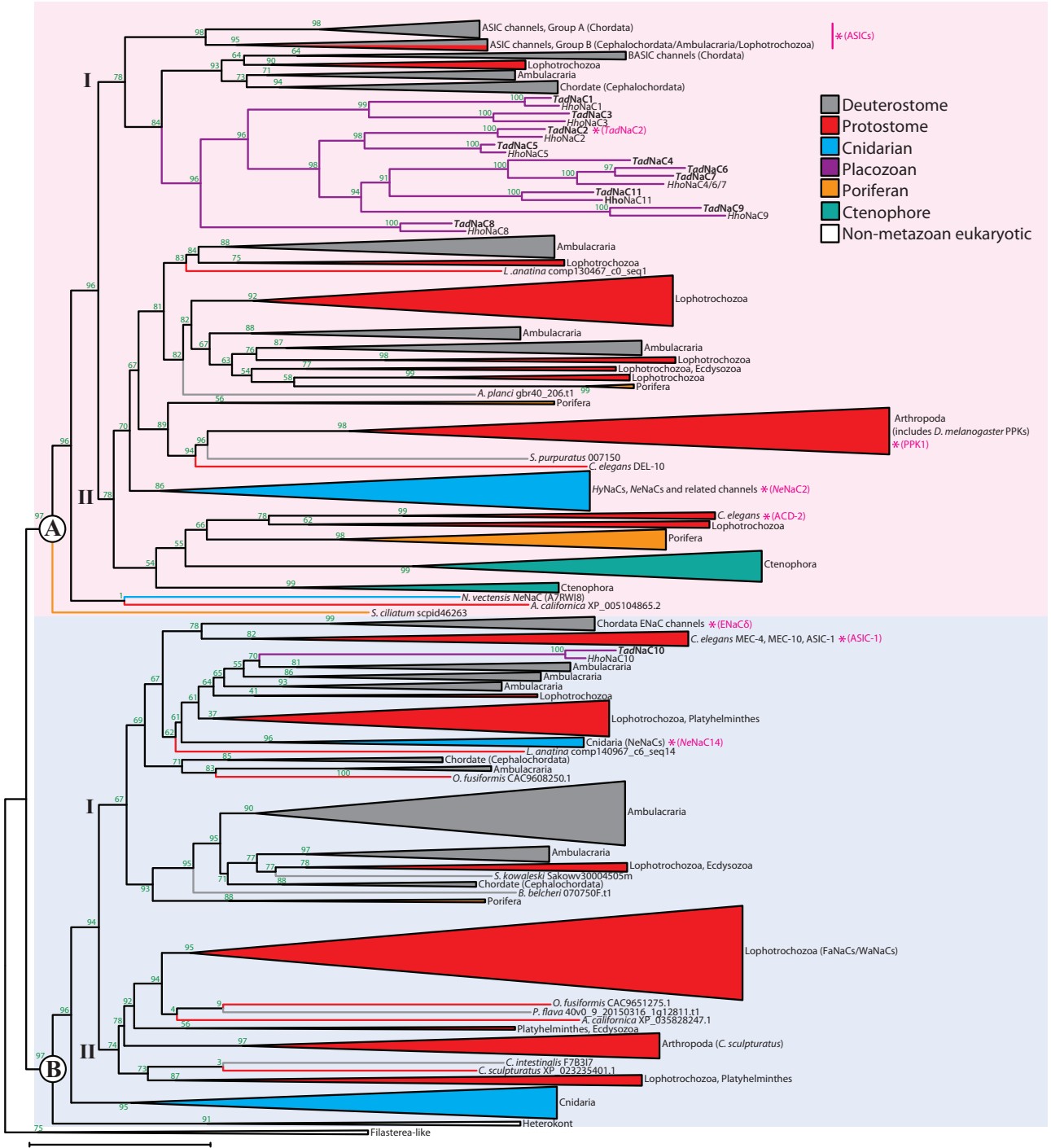

**Fig. 2 Maximum likelihood protein phylogeny delineates two major clades of metazoan Deg/ENaC ion channels.** The tree was generated with the program IQ-TREE 2 with the best-fit model WAG + F + G4 and rooted with the filarerean-related Deg/ENaC channel homologs. Node support values are for 100 standard bootstrap replicates (green). The asterisks and labels (pink) indicate single channels or clades bearing Deg/ENaC channels that have been characterized as proton-activated.

and several distinct groups of uncharacterized channels from protostomes, ambulacrarians, ctenophores, and poriferans.

Clade B similarly subdivides into two major subclades, with subclade I bearing the chordate ENaC channels and the *C. elegans* mechanosensory channels MEC-4 and MEC-10 and the proton-activated channel ASIC-1[8,14]. Also within subclade I are the placozoan channels *Tad*NaC10 and *Hho*NaC10, along with a diversity of uncharacterized channels from cephalochordates,

ambulacrarians, and protostomes, and a clade of cnidarian channels which includes the *N. vectensis* proton-activated channel *Ne*NaC14[8]. Clade B subclade II contains a large group of protostome channels which includes the peptide-gated FaNaC and WaNaC channels from annelids and molluscs[9], and several uncharacterized representatives from ambulacrarians, cephalochordates, and protostomes (i.e., lophotrochozoans and ecdysozoans including a large clade of channels from *Centroides*

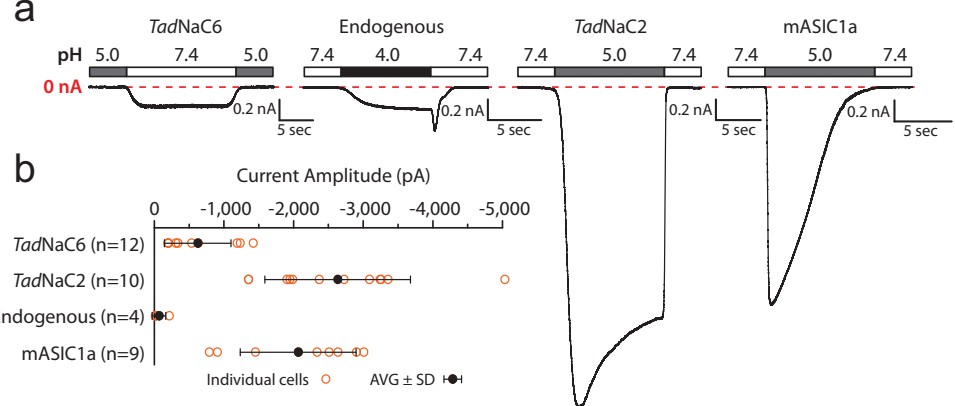

**Fig. 3 TadNaC2 conducts robust proton-activated currents in vitro. a** Sample whole-cell currents recorded for the previously characterized *Trichoplax* Deg/ENaC sodium leak channel *Tad*NaC6 that is blocked by extracellular protons[24], a newly identified endogenous current in CHO-K1 cells that becomes activated upon perfusion of strongly acidic solutions below pH 4.0, and large, prominent proton-activated currents conducted by the in vitro expressed *Trichoplax TadNaC2* and the mouse ASIC1a (mASIC1a) channels. **b** Plot of average peak inward current amplitude (in picoamps or pA) for currents shown in (**a**) ±standard deviation. Orange symbols denote values for individual cells/recordings.

*sculpturatus*). Lastly, a set of cnidarian channels and Deg/ENaC homologs from the unicellular heterokont *Cafeteria roenbergensis* form a sister clade relationship with all other Clade B channels.

Altogether, our combined CLANS and phylogenetic analysis provide strong evidence that most *Tad*NaC channels, including the previously described *Tad*NaC6 and the currently described *Tad*NaC2, are phylogenetically closer to BASIC channels than ASIC channels. Furthermore, our analysis corroborates the existence of two major groups of metazoan channels[8,19], identifies numerous groups of uncharacterized channels with phylogenetic proximity to channels with known properties, and provides phylogenetic evidence for the existence of Deg/ENaC channels outside of Metazoa.

**TadNaC2 conducts proton-activated currents in vitro that decay upon repeated activation like mouse ASIC1a.** Previously, we found that the *T. adhaerens* Deg/ENaC channel *Tad*NaC6 conducts constitutive Na$^+$ leak currents in vitro that are blocked by external protons and Ca$^{2+}$ ions[24] (Fig. 3a). Here, we set out to characterize the in vitro properties of a second *T. adhaerens* Deg/ENaC channel, *Tad*NaC2. Whole-cell patch clamp recording of Chinese Hamster Ovary (CHO)-K1 cells transfected with the *Tad*NaC2 cDNA revealed robust inward macroscopic cation currents elicited by perfusing a pH 5 solution over the recorded cells. No such currents were evident in untransfected cells, but we did observe a small endogenous inward current in these cells that became activated by solutions with a pH of 4 or lower (Fig. 3a). For comparison, we also transfected mouse ASIC1a (mASIC1a) which has been extensively studied in vitro, observing robust inward currents at pH 5 with a noticeably faster desensitization than *Tad*NaC2. *Tad*NaC2 whole-cell currents were quite large in amplitude, reaching upwards of 5000 picoamperes (Fig. 3b), despite the cDNA not being codon optimized as was required for efficient expression of the cnidarian *Hy*NaC channels in mammalian cells[30].

Next, we sought to compare the general properties of *Tad*NaC2 and mASIC1a proton-activated currents. Perfusion of external solutions of various pH revealed that *Tad*NaC2 begins activating at pH 5.5, with current kinetics that accelerate from a slow onset non-desensitizing current at pH 5.5, to a faster transient and partially desensitizing current at pH 4.0 (Fig. 4a). These currents are markedly different from those of mASIC1a, which began activating at the more basic pH of 6.7, with much faster activation

and desensitization evident across all tested values of pH. Notably, the *Tad*NaC2 currents appear biphasic, particularly upon activation with a pH 4.5 solution, with a fast/early transient component followed by a slower/late sustained component. Dose–response curves generated from these experiments revealed that *Tad*NaC2 is considerably less sensitive to external protons than mASIC1a (Fig. 4b), with a pH$_{50}$ of 5.1 ± 0.1 vs. 6.7 ± 0.1, and a Hill coefficient ($n_H$) value of only 1.7 ± 0.4 vs. 8.4 ± 2.7. Notably, these values for the mASIC1a channel are closely in line with those reported for the human ASIC1a channel recorded in *Xenopus* oocytes[25,26]. Together, the lower pH$_{50}$ and $n_H$ values observed for *Tad*NaC2 indicate a lower binding affinity and reduced cooperativity for extracellular proton binding, more inline with the sensitivity reported for the rat ASIC2a channel[31–33].

In early experiments, we found that *Tad*NaC2 currents exhibit a non-recovering decay in amplitude upon repeated activation. For example, applying paired 30 s pulses of pH 4.5 solution separated by neutral pH wash steps of either 40 or 80 s resulted in similar decreases in amplitude of 55.5 ± 16.5% with a 40 s interval vs. 50.74 ± 9.2% with an 80 s interval. Since doubling the interpulse interval from 40 to 80 s did not diminish the current decay amplitude, the observed process is not likely due to incomplete recovery from fast/acute desensitization. This feature of *Tad*NaC2 thus resembles the rodent ASIC1a channel which undergoes slow desensitization or tachyphylaxis, a unique process not observed for ASIC2 and ASIC3 proposed to involve a prolonged inactivated state that is distinct from acute desensitization[34,35]. To better characterize this property of *Tad*NaC2, we employed an experimental paradigm similar to one used previously to study tachyphylaxis of rat ASIC1a in *Xenopus* oocytes[34]. Specifically, we applied six 15-second pulses of pH 4.5 or 5.5 solutions over recorded cells expressing *Tad*NaC2 or mouse ASIC1a, separated by 55-second interpulse intervals. Consistent with observations in oocytes, mouse ASIC1a peak currents decayed upon repeated activation at pH 5.5 (Fig. 4c), decreasing to 49.4 ± 9.5% of their original value after 6 pulses (Fig. 4d). Similarly, peak *Tad*NaC2 currents declined to 57.9 ± 20.5% at pH 4.5, and 43.5 ± 16.4% at pH 5.5, while the late/sustained component of the *Tad*NaC2 current at pH 4.5 declined to 59.5 ± 22.9%. Analysis of the average data revealed that although the decline in current amplitude for each condition relative to the first pulse was statistically significant, the degree

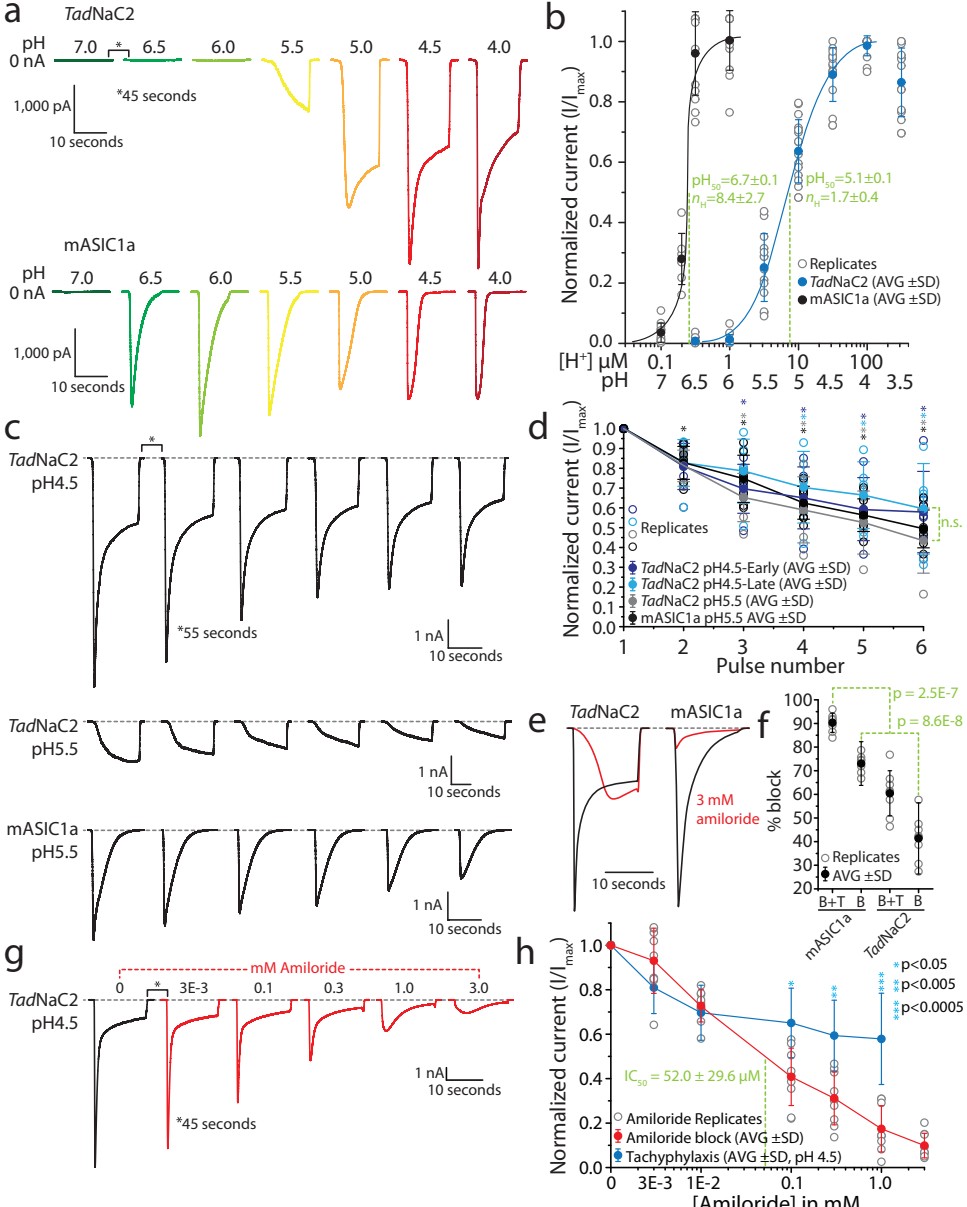

**Fig. 4 General electrophysiological properties of *Tad*NaC2 compared to mouse ASIC1a. a** Sample recordings of *Tad*NaC2 currents (top) and mouse ASIC1a currents (mASIC1a, bottom) elicited by perfusion of solutions with decreasing pH. **b** pH dose–response curves for *Tad*NaC2 ($n = 14$–17) and mASIC1a ($n = 7$–13) revealing a right shifted $pH_{50}$ for the *Trichoplax* channel relative to mASIC1a, and a smaller Hill coefficient ($n_H$). The values observed for mASIC1a are consistent with previous reports[26]. **c** Sample sequential *Tad*NaC2 currents exhibiting rundown or tachyphylaxis similar to mASIC1a. **d** Plot of average normalized current amplitude ± standard deviation through successive sweeps for *Tad*NaC2 (i.e., early and late currents at pH 4.5, $n = 6$, and peak current at pH 5.5, $n = 5$–6) and mASIC1a (peak current at pH 5.5, $n = 8$), revealing decaying amplitudes for all conditions that are statistically indistinguishable from each other (i.e., $p > 0.05$ for one-way ANOVAs comparing raw normalized values for each condition at each pulse). The asterisks indicate statistically significant $p$ values (i.e., $< 0.05$) for pairwise post hoc Tukey tests after one-way ANOVAs of each set of pulses for each condition (*Tad*NaC2 pH 4.5 early: $p = 1.0E-4$, F = 7.6; *Tad*NaC2 pH 4.5 late: $p = 2.4E-3$, F = 4.8; *Tad*NaC2 pH 5.5: $p = 4.3E-7$, F = 14.7; mASIC1 pH 5.5: $p = 1.4E-14$, F = 38.2). **e** Sample current recordings for *Tad*NaC2 and mASIC1a before (black traces) and after (red traces) perfusion of 3 mM amiloride, revealing a nearly complete block for mASIC1a (at pH 5.5) and only ~50% block for *Tad*NaC2 (pH 4.5). **f** Plot of average percent block of inward current ± standard deviation for *Tad*NaC2 ($n = 8$) and mASIC1a ($n = 7$) before and after perfusion of 3 mM amiloride. Individual replicates are included as *gray* circles. B + T indicates the total decay in average current for a successive sweep, which includes the effects of drug block (B) and tachyphylaxis (T), while B indicates the isolated component of drug block alone, obtained by subtracting the average decline in amplitude caused by tachyphylaxis. Denoted $p$ values are for post hoc Tukey's tests after one-way ANOVA ($p = 1.7E-11$, F = 56.1). **g** Sample sequential *Tad*NaC2 currents elicited by perfusion of pH 4.5 solutions bearing increasing concentrations of amiloride. **h** Average amiloride dose–response curve ($n = 9$) revealing a more pronounced decline in normalized peak inward current with increasing concentration of amiloride, compared to the rundown observed in the absence of drug attributable to tachyphylaxis.

and rate of decline between the different channels and conditions was not.

Next, we tested the sensitivity of $Tad$NaC2 to the general Deg/ENaC channel blocker amiloride, having previously found that the *T. adhaerens* $Tad$NaC6 channel was potently activated by this drug[24], a rare feature also reported for ASIC3 channels[36,37]. Application of 3 mM amiloride almost completely blocked mASIC1, but only partially blocked $Tad$NaC2, altering the current waveform such that the fast early current component was no longer evident (Fig. 4e). Given that $Tad$NaC2 and mASIC1 currents, respectively, decay by $19.1 \pm 11.6\%$ and $17.3 \pm 8.3\%$ upon successive activation, we reasoned that a component of the attenuated current amplitude in these experiments was attributable to tachyphylaxis. Subtracting the effect of tachyphylaxis to isolate the amiloride block of both channels reduced the decrease in average peak inward current from $90.3 \pm 4.1\%$ down to $73.0 \pm 4.1\%$ for mASIC1a, and from $60.5 \pm 9.5\%$ to only $41.4 \pm 9.6\%$ for $Tad$NaC2 (Fig. 4f). To better characterize amiloride block of $Tad$NaC2, we applied increasing concentrations of the drug while activating at pH 4.5, revealing a continuing decline in peak current amplitude coupled with a marked reduction in the fast transient current with amiloride concentrations greater than 1 mM (Fig. 4g). Although a component of the decline in current amplitude is likely due to the tachyphylaxis-like property of $Tad$NaC2, there is a marked difference in the current waveforms, in that tachyphylaxis does not markedly alter the macroscopic current waveform (Fig. 4c), while high concentrations of amiloride alter the kinetics of the macroscopic current such that the fast transient current is considerably inhibited (Fig. 4g). Although these observations suggest that amiloride has a more potent effect on the early compared to the late current, more detailed studies will be required to characterize this phenomenon. Analysis of the decline in peak current as a function of amiloride concentration reveals a more pronounced decay in amplitude compared to tachyphylaxis with amiloride concentrations greater than 100 μM, with an $IC_{50}$ of $52.0 \pm 29.6$ μM attributable to the combined effect of amiloride plus tachyphylaxis (Fig. 4h).

**$Tad$NaC2 currents are biphasic with different monovalent cation selectivities and are relatively insensitive to $Ca^{2+}$ block.** $Tad$NaC2 resembles mammalian ASIC3 in conducting biphasic macroscopic currents comprised of an early current that activates and desensitizes quickly, followed by a late current that activates and desensitizes more slowly[38]. These two components of the $Tad$NaC2 current become even more distinguishable at pH 3.5, where two separate peaks can be observed (Fig. 5a). We thus wondered whether these two components of the macroscopic current exhibit differences in their ion selectivity. To test this, we employed the bi-ionic reversal potential technique by perfusing different monovalent cations over recorded cells ($Li^+$, $Na^+$, and $K^+$), while maintaining equimolar $Na^+$ in the internal recording solution, and measuring changes in current reversal potential (voltage where currents reverse from inward to outward) when external permeating ions are altered[39]. This technique allows quantification of permeability ratios of desired cations relative to $Na^+$ ($pX^+/pNa^+$, where $X^+$ is the external cation). Recording $Tad$NaC2 currents at different fixed voltages at pH 4.5, with 150 mM $Na^+$ on each side of the cell membrane, produced slowly activating currents that lacked a fast transient component (Fig. 5b). As expected, these currents reversed from inward (negative) to outward (positive) near zero millivolts (i.e., $0.87 \pm 0.87$ mV; Fig. 5c). Replacement of extracellular $Na^+$ with an equal concentration of $Li^+$, which has a smaller ionic radius than $Na^+$, produced similar currents that reversed near 0 mV and

lacked a transient component ($2.51 \pm 0.72$ mV), indicating that $Tad$NaC2 is equally permeable to $Na^+$ and $Li^+$. Notably, all our previous recordings made using standard salines with external $Na^+$ and internal $K^+$ or $Cs^+$ ions produced biphasic currents at pH values below 5.5, unlike currents observed under the bi-ionic conditions of $Na^+_{In}/Na^+_{Out}$ and $Na^+_{In}/Li^+_{Out}$. Thus, it appears that the kinetics of the macroscopic current can differ depending on the types of permeating ions present across the cell membrane, an interesting observation that will require deeper analysis in future studies.

Instead, replacement of extracellular $Na^+$ with equimolar $K^+$ (i.e., $Na^+_{In}/K^+_{Out}$) produced canonical biphasic currents with a fast transient component and a late sustained component (Fig. 5b). The occurrence of these two clearly delineated current components allowed us to measure reversal potentials for each, revealing that although both exhibit a negative shift in voltage compared to bi-ionic sodium, the late current exhibited a more marked hyperpolarizing shift compared to the early current (i.e., $-60.42 \pm 2.30$ vs. $-49.01 \pm 1.62$ mV, respectively; Fig. 5c). A box plot of the reversal potential data for the different bi-ionic measurements, coupled with ANOVA and post hoc tests (Fig. 5d), substantiates the negative shift in reversal potentials for both the late and early currents in the presence of external $K^+$, reflecting a general preference of $Tad$NaC2 for $Na^+$ over $K^+$ ions. Furthermore, the more pronounced shift in reversal potential for the late vs. the early current indicates that ion selectivity changes over the course of the biphasic current, such that the early current is less selective for $Na^+$ over $K^+$ compared to the late current ($pNa^+/pK^+$ permeability ratios of $7.3 \pm 0.5$ and $11.0 \pm 1.1$, respectively; Fig. 5e).

Next, we sought to determine whether external $Ca^{2+}$ ions can block inward $Na^+$ currents through $Tad$NaC2. Perfusion of a pH 4.5 external solution containing 140 mM $Na^+$ and increasing concentrations of $Ca^{2+}$ revealed a sequential decline in current amplitude (Fig. 5f), which however was not statistically different from that attributed to tachyphylaxis (Fig. 5g). Nonetheless, 10 mM $Ca^{2+}$ appeared to cause a downward inflection in the dose–response curve (Fig. 5g), suggesting that at this higher concentration, $Ca^{2+}$ is able to mildly block $Tad$NaC2. We therefore designed a paired pulse experiment aimed at distinguishing the decline in current caused by tachyphylaxis, from that caused by 10 mM $Ca^{2+}$ block. Specifically, we applied paired pulses of pH 4.5 solutions containing either 0 mM of 10 mM $Ca^{2+}$ over recorded cells (Fig. 5h), and quantified the decline in peak current amplitude of the second pulse relative to the first (Fig. 5i). When both pulses lacked external $Ca^{2+}$, the peak current amplitude declined by $13.2 \pm 4.6\%$, while addition of 10 mM $Ca^{2+}$ to the second pulse resulted in a decline of $27.6 \pm 5.2\%$). Thus, 10 mM $Ca^{2+}$ exerts a low-affinity block of the $Tad$NaC2 current of roughly 14.4%.

**$Tad$NaC2 lacks the molecular determinants for proton activation of ASIC channels.** Deg/ENaC channels like ASIC channels are homo- and/or hetero-trimeric in nature, with each separate subunit forming a "ball in hand" tertiary structure comprised of wrist, palm, thumb, finger, knuckle, and β-ball regions (Fig. 6a). Cumulative efforts have uncovered several core molecular determinants for proton activation of ASIC channels, namely a critical histidine residue in the wrist region (H73 in mASIC1), and a lysine in the palm region (K211) situated at the extracellular interface between subunits (Fig. 6a)[25,40–43]. A protein alignment of several regions bearing these and other determinants for proton-activation of ASIC channels, including the group A ASIC channels from mice (i.e., ASIC1 to 4), selected group A and B channels from *Branchiostoma belcheri*[25], and the

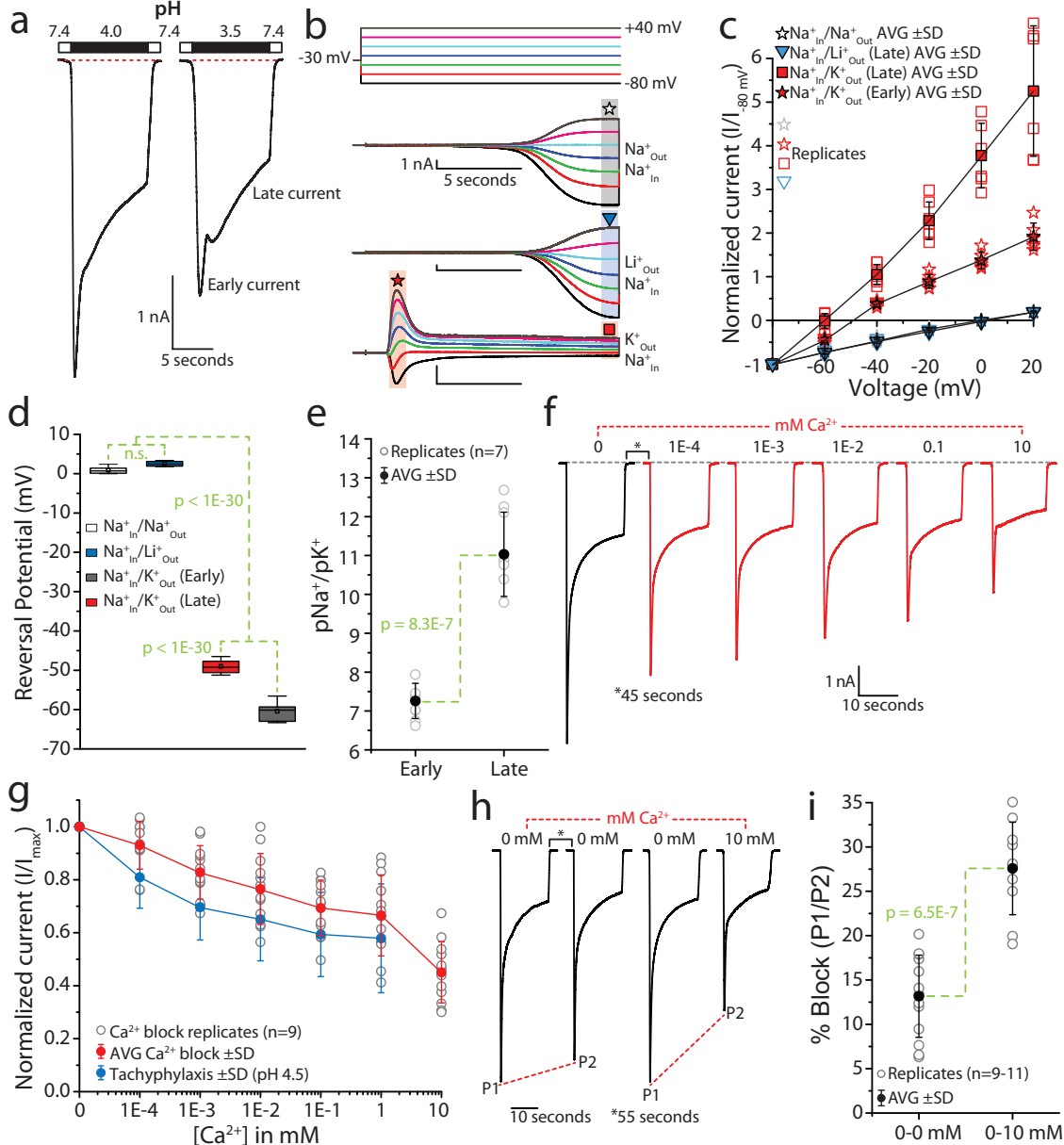

**Fig. 5 TadNaC2 conducts biphasic currents in vitro that are relatively insensitive to Ca²⁺ block. a** Sample current recordings of the TadNaC2 channel at pH 4 and 3.5, revealing a biphasic current with a fast transient component (i.e., early current), and a slower, sustained (late current) component. The biphasic current becomes more evident at pH 3.5. **b** Sample proton-activated TadNaC2 currents recorded at different voltages (voltage protocol on top), under bi-ionic conditions of equimolar intracellular Na⁺ and extracellular Na⁺ (Na⁺ ext.) or K⁺ (K⁺ ext.). The star and square symbols denote regions of the currents that were measured to determine reversal potentials. **c** Plot of average reversal potential data (± standard deviation) for the bi-ionic reversal potential experiments, revealing a leftward shift for both the early and late current components of bimodal currents in the presence of external K⁺ ($n = 6$–7) compared to external Na⁺ ($n = 7$) and Li⁺ ($n = 4$). **d** Box plot of average reversal potential data, showing statistically significant differences for both the early and late currents when extracellular Na⁺ was replaced with K⁺. The denoted $p$ values are from Tukey post hoc tests after one-way ANOVA ($p < 1E-30$, F = 2569). **e** Na⁺/K⁺ permeability ratios calculated using the bi-ionic reversal potential data, revealing that the late current exhibits higher Na⁺ selectivity compared to the early current ($p$ value is for a two-sample t-test). **f** Sample sequential TadNaC2 currents elicited by pH 4.5 solutions bearing increasing Ca²⁺ concentrations. **g** Average Ca²⁺ dose–response curve ($n = 12$) revealing a similar decline in normalized peak inward current with increasing Ca²⁺ concentration compared to tachyphylaxis (in constant 2 mM Ca²⁺). **h** Sample sequential paired currents elicited by pH 4.5 solutions bearing either 0 mM or 10 mM Ca²⁺ ions. **i** Plot of average percent block of peak inward current (e.g., 1 - P2/P1 from (**h**) ×100%) after switching from 0 mM Ca²⁺ to either 0 mM or 10 mM Ca²⁺ ($n = 13$ and 10, respectively). The denoted $p$ value is for a two-sample T-test.

singleton group B channel from *Lingula anatina*[10] reveals near complete conservation of the H73 and K211 residues (Fig. 6b). The only exception are the proton-insensitive ASIC2b splice variant which lacks H73[44], and ASIC4 which is also proton-insensitive and lacks K211[45]. Indeed, except for the zebrafish ASIC1 homolog zASIC1.1[10], all functional group A and B ASIC

channels that have been experimentally characterized in vitro bear a conserved H73 residue and most bear a K211 equivalent. The mouse ASIC5/BASIC channel, which falls in a separate clade from ASIC channels (Fig. 2) and is not activated by protons, lacks both the H73 and K211 residues. These residues are also absent in other Deg/ENaC channels that are sensitive to external protons

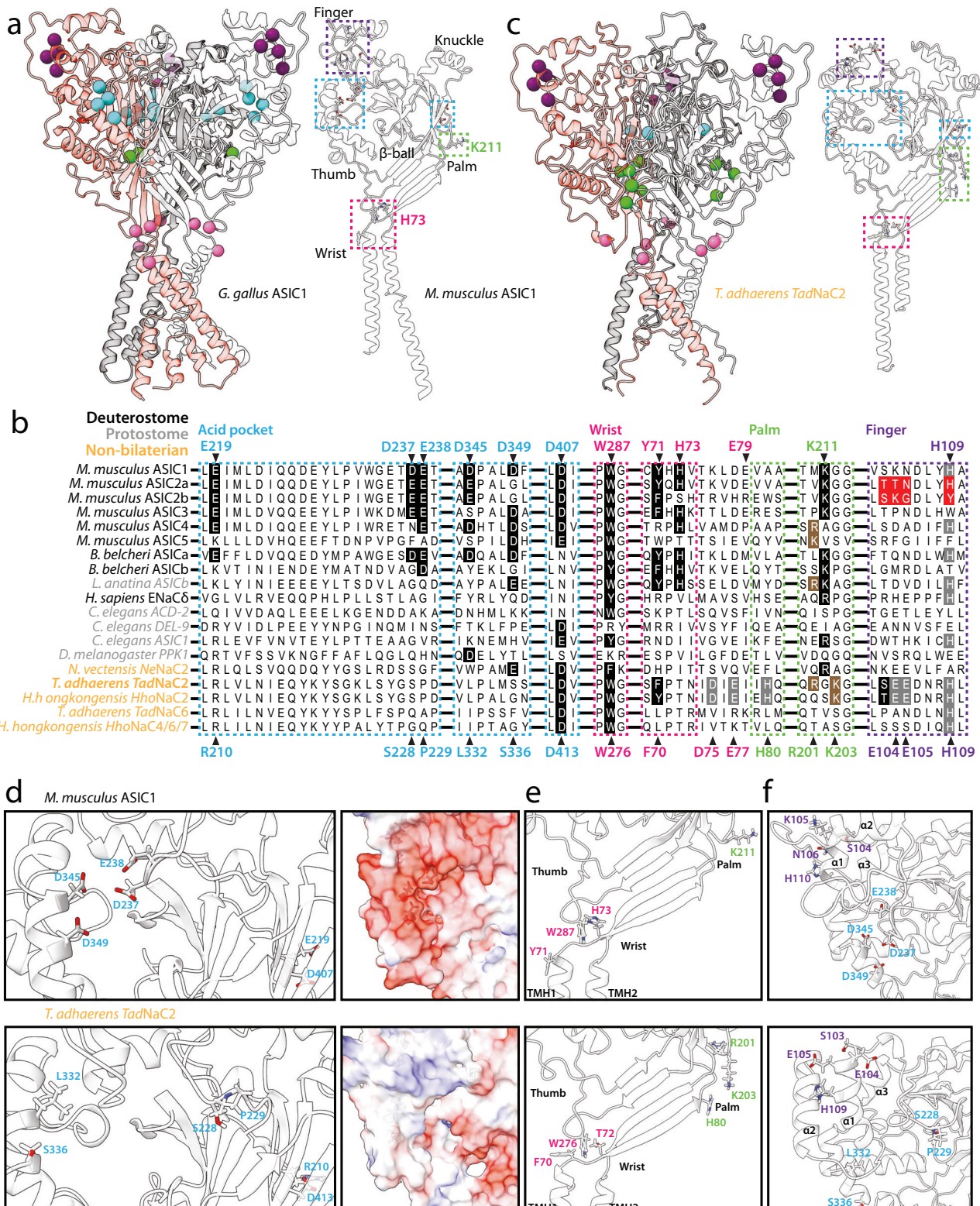

in vitro, including *Tad*NaC6, the proton-inhibited channel from *T. adhaerens*[24], and the proton-activated channels *Tad*NaC2, human ENaC-δ[6], *C. elegans* ACD-2, DEL-9, and ASIC-1[14], *D. melanogaster* Pickpocket1[16], and *Ne*NaC2 from the sea anemone *Nematostella vectensis*[8]. However, *Tad*NaC2, as well as the mouse ASIC4 and BASIC channels, possess a cationic residue just one amino acid upstream of the K211 position (i.e., R201 in

*Tad*NaC2). *Tad*NaC2 and its *Hoilungia hongkongensis* orthologue *Hho*NaC2 also possess a conserved lysine one position downstream (K203 in *Tad*NaC2). Furthermore, all of the ASIC channels except for the non-functional ASIC4 isotype bear a conserved aromatic residue 2 positions upstream of H73 (i.e., Y71). Y71 forms an aromatic bridge with a conserved tryptophan (W287) in mouse ASIC1a, shown to be important for coupling

**Fig. 6 Structural features of the *Tad*NaC2 channel. a** Ribbon diagrams of the chick ASIC1 homotrimeric channel crystal structure (left, PDB number 6VTK), and the AlphaFold-predicted tertiary structure of the mouse ASIC1a subunit (right). The three separate subunits of the homotrimeric channel are colored in red, white, and gray, and the colored circles denote the β carbon atoms of critical residues corresponding to the back-colored residues of the mouse ASIC1a channel in the protein alignment shown in (**b**) (i.e., atoms in blue are within the acid pocket, pink are within the wrist, green are within the palm, and purple are within the finger). The dashed boxes denote structural regions of the single mASIC1a subunit structure bearing these same critical residues. **b** Protein sequence alignment of the acid pocket (enclosed by blue dashed boxes), wrist (pink), palm (green), and finger (purple) regions of select proton-activated Deg/ENaC channels from cnidarians and bilaterians with *Tad*NaC2, *Tad*NaC6, *Hho*NaC2, and *Hho*NaC4/6/7 channels from the placozoans *Trichoplax adhaerens* and *Hoilungia hongkongensis*. Residues that are back-colored in black represent conserved residues for proton activation of ASIC channels, while those back-colored red denote key residues that render the ASIC2b splice variant insensitive to external protons. Residues that are back-colored in gray denote protonatable amino acids in *Tad*NaC2 within these key structural regions, some of which are conserved in cnidarian and bilaterian homologs, while those back-colored in brown denote cationic residues in *Tad*NaC2 that flank the critical K211 residue of ASIC channels, also found in several other channels. Notable is the complete conservation of the critical residues H73 and K211 in all included ASIC channels, and their absence in most non-ASIC proton-activated channels including *Tad*NaC2. **c** Homology model of the homotrimeric *Tad*NaC2 channel structure (left), and AlphaFold-predicted structure of the single subunit, with a similar annotation as described for (**a**). **d** Left panels: Close-up view of the acid pocket region of mASIC1a (top) and *Tad*NaC2 (bottom) within corresponding AlphaFold-predicted structures. The six rendered residues in the *Tad*NaC2 channel correspond to residues that align with the six acid pocket residues in mASIC1a as depicted in (**a**). Right panels: Surface rendering of the acid pocket region of mASIC1a (top) and *Tad*NaC2 (bottom) reveals a stark difference in the electrostatic potential between the two channel subunits. **e** Close-up view of the wrist and palm regions of mASIC1a and *Tad*NaC2. Apparent in the wrist region is the absence of a critical H73 proton-sensing residue in *Tad*NaC2, but conservation of the aromatic amino acids F70 and W276, which in mASIC1a (i.e., Y71 and W278) form an aromatic bridge critical for channel gating. Instead, *Tad*NaC2 bears a putative proton-sensing amino acid (H80) at the opposite end of a β strand that projects from the first transmembrane helix in wrist region (TMH1) to the palm domain, placing it near the residues R201 and K203 that flank the critical K211 residue of mASIC1a. **f** Close-up view of the finger and acid pocket regions, with rendered amino acids corresponding to the positions in the ASIC2b splice variant that make the channel insensitive to protons. Also labeled are the equivalent acid pocket residues, and the predicted α1 to α3 helices in the finger region.

conformation changes in the extracellular domain with gating of the pore helices[46]. This aromatic residue is absent in all included non-ASIC channels except for *Tad*NaC2 and *Hho*NaC2 which bear phenylalanine and tyrosine residues at this position, respectively (i.e., F70 in *Tad*NaC2), as well as a tryptophan corresponding to W287 (i.e., W276 in *Tad*NaC2). Also notable is that *Tad*NaC2 possesses several protonatable amino acids that are in proximity of the ASIC H73 position, with aspartate and glutamate residues at positions 75 and 77, and a histidine at position 80 that aligns with residues in the palm region placing it proximal to the noted R201 and K203 residues.

Another region associated with proton activation (and desensitization) of ASIC channels is the acid pocket, comprised of a cluster of four acidic amino acids located between the finger and thumb regions of the subunit monomer, and another pair in the palm region close to K211 (Fig. 6a, b). In the trimeric channel, the four acidic residues from one subunit and two from an adjacent subunit combine to form a namesake pocket-like structure where protons are thought to bind and affect channel conformation and gating[26]. Of note, mutation of these glutamate/aspartate residues does not completely disrupt proton activation, and instead, these appear to be more important for channel desensitization[26]. Accordingly, the group B ASIC channels from *B. belcheri* and *L. anatina* lack most glutamate/aspartate residues in the acid pocket (Fig. 6b), while they are largely conserved among the group A channels. Furthermore, the two *Tad*NaC channels, as well as the various other non-ASIC Deg/ENaC channels included in the alignment lack most if not all acidic residues at equivalent positions of the acid pocket.

A third region of interest with respect to proton activation is the finger region, where a motif of four amino acids distinguishes the proton-sensitive ASIC2a mRNA splice variant from the insensitive ASIC2b variant (Fig. 6b). Specifically, ASIC2a bears a motif of TTN-XXX-H and is proton-activatable, while ASIC2b bears an SKG-XXX-Y motif and is not[47]. Moreover, introducing the SKG and Y elements of the ASIC2b motif into ASIC2a, together but not separately, abrogates proton activation, and insertion of the finger region of ASIC2a into ASIC1 causes a marked reduction in proton sensitivity making the latter less sensitive to protons similar to the ASIC2a channel[48]. Notably, a histidine residue within the finger region of ASIC2a (H109) is conserved among many of the included Deg/ENaC channels including *Tad*NaC2 (Fig. 6b). However, its functional significance remains unclear, with one mutation study reporting no effect on ASIC2a activation[42], and a subsequent study reporting a contribution but not a requirement[47]. In this region, *Tad*NaC2 also bears two protonatable glutamate residues (E104 and E105).

To better infer how the structure of *Tad*NaC2 compares to the well-studied structures of ASIC channels, we generated a homology model of the homotrimeric channel using the crystal structure of chick ASIC1 as a template (Fig. 6c; left panel)[49]. We also predicted the tertiary structures of the monomeric mouse ASIC1 and *Tad*NaC2 channel subunits with AlphaFold (Fig. 6a, c; right panels)[50]. Labeling the β carbon atoms of the Y71, W287, H79, K211, acid pocket, and ASIC2 finger motif equivalents in the homotrimeric structure of the chick ASIC1 channel (Fig. 6a), and the F70, W276, H80, R201, K203, D413 (single acid pocket residue), and finger motif equivalents in the model of the *Tad*NaC2 homotrimer (Fig. 6c), illustrates the general absence of acid pocket residues in *Tad*NaC2. Also evident are the noted differences in the wrist region, where *Tad*NaC2 lacks an H73 equivalent, and in the palm, where the residues H80, R201, and K203 in *Tad*NaC2 are arranged in a triangular cluster at the interface between subunits, in a similar position as the K211 residue in ASIC channels. Furthermore, the aromatic residues F70 and W276 residues in *Tad*NaC2 are in proximity to each other, suggesting that like Y71 and W287 in ASIC1, these can form hydrophobic interactions.

The predicted structures of the mASIC1 and *Tad*NaC2 monomers also highlight key differences and similarities. First, whereas the cluster of four acid pocket residues in mASIC1 (D237, E238, D345, and D349) are arranged in a tight cluster, the equivalent residues in *Tad*NaC2 are not (S228, P229, L332, and S336) (Fig. 6d, left panels). Rendering the electrostatic potential on the surface of the two channel subunits also illustrates a stark difference in the acid pocket region, with the acidic residues of mASIC1a contributing to a highly electronegative surface, while those in *Tad*NaC2 contribute to a surface that is slightly positive and hence unlikely to attract and bind $H^+$ ions (Fig. 6d, right panels). In the wrist region, the W276 sidechain at the base of the

thumb of $Tad$NaC2 is in a similar position as the W287 side-chain in mASIC1, situated between the Y71 and H73 equivalent residues F70 and threonine 72 (Fig. 6e). Furthermore, both the SKN-XXX-H and SEE-XXX-H finger motifs of mASIC1 and $Tad$NaC2 are within a short loop and adjacent descending alpha helix, consistent with the α1 helix identified in the crystal structure of the chick ASIC1a finger region[7]. However, this helix is predicted to be two helical rotations longer in $Tad$NaC2, with a short loop connecting it to the downstream α2 helix that is also longer than its predicted counterpart in mASIC1 by one rotation (Fig. 6f). Lastly, it is notable that the finger regions of the two channels are positioned above the divergent acid pocket, suggesting that any structural alterations taking place in the finger region would be differentially transferred to the thumb and pore regions that mediate channel gating.

**Residues in the palm and finger region contribute to $Tad$NaC2 proton gating.** Despite lacking key deterministic residues for proton activation, the similar predicted structure of $Tad$NaC2 compared to mASIC1a prompted us to examine whether corresponding structural regions in the placozoan channel bear unique or conserved elements involved in channel gating. Thus, we performed site-directed mutagenesis on selected aromatic or protonatable residues in the wrist region (F70, D75, and E77), protonatable residues in the finger region (E104, E105, and H109), and protonatable or cationic residues in the palm region (H80, R201, and K203) (Fig. 6a). To assess changes in $H^+$ sensitivity and gating properties at different pH, we tested each mutant with a series of perfused solutions of various pH to generate dose–response curves of recorded macroscopic currents (Fig. 7a–c; plots of individual variants with replicates provided in Supplementary Fig. 1).

In the wrist region of rat ASIC1a, mutation of the Y71 aromatic residue to a histidine imposes a ~70% reduction in elicited current amplitude, while mutation to alanine completely disrupts activation by protons[46]. In contrast, analogous mutations of the F70 residue in $Tad$NaC2 had negligible effects on the $pH_{50}$ (Fig. 7d, e), and no effect on average peak inward current amplitude at pH 4.0 compared to the wild-type channel (Fig. 7e). Thus, this residue in $Tad$NaC2 most likely does not form a functionally analogous aromatic interaction with the conserved W276 residue in the thumb region, akin to the Y71-W287 interaction in ASIC1 channels. As noted, $Tad$NaC2 bears two protonatable residues in the wrist region (D75 and E77), within a predicted β strand that in ASIC channels projects from the H73 residue in the wrist towards the K211 residue in the palm (Fig. 6a, b). The E77 residue in $Tad$NaC2 aligns with D78 in ASIC1a and ASIC2a, which when mutated to asparagine in the rat channels disrupts proton activation[41,43]. In contrast, alanine substitution of E77 in $Tad$NaC2 had no noticeable effect, while mutation of the D75 residue two positions upstream caused a marked *increase* in the $pH_{50}$ (Fig. 7d, e). Furthermore, neither the D75A nor the E77A mutation caused a change in maximal peak current amplitude (Fig. 7f), or in the overall shape of macroscopic currents (Fig. 7f). Overall, $Tad$NaC2 appears different from ASIC channels in lacking homologous molecular determinants in the wrist region that are involved in proton gating.

In the finger region, single mutations of E104A and E105A had no effect on $pH_{50}$ or peak current amplitude (Fig. 7b, d, e). However, mutation of both together caused a moderate decrease in both metrics, and altered the macroscopic current waveform by diminishing the fast/early component (Fig. 7f). A more dramatic effect occurred for the single mutation H109A, which in addition to reducing maximal peak current amplitude (Fig. 7e), produced a biphasic macroscopic current and a complete loss of the early

current component at pH 4.5 and 4.0 (Fig. 7b, f; Supplementary Fig. 1h). Interestingly, mutation of the acid pocket residue D345 in mouse ASIC1a (Fig. 6b), which is close to the predicted finger region of $Tad$NaC2 where H109 resides (Fig. 6f), also imposes biphasic sensitivity to pH, attributed to the loss of one of two separate proton binding sites involved in channel activation (the other being in the palm domain)[51]. However, the biphasic effect is much more severe for the $Tad$NaC2 H109A mutation, where instead of plateauing between pH 5.0 and 4.0 like mAIC1a, the current amplitude first decreases from pH 5.0 to 4.5, then increases again from pH 4.5 to 4.0 (Fig. 7b, f, Supplementary Fig. 1h). This atypical feature precluded accurate fitting of the dose–response data with either standard or biphasic dose–response curves, since both serve to model strictly incremental processes (i.e., $R^2$ values of 0.64 and 0.68, respectively). Nonetheless, imposing a standard dose–response curve over the data revealed reduced sensitivity to protons compared to the wild-type channel, with an average $pH_{50}$ of 4.7 ± 0.2 vs. 5.2 ± 0.1 (Fig. 7d). Instead, fitting the data with a bimodal dose–response curve produced a $pH_{50-1}$ value of 5.5 ± 0.3 and a $pH_{50-2}$ value of 4.5 ± 0.3, both of which are statistically different from wild type (i.e., $P$ values for two-sample t-Tests ≤0.005 and 0.0005, respectively). However, since the H109A variant shows diminished sensitivity to protons at the threshold pH of 5.5 (Supplementary Fig. 1h), this mutant channel is not likely more sensitive to protons at threshold pH values, but rather, has an overestimated the $pH_{50-1}$ value caused by the poor curve fit. Of note, while macroscopic current amplitudes varied considerably for the wild-type channel, the $pH_{50}$ values were less variable (Fig. 7d, e). Furthermore, we found no correlation between current amplitude and $pH_{50}$ for the wild-type channel, altogether indicating that observed differences in pH sensitivity for the H109A mutant and other channels variants was not due to altered current amplitudes. It is also notable that the transient current observed at pH 5.0 desensitized more rapidly compared to the wild-type channel, while at more acidic conditions the transient current was absent leaving only a slowly activating sustained current that increased in amplitude from pH 4.5 to 4.0 (Fig. 7f). The most severe of all mutations tested was a triple mutation E104A/E105A/H109A, which produced a channel with very weak activation at pH 5.5 and 4.5, completely lacking transient/early currents at all tested pH (Fig. 7f). This resulted in the most significant change in proton gating with an average $pH_{50}$ of 4.4 ± 0.1 (Fig. 7d, e). Altogether, it appears as though the H109 residue, together with E104 and E105, plays an important role in the proton gating of $Tad$NaC2.

In ASIC1a, deletion of the K211 palm residue results in a strong decrease in proton sensitivity, while mutation to glutamate causes a more moderate effect[25]. In $Tad$NaC2, mutation of the two cationic residues that flank the K211 position, R201 and K203, produced an increase in proton sensitivity with respective $pH_{50}$ values of 5.3 ± 0.1 and 5.5 ± 0.0 (Fig. 7c, d). Notably, the R201A mutation also altered the macroscopic current waveform such that the amplitude difference between the early and late components was greater at pH 5.0 and 4.5 compared to wild-type, but not at pH 4.0 (Supplementary Fig. 2). Instead, the K203E mutation disrupted the early current such that it only became evident at very acidic pH values (Fig. 7f). Deletion of this same residue (K203Δ), to emulate K211Δ variants of ASIC channels, resulted in an inability to detect currents even with very acidic pH. Alanine substitution of the unique protonatable H80 residue in the palm region, which is proximal to R201 and K203 in our predicted structures (Fig. 6), caused the dose–response data to become more variable, and the pH sensitivity to become biphasic similar to the H109A mutation (Fig. 7c, f; Supplementary Fig. 1j). Furthermore, and like the K203E mutation, the H80A mutation

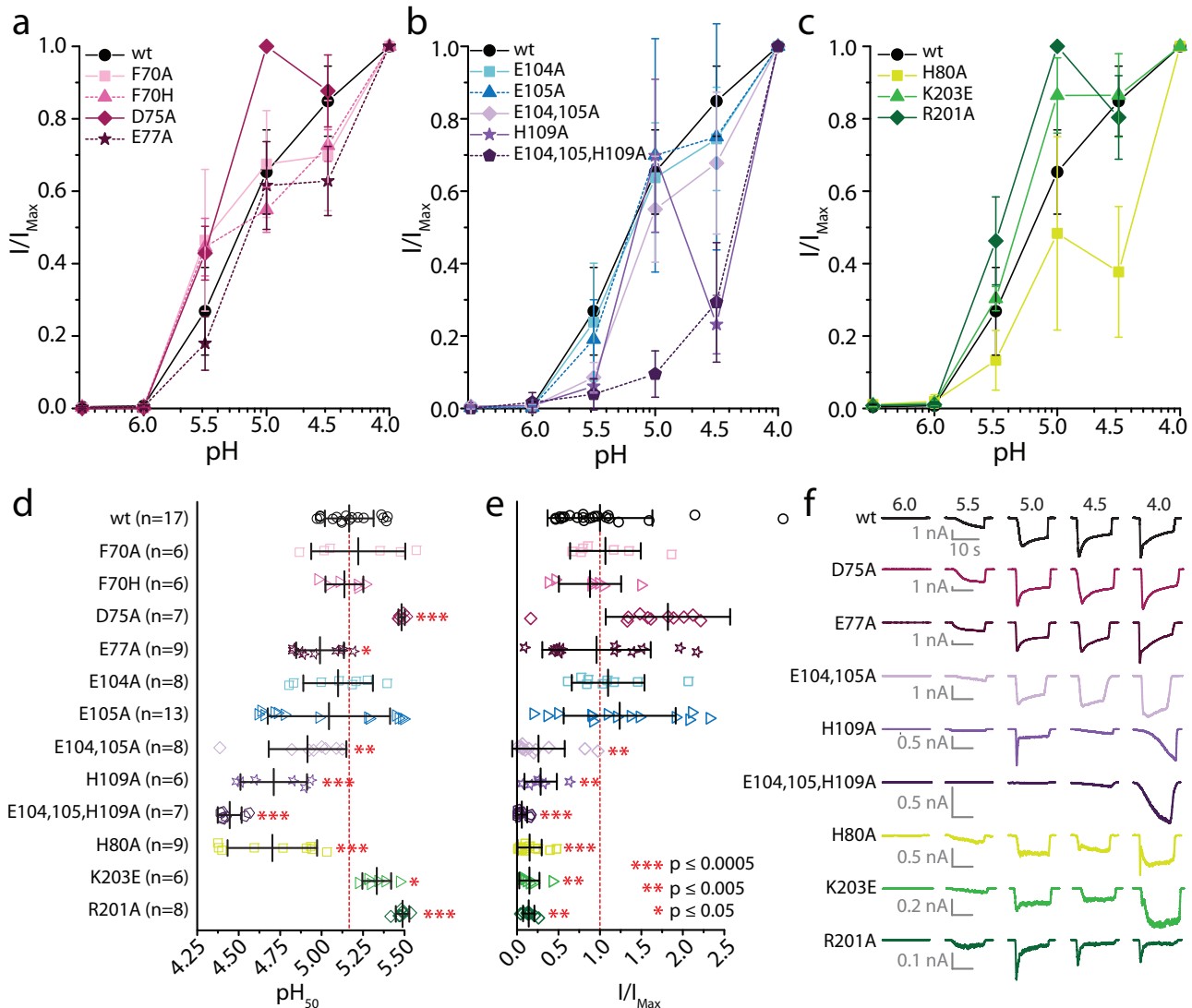

**Fig. 7 Unique residues contribute to proton gating of *Tad*NaC2. a–c** Average pH dose–response curves ± standard deviation for wild-type (wt) *Tad*NaC2 and variants bearing amino acid substitutions within the wrist (**a**), finger (**b**), and palm (**c**) regions. **d**, **e** Plot of average ± standard deviation $pH_{50}$ (**d**) and normalized peak current amplitude (**e**) for wild-type (wt) and various point mutated *Tad*NaC2 channels. Red-colored asterisks denote *p* value thresholds for two-sample t-tests comparing wild-type to mutant values. **f** Sample whole-cell currents of wild-type *Tad*NaC2 and select mutant variants.

strongly disrupted the transient current, which was only evident under very acidic pH conditions (Fig. 7f). Fitting a standard dose–response curve over the data revealed a decrease in pH sensitivity compared to wild-type (i.e., $pH_{50} = 4.7 \pm 0.3$; $R^2$ for global fit = 0.91). Instead, a biphasic curve fit produced a $pH_{50-1}$ value that was not statistically different from wild-type ($pH_{50-1} = 5.1 \pm 0.7$), but a $pH_{50-2}$ value that was considerably lower ($4.3 \pm 0.3$; *P* value for two-sample t-Test ≤0.0005; $R^2$ for global fit = 0.96). Of note, mutation of a glutamate residue in mouse ASIC1a, just two amino acids upstream of H80 in our protein alignment and in a similar region of the palm domain (Fig. 6b, e), also imposed a biphasic sensitivity to pH[51]. Altogether, these observations indicate that the H80 residue also plays a role in the proton gating of *Tad*NaC2. Furthermore, the R201 and K203 residues also contribute to *Tad*NaC2 gating, however, their mutation did not produce a rightward shift in the pH dose–response curve as it did for the analogous K211 residue in ASIC channels[25], indicating key functional differences. Finally, all tested mutations in the palm region caused a significant decrease in maximal current amplitude (Fig. 7e), most extreme

for the K203Δ variant which was either completely non-functional, not trafficking to the cell membrane, or both.

Next, we wanted to determine whether the noted decrease in current amplitude caused by select mutations was due to reduced functionality or a reduction in channel protein expression. Hence, we N-terminally tagged the wild-type channel with enhanced green fluorescent protein (EGFP), as well as the mutants H80A, H109A, and K203Δ that, respectively, caused moderate, strong, and severe effects on current amplitude. This permitted inference of the total channel protein levels in transfected CHO-K1 cells via EGFP fluorescence quantification, relative to a co-transfected blue fluorescent protein from the empty vector pIRES2-EBFP. Of note, we tested whether tagging the wild-type *Tad*NaC2 channel with EGFP disrupted its function, finding it to conduct proton-activated currents that were visually indistinguishable from the untagged channel (Supplementary Fig. 3). Fluorescence micrographs of transfected cells reveal a noticeable decrease in EGFP fluorescence of all three mutant channels relative to wild type (Fig. 8a), with respective normalized average integrated density values of $67 \pm 5\%$, $43 \pm 4\%$, and $57 \pm 7\%$ for the H80A, H109A,

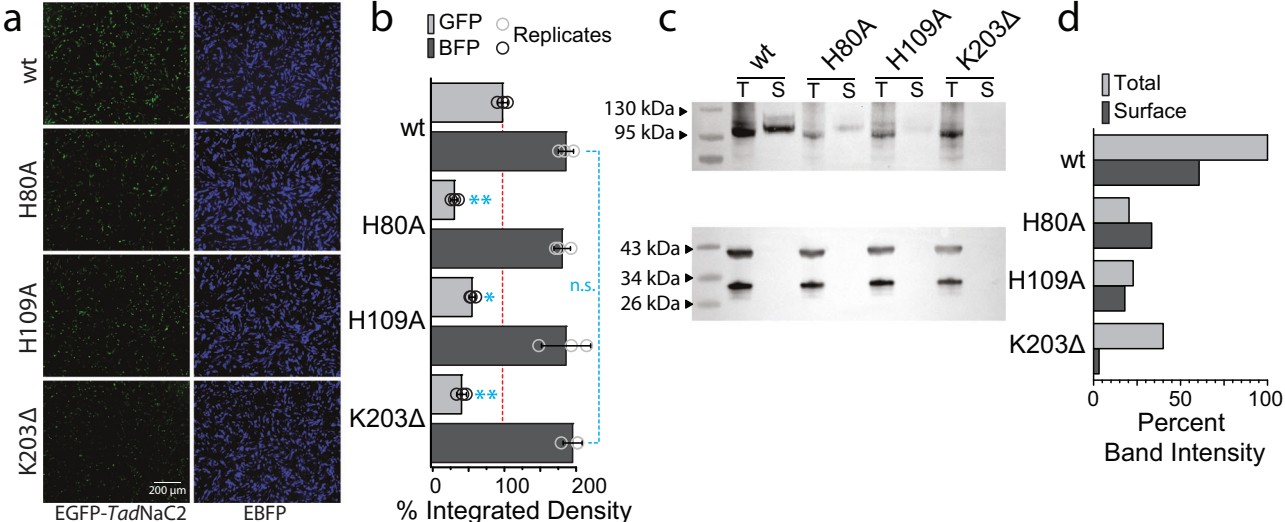

**Fig. 8 Mutations impose reduced expression and membrane trafficking of *Tad*NaC2 channels in vitro. a** Representative fluorescence micrographs of CHO-K1 cells co-transfected with pEGFP-*Tad*NaC2 fusion vector (left panels) and an empty pIRES2-EBFP vector (right panels). **b** Plot of percent average integrated density ± standard deviation, quantifying the emitted fluorescence of pEGFP-*Tad*NaC2 wild type (wt) and mutant channels, normalized to the average integrated density of wild-type *Tad*NaC2 (n = 3 for each transfection condition). EBFP fluorescence was also quantified to determine transfection efficiency. Cyan-colored asterisks denote p value thresholds for Tukey post hoc means comparisons of fluorescence signals between wild-type and mutant channels after one-way ANOVAs (EGFP: $p = 5.6E-11$, F = 73.6; EBFP: not significant). **c** Top panel: Western blot of select EGFP-tagged *Tad*NaC2 channel variants in CHO-K1 cell lysates using anti-GFP polyclonal antibodies, comparing total channel protein content (T) with membrane/surface expressed channel protein content (S) for each variant. Bottom panel: Western blot of the lower half of the membrane used in the top panel, using anti-GAPDH (*top* bands) and anti-EBFP (bottom bands), polyclonal antibodies. **d** Quantified band intensity (mean gray area) of *Tad*NaC2 bands in (**c**), relative to the wild type EGFP-*Tad*NaC2 total protein band, revealing decreased total and surface protein expression of *Tad*NaC2 channels bearing mutations, and a near complete absence of membrane expressed variants harboring a K203 deletion, consistent with our inability to record current for this channel in vitro. Bands for each channel variant were also normalized to the intensity of EBFP present in corresponding total protein lanes.

and K203Δ mutants (Fig. 8b). Average integrated density measurements for the co-transfected EBFP were statistically indistinguishable for all transfections, indicating that the differences in EGFP fluorescence were not due to differences in transfection efficiency. Thus, all three of the tested mutations cause a decrease in total protein expression in vitro.

Using a cell surface biotinylation strategy, we also wanted to characterize the effect of mutations on total protein and membrane expressed protein levels in transfected cells. A Western blot probed with anti-EGFP antibodies revealed a marked reduction in both total protein and membrane expressed (surface) protein levels of mutant *Tad*NaC channels relative to wild-type (Fig. 8c). Measurements of the mean gray value of the different bands on the blot reveals similar reductions in total protein levels for all three mutants, and notably, extreme reduction of membrane expressed K203Δ (Fig. 8d). Altogether, this data is consistent with our current amplitude measurements and inability to record currents for the K203Δ variant of *Tad*NaC2.

## Discussion

Together, our CLANS and phylogenetic analyses provide evidence for the existence of two major clades of metazoan Deg/ENaC channels (Figs. 1 and 2), A and B, in agreement with several previously published clustering and phylogenetic analyses[8,14,19]. Most animals possess Clade A and B channels, indicating that these two subfamilies emerged early during animal evolution. The exception are ctenophores, which lack group B channels. Thus, depending on whether poriferans or ctenophores were the first to diverge from other animals[52–55], ctenophores either lost Clade B channels, or these emerged in other animals after ctenophores diverged[8]. Alternatively, the ctenophore Clade B channels

underwent extreme sequence divergence obscuring their phylogenetic history. Indeed, ctenophores possess two groups of channels that did not associate with other Deg/ENaC channels in our CLANS analysis, and similar divergent clusters were apparent for cnidarians and platyhelminths. Future studies with increased taxon sampling will help determine whether these divergent channels are phylogenetically-related to Clade A or B channels.

In this study, we provide phylogenetic evidence that Deg/ENaC channels are not unique to animals, being present in unicellular eukaryotic lineages of Heterokonta and the filasterea-related species *T. unikontum*. Although our search was not exhaustive, we were unable to find Deg/ENaC sequences in the intervening opisthokont lineages that separate Metazoa and Filasterea, including choanoflagellates, or between these lineages and Heterokonta. This indicates that there was either extensive gene loss of Deg/ENaC channels in the intervening lineages of single-celled eukaryotes, or alternatively, lateral gene transfer between animals, filastereans, and heterokonts. Interestingly, similar phylogenetic gaps are apparent for other major cation channels shared between animals and unicellular eukaryotes. For example, Ca$_V$3 voltage-gated calcium channels, conserved between animals and choanoflagellates, are also found in algae but not in the vast eukaryotic lineages that fall between these organisms[56]. Similarly, the metazoan Na$^+$ leak channel NALCN and its extracellular subunit FAM155A have homologs in fungi but no lineages in between[56,57]. Hence, it is conceivable that Deg/ENaC channels arose in the metazoan ancestor through lateral gene transfer, perhaps giving rise to the ENaC family, and that the ASIC family evolved thereafter. Alternatively, Deg/ENaC channels first evolved in animals and then transferred to select eukaryotes.

Like Aguilar-Camacho et al.[8], we found that most placozoan Deg/ENaC channels form a monophyletic clade within Clade A

that has a sister relationship with BASIC channels, and together, these have a sister relationship with ASIC channels. Also similar, the singleton placozoan channels *Tad*NaC10 and *Hho*NaC10 fall into Clade B, among the chordate ENaC sodium leak channels, the *C. elegans* mechanically-gated channels MEC-10 and MEC-4, the *C. elegans* proton-gated channel ASIC-1, and the lophotrochozoan peptide-gated channels FaNaC and WaNaC.

Our phylogenetic analysis also identified several clades of bilaterian Deg/ENaC channels with unknown properties that are phylogenetically proximal to channels with described functional and/or physiological properties. In Clade A subclade I, we identified several groups of chordate (cephalochordate and urochordate), ambulacrarian, and lophtorochozoan channels that form strong phylogenetic relationships with the BASIC channels, the *Tad*NaC channels, and the proton-sensitive channels ACD-1, ACD-5, FLR-1, and ACD-2 from *C. elegans*. In Clade A subclade II, we identified groups of ambulacrarian, protostome, ctenophore, and poriferan channels among the PPK channels from *D. melanogaster*, the peptide-gated *Hy*NaC channels from the cnidarian *H. magnipapillata*, and the *Ne*NaC channels from *N. vectensis*. Similarly in Clade B, we identified numerous groups of uncharacterized channels from a broad range of animals among the ENaC channels, the *C. elegans* channels MEC-10, MEC-4, and ASIC-1, and the lophotrochozoan FaNaC and WaNaC channels. Future efforts aimed at determining the gating and physiological properties of channels within these various uncharacterized clades, especially from the earliest diverging animal lineages and unicellular eukaryotes, will be important as we seek to fill gaps in our understanding of Deg/ENaC channel evolution.

Initially, ASIC channels were thought to be unique to chordates[40,58,59], however, more recent studies indicate a much broader presence, with the identification of group A and B ASIC channels thought to have emerged in an early bilaterian ancestor[10]. In our sequence alignments, all included group A and B ASIC homologs that are proton-sensitive bear the quintessential H73 and K211 residues, considered core determinants for ASIC channel proton-activation (Fig. 6)[25]. Also fully conserved are the pair of aromatic amino acids, W287 and Y71 in mouse ASIC1a, which form a hydrophobic bridge between the thumb and wrist regions essential for channel gating[46]. Instead, the protonatable glutamate and aspartate residues that make up the acid pocket are absent in group B ASICs, indicating that the acid pocket is a unique feature of group A channels, altogether consistent with the dispensability of the acid pocket for proton activation[26].

Numerous non-ASIC Deg/ENaC channels have been identified that are also activated by extracellular protons but lack most key residues involved in proton-activation of ASIC channels. This includes *Tad*NaC2, the human ENaC-δ channel[6] and the *C. elegans* channels ASIC-1, ACD-2, DEL9, and[14], which form homotrimeric channels in vitro that conduct slow onset proton-activated currents with minimal desensitization, the *Ne*NaC2 and *Ne*NaC14 channels from *N. vectensis*, which, respectively, conduct moderately and non-desensitizing currents in vitro[8], and the *D. melanogaster* channel PPK1, which conducts transient, fast desensitizing cation currents in sensory neurons[16]. Notably, ASICs, *Tad*NaC2, *N. vectensis* *Ne*NaC2 and *Ne*NaC14, chordate ENaC-δ, and *C. elegans* ASIC-1/ACD-2 are all separated by intervening lineages of Deg/ENaC channels that are not known to be proton-activated (Fig. 2). One possible explanation for this is that proton-activation evolved numerous times independently, which is supported by the absence of key molecular determinants for proton-activation of ASIC channels in all non-ASIC proton-activated channels (Fig. 6b). Alternatively, the ancestral channel that gave rise to Clade A and B channels was proton-activated,

and this gating feature was then lost in the various lineages that are not proton-activated, including BASIC channels, *Hy*NaC channels, *Tad*NaC6, FaNaC and WaNaC channels, and the *C. elegans* channels MEC-4 and MEC-10. Although neither of these scenarios can be discounted with certainty, it is clear that future functional characterization, particularly of channels from lineages with unknown gating properties, will be important for better understanding Deg/ENaC channel evolution.

Here, we provide a detailed functional characterization of *Tad*NaC2, revealing a functional resemblance to ASIC channels and a conglomeration of isotype-specific biophysical features. With respect to gating, *Tad*NaC2 more resembles ASIC2a in having a lower $pH_{50}$ value of ~5 compared to ASIC1a and ASIC3 channels, reflecting reduced proton-sensitivity (Fig. 4a, b)[33]. Instead, the biphasic macroscopic currents of *Tad*NaC2, consisting of early transient and a late sustained components (Fig. 5a), resemble those of ASIC3[38]. Qualitatively, these two current components of *Tad*NaC2 appear to be differentially sensitive to amiloride, with 3 mM amiloride causing a marked reduction of the early current, but less of the sustained current (Fig. 4e, g). Lastly, the observed rundown of *Tad*NaC2 currents in response to repeated activation, or tachyphylaxis, is a unique negative feedback feature of ASIC1a channels with an undetermined physiological function[34,35]. Another interesting feature of the biphasic *Tad*NaC2 currents are their different ion selectivities, with the early current being less selective for $Na^+$ over $K^+$ ions compared to the late/sustained current (Fig. 5). Dual cation selectivity has also been reported for the early and late current components of the mammalian ASIC3/ASIC2b heteromeric channel[44], as well as the homomeric shark ASIC1b channel[40]. However, the monovalent selectivity profiles reported in these studies are reversed relative to *Tad*NaC2, with the transient current being more $Na^+$ selective than the sustained.

Our work provides insights into the molecular determinants for proton activation of a non-ASIC Deg/ENaC channel. Focusing our attention on channel regions that are critical for ASIC channel gating enabled us to uncover fundamental insights into how *Tad*NaC2 operates and differs from ASIC channels. In the wrist region, *Tad*NaC2 and *Hho*NaC2 channels bear conserved aromatic residues W276 and F70 which align with the W287 and Y71 residues of ASIC channels (Fig. 6b). Nevertheless, alanine mutation of the F70 residue in *Tad*NaC2, which renders ASIC1a channels non-functional (i.e., Y71A)[46], had no effect on proton-activation (Fig. 7a, d). In ASIC channels, the aromatic interaction between W287 and Y71 couples conformational changes that occur in extracellular regions such as the thumb and acid pocket, with the first transmembrane helix of each subunit that contributes to the pore in the holomeric channel. This, combined with the absence of an H73 equivalent in *Tad*NaC2, indicates a fundamental difference in gating between *Tad*NaC2 and ASIC channels. This notion was also supported by the E77A mutation in *Tad*NaC2, which unlike mutation of the aligned residue D78 in ASIC1a and ASIC2a channels[41,43], did not impact proton activation, and the upstream D75A mutation, that enhanced activation by increasing $pH_{50}$ (Fig. 7a, d). Notably, the cnidarian channel *Ne*NaC2 also lacks an H73 equivalent, but bears a histidine two positions upstream, aligning with residue Y71 in ASIC channels (Fig. 6b). However, mutation of this residue does not disrupt proton activation[8], indicating that like *Tad*NaC2, this channel differs from ASIC channels in lacking critical determinants for proton-gating within the wrist region.

In the palm region, we found that the H80 residue in *Tad*NaC2 plays an important role in proton activation, wherein its mutation caused a significant reduction in $pH_{50}$ (Fig. 7c, d). An interesting feature of the H80A mutant channel was its plateaued or biphasic activation between pH 5.0 and 4.5 (Fig. 7c, e). In mouse ASIC1a,

the protonatable residue E79 sits in a similar position as H80 in *Tad*NaC2 (Fig. 6b), at the distal end of the first beta strand that projects from transmembrane helix 1 in the wrist domain into the palm domain[51]. Notably, like the H80A mutation in *Tad*NaC2, mutation of E79 to a lysine, expected to disrupt a putative proton-binding site comprised of this residue and E416 on an adjacent beta strand[60], produced a biphasic proton sensitivity. The authors of this study found that a separate mutation, of residue D345 within the acid pocket (Fig. 6b), also produced biphasic proton sensitivity, and that mutation of E79 and D345 together produced severely diminished proton sensitivity that was not biphasic. This suggests that ASIC1a possesses at least two separate proton binding sites that contribute to channel activation, one within the palm domain and another within the finger-thumb region, and that these sites operate somewhat independently of each other[51].

Nonetheless, the H80 residue in *Tad*NaC2, which is conserved in *Hho*NaC2, is notably absent in all other channels that we included in our analyses (Fig. 6b), representing a unique putative determinant for proton activation in the placozoan channel. Based on our structural modeling, this residue appears proximal to the residues R201 and K203 in *Tad*NaC2, all clustered within the same structural region that harbors K211 in ASIC channels (Fig. 6). In this region, alanine mutation of R201 in *Tad*NaC2, which is absent in the *H. hongkongensis* orthologue *Hho*NaC2, enhanced proton sensitivity evident in an increased $pH_{50}$ value, as did mutation of the conserved residue K203E (Fig. 7d). Instead, deletion of K203 resulted in channels that did not traffic to the cell membrane and were thus likely non-functional (Fig. 8). This contrasts with ASIC channels, for which a K211E mutation, which is expected to disrupt molecular interactions with residues in the adjacent subunit, causes a moderate reduction in proton-sensitivity, while its deletion imposes a more marked effect but does not completely disrupt channel function[25]. In ASIC channels, a hub of hydrophobic residues that lies adjacent to K211 (i.e., residues F87, F174, F197, and L207), is thought to functionally couple conformational alterations between the palm and wrist regions during gating[25]. These hydrophobic residues are conserved in *Tad*NaC2, which has four hydrophobic residues in corresponding amino acid positions (i.e., F86, W165, F188, and L198; Supplementary Fig. 4). Moreover, in ASIC1 the K211 residue is thought to form transient inter-subunit interactions with acidic residues in the thumb region of an adjacent subunit during activation[61]. Indeed, future studies will be required to determine whether the non-homologous H80 residue in *Tad*NaC2, which contributes to proton activation and resides in a similar structural region as K211 (Fig. 6d), similarly operates through the hydrophobic hub and inter-subunit interactions.

In the finger region, we identify the residue H109, E104, and E105 as important for proton activation of *Tad*NaC2, with a single mutation of H109 causing a moderate reduction in $pH_{50}$, and the triple mutation of E104A/E105A/H109A having the most severe effect compared to all other tested mutants (Fig. 7b, d). These residues are fully conserved between *Tad*NaC2 and *Hho*NaC2, while H109 is also found in several other Deg/ENaC channels including several ASIC channels, and the E104/E105 doublet is found in *Ne*NaC2 (Fig. 6b). As noted, the finger region is of particular importance for ASIC2 channels where four amino acids differentiate proton-sensitive ASIC2a channel variants from non-functional ASIC2b variants[47]. Alanine substitution of the H109 equivalent in ASIC2a (i.e., H109A) produced different effects in separate studies, two describing no effect[42,47], and another a complete loss of proton-activation perhaps attributable to a reduction in membrane expression[43]. To the best of our knowledge, whether this equivalent residue in ASIC1a, or a histidine residue one position upstream in ASIC3 (Fig. 6b), contribute to proton activation has not been explored. However, as

noted above, mutation of residue D345 in mouse ASIC1a, which sits adjacent to the finger region where H109 resides in *Tad*NaC2 (Fig. 7f), has a strong impact on proton activation and similarly imposes a biphasic pH sensitivity[51]. Thus, an interesting prospect emerges that like ASIC1, *Tad*NaC2 possesses two separate loci for coordinating protons and channel activation, one in the finger-thumb region and the other in the palm domain, although utilizing different molecular determinants. An important caveat about the experiments and data presented in this study is that they do not permit direct inferences about the specific contributions of H80, H109, and other tested amino acids toward *Tad*NaC2 proton gating. Specifically, future functional and structural experiments will be required to determine whether these amino acids are acting as direct proton binding sites or play separate roles in channel gating (e.g., activation, desensitization, and/or recovery from desensitization). This will also require a deeper characterization of the transient and sustained components of the *Tad*NaC2 macroscopic current, and a determination of how (and if) different structural elements of the channel protein contribute to these currents.

Lastly, during our experiments we noticed that several *Tad*NaC2 mutations caused significant decreases in current amplitude, most marked for the K203Δ variant. We selected the H80A, H109A, and K203Δ variant channels, which produce moderate to absolute reduced current amplitude (Fig. 7e), to explore whether these specific mutations affect total protein expression and/or targeting of the channels to the cell membrane. Interestingly, and corroborating the electrophysiological recordings, all three mutations caused a reduction in total and surface/membrane protein expression, with the K203Δ variant being almost undetectable at the cell membrane (Fig. 8a–d). Thus, in addition to affecting proton sensitivity, all three of these mutations also affect total protein expression, perhaps reflecting decreased translation or stability of the single subunit protein. Instead, only the K203Δ variant had a noticeable effect on the ratio of membrane vs. total expression (Fig. d). This indicates that this particular mutation prevents trafficking to the cell membrane, perhaps via disrupted assembly of the homotrimeric channel, or aggregation of channel subunits.

*T. adhaerens* is simple seawater invertebrate that lacks body symmetry, has only six ultrastructurally distinguishable cell types, and lacks a nervous system and synapses[62]. Nonetheless, the animal is able to coordinate its various cell types to conduct directed locomotion including chemotaxis[63], gravitaxis[64], and thermotaxis[65]. Locomotion is achieved via the action of asynchronously beating cilia on its ventral epithelium, coupled with secretion of mucous thought to facilitate ciliary gliding[66]. Furthermore, various endogenous neuropeptides have been identified that when applied ectopically regulate *T. adhaerens* locomotive behavior[67,68]. Interspersed among the ventral ciliated cells are Lipophil cells which bear large acidic vesicles that are secreted over food algae during feeding[62]. Secretion of these substances causes rapid breakdown of algae (i.e., within seconds), after which the animal is conducts churning motions thought to promote nutrient uptake[69]. Interestingly, a single-cell transcriptome study of *T. adhaerens* revealed that *Tad*NaC2 is expressed in lipophil cells[70]. In agreement, we previously showed through fluorescence in situ hybridization that *Tad*NaC2 mRNA is restricted within a central region of the animal[24], consistent with what is referred to as the lipophil zone where these namesake cells reside[66]. Instead, mRNA of the proton-blocked channel *Tad*NaC6 was detected along the periphery of the flat, plate-like animal[24], in a region that contains neuroendocrine-like gland cells that also express voltage-gated calcium channels[71,72]. The expression of *Tad*NaC2 within the lipophil zone positions it to detect sudden drops in pH upon secretion of acidic vesicles at the ventral epithelium during

digestion. Since the aqueous volume between the epithelium and the underlying substrate is expected to be quite small, lipophil secretion and external digestion could conceivably lower the extracellular pH to a range that effectively activates *Tad*NaC2, allowing the channel to provide feedback on external digestion by depolarizing lipophil cells.

## Methods

**Transcriptomic resources**. To identify in which phyla to search for ASIC and Deg/ENaC channels, we performed an initial BLASTp analysis in the NCBI databases in the following taxa: Picobiliphytes (taxid:419944), Ancoracysta (taxid:2056028), Rhodophyta (taxid:2763), Chloroplastida (taxid:33090), Glaucophyta (taxid:38254), Palpitomonas (taxid:759891), Katablepharida (taxid:339960), Cryptophyta (taxid:3027), Centrohelida (taxid:193537), Haptophyta (taxid:2830), Telonemida (taxid:589438), Discoba (taxid:2611352), Metamonada (taxid:2611341), Malawimonadidae (taxid:136087), Collodictyonidae (taxid:190322), Mantamonadidae (taxid:1238961), Breviatea (taxid:1401294), Amoebozoa (taxid:554915), Apusomonadida (taxid:172820), Rhizaria (taxid:543769), Alveolata (taxid:33630), Stramenopiles (taxid:33634), Fungi (taxid:4751), and Holozoa (taxid:33208). The ASIC and Deg/ENaC receptors from human, *Platynereis dumerilii*, *Caenorhabditis elegans*, and *Aplysia californica* were used as queries (sequences and accession numbers provided in Supplementary Data 1).

Candidate sequences were identified in holozoans (multiple species), Alveolata (one species), and Stramenopiles (one species). Thus, whole transcriptomes of predicted proteomes from different classes of holozoans, including: metazoans, choanoflagellates, one filasterean (*Capsaspora owczarzaki*), and the filasterean-related *Tunicaraptor unikontum* were obtained. The more distantly related eukaryotic species, *Symbiodinium sp KB8* (Alveolata) and *Cafeteria roenbergensis* (Stramenopiles) were also added to the list. The transcriptomic databases were translated to protein using the tool TransDecoder (http://transdecoder.github.io/) with a minimum length of 75 amino acids. The databases that were available as predicted proteins were used directly. For completeness assessment of the transcriptomes, we ran BUSCO v5.2.1[73] in protein mode and with the lineage set to 'eukaryote' with the database 'eukaryota_odb10' (database creation: April 2022; number of BUSCOs: 255). The sources of databases used for this analysis and the results of the BUSCO completeness analysis are described in Supplementary Table 1.

**Clustering and phylogenetic analyses of Deg/ENaC channel protein sequences**. Multiple-species sequences of the families Deg/ENaC (PF00858) channels were obtained from the PFAM database (https://pfam.xfam.org). These sequences were aligned using MUSCLE, and automatically trimmed with trimAl[74] using the gappy-out mode. The trimmed sequences were used to produce Hidden Markov Models using HMMR3[75]. The subsequent search for receptors in the database obtained as described above was performed using HMMER3 with an expected value cut-off of 1E−10[76]. The identified sequences are provided in Supplementary Data 2.

The relationship between candidate Deg/ENaC sequences was first analyzed using a cluster-based strategy with the CLuster ANalysis of Sequences CLANS algorithm[27]. The sequences were clustered using $P$ value cut-offs of 1E−10, 1E−20, 1E−30, 1E−40, and 1E−50 (Supplementary Data 3 to 7, respectively). We selected a $P$ value cut-off of 1E−30 to pre-filter Deg/ENaC channel sequences prior to phylogenetic analysis, permitting the extraction of 874 sequences with at least one connection to the main cluster. These selected sequences were then analyzed using

DeepTMHMM[77] to predict the transmembrane helices (TMHs), and those with no predicted TMH were discarded in order to remove sequences that were too fragmented to contribute to the phylogenetic analysis. Several additional sequences identified as duplicates via their sequence identifiers were also manually removed, after confirmation via pairwise sequence alignment. Selected and filtered protein sequences (798 total; provided in Supplementary Data 8) were aligned with MAFFT version 7 with the iterative refinement method L-INS-I[78], and alignments were trimmed with trimAl in gappyout mode[74] producing an alignment with 455 informative positions. The maximum-likelihood tree presented in Fig. 2 was generated using IQ-TREE 2[79] with the best-fit model WAG + F + G4. To calculate branch support, we ran 100 standard bootstrap replicates[80]. Select clades on the trees were collapsed with Figtree v1.4.4[81], and then colored and the trees annotated using Adobe Illustrator. The nexus file for the tree is available in Supplementary Data 9, while the aligned sequences and aligned-trimmed sequences are available in Supplementary Data 10 and 11, respectively.

**Cloning of Deg/ENaC channel cDNAs for in vitro expression**. The *Tad*NaC2 and *Tad*NaC10 cDNAs were cloned from a cDNA library prepared from whole-animal total RNA, as reported previously[24,82]. Briefly, corresponding cDNAs were amplified by nested PCR using two sets of primers for each channel, with the second forward primer incorporating an *Xho*I restriction site followed by a Kozak translation initiation sequence flanking the channel start codon, and the second reverse primer incorporating a *Bam*HI restriction site followed by a stop codon (Supplementary Table 2). PCR reactions were done with the Phusion high fidelity DNA polymerase (New England Biolabs Canada, Whitby Ontario) using standard protocols. The included restriction sites permitted cloning into both the pIRES2-EGFP and pEGFP-C1 mammalian expression vectors (Clontech), the former a bicistronic vector expressing enhanced green fluorescence protein (EGFP) from the same transcript bearing the *Tad*NaC channel coding sequence (through an internal ribosome entry site), and the latter expressing the channel with EGFP fused to its N-terminus. These plasmid vectors were, respectively, named p*Tad*NaC2-IR-EGFP and pEGFP-*Tad*NaC2. The various mutations of *Tad*NaC2 were generated via site-directed mutagenesis using the primers listed in Supplementary Table 2, using a modified PCR protocol of only 15 cycles, after which the PCR reactions were each treated with 1 μL of the restriction enzyme *Dpn*I, then transformed into the *E. coli* NEB Stable strain from New England Biolabs (NEB, MA, USA) for plasmid isolation. All generated plasmid vectors were verified through diagnostic restriction digestion followed by Sanger DNA sequencing. The mouse ASIC1a subunit was synthesized by GenScript (New Jersey, USA) with *Sac*I and *Bam*HI restriction sites flanking the start and stop codons, and a 5' Kozak sequence flanking the start codon, which was cloned into the bicistronic vector pIRES2-EGFP to provide the construct pmASIC1-IR-EGFP.

**In vitro expression and electrophysiology**. Chinese Hamster Ovary cells (CHO-K1; Sigma) were cultured and transfected in vented T-25 flasks at 37 °C in a 5% $CO_2$ incubator. Cells were grown in a DMEM/F12 and DMEM/Ham's F-12 50/50 MIX media supplemented with 10% fetal bovine serum (both from Wisent Inc., Saint-Jean-Baptiste, Quebec). For in vitro expression, 3 μg of the p*Tad*NaC2-IR-EGFP vector, or 2 μg of the pmASIC-IR-EGFP vector, were transfected into CHO-K1 cells grown to 80–90% cell confluency using the transfection reagent PolyJet™ (FroggaBio, Concord Ontario), according to the manufacturer protocol. Cells were incubated with the transfection mix for 6 h,

and then washed three times with serum-free media, followed by an overnight incubation at 37 °C. For electrophysiological recordings, cells were briefly treated with a trypsin solution (Sigma-Aldrich), plated onto glass coverslips in 35-mm tissue culture-treated dishes (Eppendorf, Mississauga Ontario), and then left to adhere for a minimum of 6 h before recording.

Before performing the patch-clamp electrophysiology experiments, cells on glass coverslips were transferred into 35-mm cell culture dishes (Eppendorf) containing ~3 mL of extracellular recording solution. For all pH-dose–response curve experiments, the external solution contained 140 mM NaCl, 4 mM KCl, 2 mM CaCl$_2$, 1 mM MgCl$_2$, 5 mM HEPES, and 5 mM MES (pH 3.5 to 7.5 with HCl/NaOH; osmolarity set to 320 mOsM with glucose), while the internal solution contained 120 mM KCl, 2 mM MgCl$_2$, 10 mM EGTA, 10 mM HEPES (pH 7.2 with KOH; osmolarity set to 300 mOsM with glucose). For the Na$^+$/K$^+$ ion selectivity experiments, the external solution contained 150 mM XCl (where X = Na$^+$ or K$^+$), 10 mM TEA-Cl, and 10 mM of MES buffer (pH4.5 with TEA-OH). The internal solution contained 150 mM NaCl, 10 mM EGTA, 10 mM TEA-Cl, and 10 m M HEPES (pH 7.2 with TEA-OH). For the Ca$^{2+}$ dose–response curve solutions, the external consisted of 140 mM NaCl, CaCl$_2$ (0 to 10 mM), 4 mM KCl, 1 mM MgCl$_2$, 5 mM HEPES, and 5 mM MES (pH 4.5 with HCl) with equimolar NMDG$^+$ used to substitute Ca$^{2+}$ ions in the 0 to 1 mM Ca$^{2+}$ solutions. For the amiloride pharmacology experiments, amiloride hydrochloride hydrate (MilliporeSigma, Ontario Canada) was dissolved at the indicated concentrations in extracellular salines having the same formulation as the one used for the dose–response experiments, while the internal solution remained the same. The single-dose pharmacology experiments were done by first perfusing pH 4.5 or pH 5.5 solutions for TadNaC2 and mASIC1a, respectively, followed by the same respective solutions but containing 3 mM amiloride. For the amiloride dose–response experiments, amiloride was dissolved to between 0 and 3 mM in a pH 4.5 extracellular saline. Whole-cell patch clamp recordings were done on an inverted fluorescent microscope, with recording electrodes coupled to an Axopatch 200B amplifier interfacing with a Digidata 1550A data acquisition digitizer controlled on a personal computer using the pClamp 10 software (Molecular Devices, California USA). Patch pipettes were made from thick-walled borosilicate glass (1.5 × 0.86 mm outer and inner diameter, respectively) pulled using a P-1000 micropipette puller (Sutter, Novatto California) with a resistance in the bath solution of 2 to 6 megaohms. All recordings were sampled at 2000 Hz and filtered at 500 Hz using the pClamp10 software (Molecular Devices). Perfusion of the various solutions was achieved using a Valvelink8.2 gravity flow perfusion system (AutoMate Scientific, Berkeley CA). Dose–response curves were fitted using monophasic exponential function using the program Origin 2017 (OriginLab), which produced pH$_{50}$ (EC$_{50}$) and Hill coefficients ($n_H$) according to Eq. 1 below. The relative permeability of TadNaC2 to Na$^+$ and K$^+$ ions (i.e., pNa$^+$/pK$^+$) was calculated using Eq. 2 below, where $E_{Rev,K}$ and $E_{Rev,Na}$ denote the reversal potentials observed with K$^+$ and Na$^+$ present in the external solution, respectively.

$$\frac{I}{I_{Max}} = \frac{1}{1 + \left(\frac{EC_{50}}{[H^+]}\right)^{n_H}} \quad (1)$$

$$\triangle E_{Rev} = E_{Rev,K} - E_{Rev,Na} = \frac{RT}{zF} \times \ln \frac{K^+}{Na^+} \quad (2)$$

**Statistics and reproducibility**. All statistical analyses were conducted using the program Origin version 2017. Replicate numbers and output parameters for the various statistical analyses, including p values, are provided throughout the text.

**Structural analyses**. For prediction of the TadNaC2 homo-trimeric structure, a monomeric structure was first generated using Phyre2[83] run in intensive mode. The ternary complex was assembled using the three-fold rotational symmetry of the Gallus gallus ASIC1a channel structure in its resting state (PDB number 6AVE)[49]. Energy minimization, to minimize steric clashes, and geometry optimization were performed using Chiron[84] and Phenix[85], respectively. Prediction of the TadNaC2 monomeric structure was achieved with the AlphaFold algorithm[50] available on the Google Colab webserver. All structural annotations were prepared using the visualization software UCSF ChimeraX version 1.3[86].

**Imaging and quantification of EGFP fluorescence**. CHO-K1 cells were transfected and imaged in 6 mL vented T-25 flasks on an Axio Observer 3 inverted microscope (Zeiss Canada, Toronto Ontario) bearing an LED light source and appropriate fluorescence filters. Micrographs were acquired through a 10x objective using a mounted Axiocam 506 camera with the ZEN lite software (Zeiss), selecting fully confluent areas of each dish. Fluorescence quantification was done via integrated density analysis of the acquired micrographs using ImageJ[87].

**Quantification of total and surface expressed TadNaC2 channel protein in vitro**. The Pierce$^{TM}$ Cell Surface Protein Biotinylation and Isolation Kit (Thermo Fisher Scientific, Massachusetts USA) was used to quantify total and cell surface expression levels of N-terminal GFP-tagged TadNaC2 protein expressed in CHO-K1 cells from the pEGFP-TadNaC2 vector, as well as GPG-tagged TadNaC2 variants bearing H80A, H109A, and K203Δ mutations. Briefly, for each TadNaC2 variant, two T-75 flasks bearing CHO-K1 cells at ~80% confluency were each transfected with 9 μg of DNA using the PolyJet$^{TM}$ reagent. The flasks were washed with 10 mL of phosphate-buffered saline (PBS; MilliporeSigma, Ontario Canada), then 5 mL of a PBS solution bearing 0.25 μg/ml EZ-Link$^{TM}$ Sulfo-NHS-SS-Biotin was applied over the cells and these were incubated on ice for 30 min. This solution was removed by aspiration, then cells were washed twice with 10 mL of ice-cold Tris Buffered Saline (TBS; 150 mM NaCl, 50 mM Tris-Cl pH 7.5) and suspended in 5 mL TBS using a cell scraper. For each TadNaC2 variant, corresponding sets of suspended cells were combined and lysed with 500 μL of the provided lysis buffer supplemented with Protease Inhibitor Cocktail I (MilliporeSigma). The protein lysates were clarified by centrifugation at 10,000 × g for 2 min, transferred to a new tube, then quantified using the Pierce™ BCA Protein Assay Kit (Thermo Fisher). The volumes of all samples were adjusted with lysis buffer to dilute them to the same concentration (5 μg/μL), then 50 μL of each lysate was set aside, representing the total TadNaC2 protein component. Membrane expressed TadNaC2 proteins were isolated by applying 600 μL of the clarified protein lysates to columns loaded with the 250 μL of the provided NeutrAvidin$^{TM}$ Agarose slurry, followed by a wash using the provided buffer, and elution with 200 μL of the provided SDS-PAGE sample elution buffer. For each experiment, 40 μg (8 μL) of the total lysate and 40 μL of the isolated surface proteins, which corresponds to a 5-fold enrichment of surface relative to total protein fractions, were loaded onto a 4–12% Bis-Tris gel for western blotting.

For western blotting, protein samples were first electrophoretically separated using a 4–12% Bis-Tris acrylamide gel (Invitrogen, Massachussets USA), and transferred onto nitrocellulose membrane using a wet transfer system at 25 mV overnight.

Membranes were then washed with TBS containing 0.05% v/v Tween-20 (TBS-T), then incubated for 1 h in TBS-T with 5% w/v skim milk powder at room temperature. The membrane was then washed with TBS-T and incubated at 4 °C overnight in TBS-T plus 5% milk and rabbit monoclonal anti-EGFP antibody (1:1000 dilution) and rabbit monoclonal anti-GAPDH antibody (1:50,000 dilution; both from Cell Signalling Technology, Massachusetts, USA). The membranes were then incubated for 2 h at room temperature in TBS-T containing 5% milk and a 1:3000 dilution of goat anti-rabbit secondary antibodies conjugated to horseradish peroxidase (Cell Signalling). Imaging of membranes was acheived by applying 5 mL of Clarity Western ECL Substrate (Bio-Rad, California USA) then imaging them using an IBright™ FL1500 Imager System (Thermo Fisher). Protein band intensity was quantified as mean gray area using with the software ImageJ.

**Reporting summary.** Further information on research design is available in the Nature Portfolio Reporting Summary linked to this article.

## Data availability

All presented data are available in the text, Supplementary Figures, and Supplementary Files. Uncropped original images of Western blots presented in Fig. 8c are provided in Supplementary Fig. 5. Source data for all plots are provided in Supplementary Data 12.

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

## Acknowledgements

This research was funded by an NSERC Discovery Grant (RGPIN-2021-03557), an NSERC Discovery Accelerator Supplement (RGPAS-2021-00002), an Ontario Early Researcher Award (ER17-13-247), and a Canadian Foundation for Innovation Grant (35297) to A. Senatore, An Ontario Graduate Scholarship to W.E., a Biotechnology and Biological Sciences Research Council Grant (BBSRC; BB/W010305/1) to L.A.Y.-G., and an NSERC Discovery Grant (RGPIN-2023-05615) and a Canadian Foundation for Innovation Grant (40684) to M.A.C. We would like to thank Dr. Carolyn Smith for early discussions about putative pH sensing behavior by *T. adhaerens*.

## Author contributions

Conceived the study: A. Senatore and W.E; design of experiments: A. Senatore and W.E.; electrophysiology and pharmacology: W.E.; molecular biology: A. Senatore, W.E., A. Singh, and J.G.; structural analyses: W.E., A. Senatore, and M.A.C.; bioinformatics analyses: L.A.Y.-G., A. Senatore, and W.E.; writing of the manuscript: A. Senatore, W.E., L.A.Y.-G., T.D.M., M.A.C., J.G, A. Singh, and M.P.; revision of manuscript: A. Senatore, W.E., L.A.Y.-G., T.D.M., M.A.C., J.G., A. Singh, and M.P.

## Competing interests

The authors declare no competing interests.

## Ethics

All authors of this study fulfilled the criteria for authorship, with roles and responsibilities agreed upon before starting the research. The research was not restricted or prohibited in any way, and does not result in stigmatization, incrimination, discrimination or otherwise personal risk to participants.
