## [Peer Review File · Communications Biology]

Reviewers' comments:

Reviewer #1 (Remarks to the Author):

In this paper, Elkhatab et al. performed a phylogenetic analysis of DEG/ENaC ion channels with a focus on members from *Trichoplax*. Moreover, they functionally characterized one of these channels, TadNaC2, and showed that it was activated by protons with low sensitivity. In addition, they performed extensive mutagenesis to show that residues important for acid activation in the subgroup of bona fide acid-sensing ion channels (ASICs) are not conserved in TadNaC2, suggesting the independent evolution of proton sensitivity in different DEG/ENaC subgroups. Finally, they go on to show that acidic medium induces an avoidance behavior and induces contractions of *Trichoplax*. At high doses (1 mM), amiloride, a diuretic that inhibits most DEG/ENaCs, increased the displacement speed of *Trichoplax* and reduced contractions induced by solution exchange.

This is an interesting manuscript with lots of data that increases our understanding of the evolution of DEG/ENaCs, particularly the evolution of proton sensitivity in this ion channel family. While a recent paper in the same journal already reported an extensive phylogenetic analysis of DEG/ENaCs across metazoan species (Aguilar-Camacho et al, 2023), reducing the novelty of this part, the analysis reported in the present manuscript provides additional information. Moreover, mutational analysis of TadNaC2 provides important novel insights into the molecular mechanisms of proton sensitivity in this ion channel family. The last part (effects of acidic medium on *Trichoplax* behavior) is preliminary, however, and only loosely connected to the rest of the manuscript. In addition, not all the conclusions are supported by the data.

Major comments:

1) There is no convincing evidence for the involvement of TadNaC2 or any other TadNaC in the acid-evoked behaviors of *Trichoplax*. The link between these behaviors and TadNaCs rests entirely on the effects of high doses of amiloride, which were mainly unexpected (for example, increased instead of decreased displacement speed of *Trichoplax*). Moreover, at 1 mM, amiloride is not specific to DEG/ENaCs and could influence other membrane proteins, such as NHEs. Because the manuscript is quite complex and long, I recommend deleting the whole section on the effects of acidic medium on *Trichoplax* and report these findings in an independent paper.

2) I understand that the authors used PPK channels as an outgroup because they formed a separate cluster in the CLANs analysis. But from an evolutionary point of view this does not make a lot of sense. Therefore, the authors should provide an alternative tree with DEG/ENaCs from filasterean species as an outgroup to root their tree. I also recommend discussing the phylogenetic tree more cautiously, as new trees with even broader sampling might change the relative positions of some branches. The authors mention that PPKs and ACD/FLR fall outside of ASIC and ENaC superclades (lines 552-554). How is this possible? Do the authors claim that these DEG/ENaCs evolved independently?

Moreover, the exact filasterean species containing DEG/ENaCs should be identified (I could not find this information).

Finally, the support for the maximum likelihood trees should be calculated as bootstrap values, which is better than the SH-like and aBAYES (which is not "real" Bayesian inference) methods used by the authors.

3) The functional characterization of TadNaC2 is not entirely convincing and is sometimes confusing. I noticed that at pH 4.5, tachyphylaxis was more pronounced for the transient than for the sustained current (Fig. 4C). Was the time interval between activations sufficient to allow for complete recovery (of the transient current)? Please also analyse tachyphylaxis of the sustained current at pH 4.5. The authors mention potentiation of TadNaC2 by 3 μ M amiloride. It would be helpful to have a complete concentration-response curve to see what happens at higher concentrations of amiloride. Why was there no transient current with Na⁺ as the sole extracellular cation (Fig. 5b)? And why was the transient current in this panel discovered only with K⁺? In Fig. 5c, the authors compared early

reversal potentials with Na⁺ and with K⁺. But these reversal potentials were measured at different time points and cannot be compared. Early and late Na⁺ report the reversal potential of the same current component. Thus, ion selectivity of the early current cannot be calculated. This part is confusing and sheds some doubt on the electrophysiological characterization, in particular with regard to ion selectivity.

Was the transient current with Ca²⁺ as the sole extracellular cation amiloride-sensitive? I do not see a sustained outward current under this condition. Could Ca²⁺ slowly block this channel? In the Discussion section (line 642), it is mentioned that the early TadNaC2 current is highly Ca²⁺ permeable. But because Ca²⁺ permeability has only been qualitatively assessed, this statement is not supported by the data.

4) Figure 7: It would be helpful to show current traces at different pH values for selected mutants.

5) The authors should mention that there are also bona fide ASICs that are activated by protons and lack the critical histidine corresponding to H73.

6) The question arises as to whether the relatively low proton sensitivity of TadNaC2 is relevant to Trichoplax. A short discussion of the natural environment of this organism and the pH fluctuations that it might experience would be helpful.

7) Lines 582-591: I found the section on the relationship of DEG/ENaCs to P2X receptors and PAC channels highly speculative with no experimental support. I recommend deleting this part of the manuscript.

Minor comments

1) Approximately half of the introduction is a summary of the findings. This part could be substantially shortened to briefly summarize only the most important findings.

2) The abstract claims that TadNaC2 has strikingly similar biophysical features to ASICs, but the Results show that this channel has rather some unique features.

3) Mention the pH used on Figure 5B.

4) Figures 3-5: I suggest labeling lines in the graphs directly and to delete the legend from the figures (including "AVG ± SD", an information that should be provided in the real legend below the figures). This makes it easier for the reader.

5) Line 410: Should it read "K203" instead of "K205"?

6) Line 432: the authors mention that they wanted to determine whether channel membrane expression was reduced. But in this paragraph, they only assessed total protein expression.

7) Lines 630-631: because ASIC3 develops a sustained current (different from its window current) only at very low pH, it is an outdated concept that this sustained current contributes to nociceptive signaling.

8) Lines 636-640: personally, I found this Discussion to be highly speculative. This is just a note for consideration.

9) Lines 679-681: the authors mention results from P71H and T72H mutants, which are not shown. Either the results should be shown or this sentence deleted.

10) Lines 687-688: "which is notably absent in HhoNaC2". Because there is no functional data for

HhoNaC2, it is not clear why the absence is notable.

11) Line 696: Please provide a reference for the hydrophobic hub.

12) Lines 699-702: it was not completely clear to me why H80 in TadNaC fulfills functions similar to K211 in ASICs.

13) Lines 727-729: The K203Delta mutant may lead to a loss of function by many mechanisms. That it imposes instability on homotrimeric assembly is a speculation.

14) Line 730: "ability to record current for these variant channels" – there was no current with K203Delta.

Reviewer #2 (Remarks to the Author):

In "Independent evolution of an ASIC-like Deg/ENaC sodium channel in the Placozoa" Elkhatib et al., present clustering analysis and a detailed phylogenetic analysis of the ASIC/ENaC channel superfamily, new functional and structural analysis of a proton-activated placozoan ortholog and analysis of behavioral responses to pH changes in placozoans. The authors state their phylogeny resolves true evolutionary relationships between important ASIC/ENaC clades that have been missed in similar recent analysis because of refinements in their phylogenetic methodology. The authors posit that proton-activation is independently evolved in their placozoan ortholog TadNaC2 and vertebrate ASIC channels based on differences in the underlying mechanism and separate branching in the phylogeny and point to several other independent evolutions of proton gating in the channel family that others have observed. The results from behavioral analysis show several acid-dependent behaviors, only one of which appears sensitive to the non-specific ASIC/ENaC blocker amiloride. The manuscript concludes that they have resolved the true evolutionary relationships of ASIC/ENaC channels and point out potential flaws in previous analyses, but they do not discuss major caveats with their own analysis and do not provide all the information required to assess its validity. Results on the proton-gating mechanism of TadNaC2 are similarly overinterpreted, though the conclusion the mechanism differs from ASIC channels appears solid. It also appears premature for the authors to include behavioral analysis on proton-dependent behaviors in the same manuscript as the TadNaC2 work. Since they do not have the ability to provide genetic evidence or even solid pharmacological evidence of a link between these behaviors and TadNaC channels, the behavioral analysis should be presented elsewhere as a separate study. There are numerous examples of proton dependent behaviors that do not involve these channels and including it here biases the field to see a link that might not exist. Specific comments on issues in the manuscript that need to be addressed are listed below.

Major issues:

1. While the cluster analysis is a useful tool for visualization of the relatedness of sequences in this channel family, its use in selecting sequences for phylogenetic analysis as explained is dubious. The key issue seems to be how the cut-off score for displaying linkage was decided. The chosen value makes a nice display in Figure 1, but is there a scientific reason the value was chosen or was it arbitrary? This matters because the cluster analysis was used to select sequences for phylogenetic analysis. Would those results be the same if a higher or lower cut off value was used? Selective elimination of data is sometimes warranted, but it needs to be carefully justified.
2. The phylogenetic analysis is lacking some key details that are needed to assay how strong the results truly are. Most importantly there are no details on where the phylogeny was trimmed and how many informative positions it contained. File S4 is supposed to have the trimmed alignment but it looks like a full untrimmed alignment and it is impossible to determine the number of positions from it. Because the majority of the alignment is gap, its hard to visualize how the sequences align to each other and whether the alignment is reasonable. In this gene family the distances between key nodes are very small in comparison to amount of sequence divergence that has occurred afterwards. If the

number of phylogenetically informative positions is small (just a couple hundred amino acids for example), then only a very few positions are determining the branching orders the authors state this study resolves, and makes that statement an overreach. The results presented here would be easier to consider as a "resolution" or a significant advance over other published studies if the authors showed results were robust to sequence selection (cut off for inclusion) and phylogenetic method (is it sensitive to algorithm or selection model, for example). Alternatively if they feel they can present a detailed justification as to why their sequence selection cut-offs, alignment trimming and phylogeny methods are the best choices to answer the question in hand, they could approach it in that way. The authors point out valid potential caveats with recent, but they don't point out these caveats of their analysis. It is really difficult to build robust phylogenies from short alignments of large poorly-conserved gene families, and the truth is the detailed evolutionary history may not be fully resolvable. It is relatively easy to produce statistically robust trees by eliminating the right sequences, but does this really get to the true answer?

3. Despite the caveats above, it is interesting that this is the 2nd recent study to find the two distinct ancient clades of these channels in animals, and most overlapping sequences go to the same clade in both studies. These parts of the results seem most robust, and it may be that the differences between the two studies are less robust. The discussion on these points should be more balanced. It seems that both studies have some potential value to offer in supporting each other. The authors should also not speculate that long branch attraction as the reason for the differences between the studies unless they can point to evidence that this has occurred and is responsible for the differences. They should also consider if any of the branching patterns in their tree could also be influenced by long branch attraction.

4. The choice of divergent *Drosophila* and *C. elegans* channels as an outgroup is inappropriate, and the tree should be run again either with a true outgroup or it could simply be presented as an unrooted phylogeny. But the selected outgroup channels are neither an outgroup by species or by evolutionary origin. Any phylogeny of the gene family rooted to them by definition does not reflect the true evolutionary relationships within the gene family. They must logically branch from somewhere within the ASIC or ENaC groups, the tree is not able to resolve where because of their high sequence divergence. But presenting them as an evolutionary outgroup is not at all accurate to the evolutionary history of the gene family. It does illustrate a major difficulty with phylogenetic analysis of large gene families – new duplicates are often under lower or divergent selective pressures and thus change faster and sometimes appear basal in phylogenies even though we know they are derived.

5. The paper conclusions are based on the premise that the ancestor of the ASIC group was not proton activated. The alternative hypothesis, that the ASIC group ancestor was proton activated and that proton sensitivity has been lost in vertebrate BASICs is not considered, but is actually better supported by the phylogenetic distribution of proton-activated vs. proton-insensitive channels. The best evidence to support proton-insensitivity of the ancestor is that the mechanisms of proton activation in vertebrate ASICs and TadNaC2 are different. But this may not suggest the absence of low affinity proton activation in the ancestor (as seen in TadNaCs), but instead addition of high proton affinity in ASICs. There simply isn't enough functional data to distinguish between these possibilities, and the merits of both should be considered.

6. The data used to show alterations in proton sensitivity for various mutants is a not great, in a large part because the small sequence sizes give measurements with large errors. But more importantly, there is no control for maximal channel opening. Without this, it is impossible to tell if these mutations selectively reduce proton activation or simply make it harder to open the channels that do get to the plasma membrane. This leaves the only good evidence as changes in affinity, and the data quality is not high enough in most cases to give a lot of confidence. It is at least suggestive but not definitive, and leaves some risk it won't pan out. The data that the mechanism differs from ASIC is strongest.

7. The animal behavioral data is interesting but does not belong here because it doesn't provide significant evidence for the involvement of TadNaCs and there are many other mechanisms that produce proton sensitive behaviors. Amiloride is not specific enough to conclusively establish involvement of TadNacs, and most of the behaviors are insensitive to it as well. Presenting this data here unfairly establishes an expected link to TadNaCs over other possible mechanisms that isn't warranted at this point. However, the data is interesting in its own right and could probably stand on

its own in a 2nd separate publication.

Very minor point:

There are a number of places where the authors use superlatives like "Remarkably" for phenomena that aren't unusual at all. For example, it isn't remarkable that you can get big currents from non-codon optimized channels, when many of the historically best expressing proteins in vertebrate expression systems are invertebrate orthologs that haven't been codon optimized. It's also not remarkable that *Trichoplax* can respond to pH changes without a nervous system when virtually every living organism across the tree of life responds to pH changes. On the other hand, it is remarkable how similar P2X and ASIC folds are despite the huge sequence divergence. More reserved use of superlatives would focus attention to the places they are really needed.

Reviewer #3 (Remarks to the Author):

The authors report the discovery of a proton-activated Deg/ENaC channel in the non-bilaterian early branching phylum Placozoa, called TadNaC2. Using sequence comparison, structural modeling, and mutation analysis, the authors show that TadNaC2 proton-activation differs substantially from the phylogenetically related Acid Sensing Ion Channels (ASICs), exclusively present in bilaterian animals. Furthermore, the results of this study suggest that proton activation developed many times independently within the Deg/ENaC superfamily. The findings in this study will be not only relevant in the field of ion channels, but in the field of protein evolution and the emergence of protein function.

Overall, this study is of good quality, including the methods, results, and data interpretation. I have only one concern. One of the main points of the manuscript is that the newly discovered TadNaC2 channel and ASICs channels proton-activation are fundamentally distinct from one another, necessitating certain residues in the palm and finger regions (i. e. H80 and H109). However, the authors show that the protein level for those mutants (i.e. H80A and H109A) in transfected cells is lower than WT protein levels. Authors should consider transfecting cells with lower WT vector concentration (with current amplitudes similar to mutants) to assess that the shift in the pH50 is not due to low protein level. Furthermore, authors complement their idea of different residues are needed for proton activation in the different families (i. e. H80 in TadNaC2 and H73 in ASICs), by making TadNaC2 look like an ASIC (i.e., mutating P71 and T72 to H in a TadNaC2 with H80A background). Because the authors could not record currents from those mutants, they claim that those positions are not equivalent between channels. However, the authors do not show if the construct is expressing, so the lack of function could be due to a lack of protein and not a lack of rescue.

In addition to the point mentioned above, the following minor points should be considered in further versions of the manuscript:

- Authors use the term "chordate" as a synonym of "vertebrate" or "craniate" which is not completely true. "Chordate" refers to a group of animals which include "Cephalochordata", "Urochordata" and "Craniata/Vertebrata". In lines 166, 169, 189, and 557 authors should change "chordate" to "vertebrate". In Figure 2 authors should change "Chordate/Cephalochordate ASICs (Group A)" by "Vertebrate//Cephalochordate ASICs (Group A)", "Chordate/Ambulacraria/Lophotrochozoan ASICs (Group B)" by "Cephalochordate/Ambulacraria/Lophotrochozoan ASICs (Group B)", "Chordate/Urochordate BASIC channels" by "Vertebrate/Urochordate BASIC channels", and "Chordate ENaC channels" by "Vertebrate ENaC channels".

- Authors define two superclades among the DEG/ENaC family (i.e., ASIC and ENaC superclades), however, IMHO assigning a superclade name based on the function of a small portion of the channels of that big clade could bring some misinterpretations. I encourage the authors to use a more neutral nomenclature (e. g. Aguilar-Camacho et al., 2023).

- Figure 2. Branch names like "Ambulacrarian FaNaC/WaNaC- related channels" can bring misunderstanding. Assigning a function-related name to an untested channel knowing how variable in

functions are DEG/ENaC genes is risky. I recommend reconsidering.

- Supplementary Table 1. Maybe authors could consider adding a third taxonomic column (e.g. Clade/Phylum/Subphylum) to make the categories more adequate (e.g. Cephalochordata is not a Phylum is a Subphylum).
- Line 307. Full species names (binomial nomenclature) should be employed the first time that an animal is mentioned. (i.e., *Branchiostoma belcheri* and *Lingula anatina*).
- Line 415. Which is the mutant that the authors refer to H80A or H109A?
- Line 424. As the authors have shown is not that the mutation disrupted the function, it seems that the channels are not expressed in the membrane.
- Line 427. K2034E. Can you fix it?
- Line 439. Can the authors show the data?
- Line 896. Can the authors confirm the concentration was "micro-" and not "mili-" molar?
- Figure 6b. Mammalian numbering of the residues. There are two D345, I guess the second is D407. Can the authors fix it?

We would like to thank the reviewers for their thorough review and thoughtful comments. As they suggested, we removed the behavioral experiments and conducted all requested additional experiments and analyses. Altogether, we feel this version of the manuscript is greatly improved, and would like to once again thank the reviewers for their excellent feedback.

Below, you will find our responses to reviewer comments in red, along with any substantially modified figures. As requested, we quoted sections of the manuscript text in order to provide quick access to revisions we made.

Reviewers' comments:

Reviewer #1 (Remarks to the Author):

In this paper, Elkhatab et al. performed a phylogenetic analysis of DEG/ENaC ion channels with a focus on members from Trichoplax. Moreover, they functionally characterized one of these channels, TadNaC2, and showed that it was activated by protons with low sensitivity. In addition, they performed extensive mutagenesis to show that residues important for acid activation in the subgroup of bona fide acid-sensing ion channels (ASICs) are not conserved in TadNaC2, suggesting the independent evolution of proton sensitivity in different DEG/ENaC subgroups. Finally, they go on to show that acidic medium induces an avoidance behavior and induces contractions of Trichoplax. At high doses (1 mM), amiloride, a diuretic that inhibits most DEG/ENaCs, increased the displacement speed of Trichoplax and reduced contractions induced by solution exchange.

This is an interesting manuscript with lots of data that increases our understanding of the evolution of DEG/ENaCs, particularly the evolution of proton sensitivity in this ion channel family. While a recent paper in the same journal already reported an extensive phylogenetic analysis of DEG/ENaCs across metazoan species (Aguilar-Camacho et al, 2023), reducing the novelty of this part, the analysis reported in the present manuscript provides additional information. Moreover, mutational analysis of TadNaC2 provides important novel insights into the molecular mechanisms of proton sensitivity in this ion channel family. The last part (effects of acidic medium on Trichoplax behavior) is preliminary, however, and only loosely connected to the rest of the manuscript. In addition, not all the conclusions are supported by the data.

Major comments:

1) There is no convincing evidence for the involvement of TadNaC2 or any other TadNaC in the acid-evoked behaviors of Trichoplax. The link between these behaviors and TadNaCs rests entirely on the effects of high doses of amiloride, which were mainly unexpected (for example, increased instead of decreased displacement speed of Trichoplax). Moreover, at 1 mM, amiloride is not specific to DEG/ENaCs and could influence other membrane proteins, such as NHEs. Because the manuscript is quite complex and long, I recommend deleting the whole section on the effects of acidic medium on Trichoplax and report these findings in an independent paper.

We agree with this suggestion and have removed this data from the manuscript.

2) I understand that the authors used PPK channels as an outgroup because they formed a separate cluster in the CLANS analysis. But from an evolutionary point of view this does not make a lot of sense. Therefore, the authors should provide an alternative tree with DEG/ENaCs from filasterean species as an outgroup to root their tree.

We agree that choosing the PPK as an outgroup without properly discussing the reasons for this choice is biased. Thus, we have prepared a new phylogenetic tree using the filasterean Deg/ENaC channels as the outgroup (new Figure 2). While the new tree has a different outgroup, the overall branching patterns our new tree are mostly consistent with our previous one, retaining two major clades also consistent with the recent study by Aguilar-Camacho et al, 2023. For congruency with the nomenclature put forward by Aguilar-Camacho et al, we changed the name of our two major clades previously described as "ASIC and ENaC superclades" to Clades A and B, respectively.

Figure 2. Maximum likelihood protein phylogeny delineates two major clades of metazoan Deg/ENaC ion channels. The tree was generated with the program IQ-TREE 2 with the best-fit model LG+F+G4, and rooted with the filastrian-related Deg/ENaC channel homologues. Node support values are for 1,000 bootstrap replicates (green). The asterisks and labels (pink) indicate single channels or clades bearing Deg/ENaC channels that have been characterized as proton-activated.

I also recommend discussing the phylogenetic tree more cautiously, as new trees with even broader sampling might change the relative positions of some branches.

We agree with this suggestion and have tamed our interpretation of the phylogenetic data throughout the manuscript, focusing on the similarities between our study and that by Aguilar-Camacho *et al*, and on the novel findings that our analysis brings forward. We also changed the naming of the different clades in our phylogenetic trees to avoid suggestions that channels in common clades are related (e.g., “Ctenophore HyNaC/NeNaC-related channels” renamed as “Ctenophore” channels, etc.).

The authors mention that PPKs and ACD/FLR fall outside of ASIC and ENaC superclades (lines 552-554). How is this possible? Do the authors claim that these DEG/ENaCs evolved independently?

In our new phylogenetic tree in Figure 2, the PPK channels and ACD/FLR channels fall within Clade A, similar to the analysis by Aguilar-Camacho et al. However, when we generated a second phylogenetic tree using the P2X receptors as outgroup, both the ACD/FLR and PPK channels fell outside of Clade A, branching off near the base of the tree after the P2X receptors (new Supplementary Figure 1). We do not believe these channels evolved independently, but that their considerable sequence divergence makes their phylogenetic position less certain than some of the other well supported clades. Nonetheless, we hope that in this version of the manuscript our description of the various clades in our trees is more cautious and clear, and have removed lines 552-554.

Supplementary Figure 1. Maximum likelihood protein phylogeny rooted with phylogenetically-unrelated P2X receptors. Node support values are for 1,000 bootstrap replicates (green), and the asterisks and labels (pink) indicate single channels or clades bearing Deg/ENaC channels that have been characterized as proton-activated.

Moreover, the exact filasterean species containing DEG/ENaCs should be identified (I could not find this information).

Our apologies for not directly indicating this in the text. The information is available in Supplementary Table 1 (BUSCO analysis) where we indicate all species used in the phylogenetic analysis (*i.e.*, the filasterean *Capsaspora owczarzaki* and the filasterean-related *Tunicaraptor unikontum*, heterokonts *Symbyodinium* sp_KB8 and *Cafeteria roenbergensis*). To make this information more evident, we now indicate the following in the results section (lines 114 to 118):

“To better understand the phylogeny of the *T.adhaerens* TadNaC channels, and of Deg/ENaC channels in general, we conducted a CLuster ANalysis of Sequences (CLANS) (27) with gene sequences from representative species spanning the major animal groupings, extracted from high-quality gene datasets with high BUSCO (28) completeness scores (species names and BUSCO scores provided in Supplementary Table 1).”

Finally, the support for the maximum likelihood trees should be calculated as bootstrap values, which is better than the SH-like and aBAYES (which is not “real” Bayesian inference) methods used by the authors.

We have calculated bootstrap values for the new phylogenetic trees, and denoted these in Figure 2 and Supplementary Figure 1.

3) The functional characterization of TadNaC2 is not entirely convincing and is sometimes confusing.

I noticed that at pH 4.5, tachyphylaxis was more pronounced for the transient than for the sustained current (Fig. 4C). Was the time interval between activations sufficient to allow for complete recovery (of the transient current)? Please also analyse tachyphylaxis of the sustained current at pH 4.5.

Thank you for pointing this out. We have quantified the decay in the late current component for our tachyphylaxis recordings, revealing a similar rate of decay compared to the transient current (Figure 4D).

We also modified the results section to indicate that the decay in currents was not attributable to incomplete recovery from desensitization, given that this was similar for a set of paired pulses with either a 40 second or an 80 second interpulse interval (lines 211 to 229):

“In early experiments, we found that *TadNaC2* currents exhibit a non-recovering decay in amplitude upon repeated activation. For example, applying paired 30 second pulses of pH 4.5 solution separated by neutral pH wash steps of either 40 or 80 seconds resulted in similar decreases in amplitude of $55.5 \pm 16.5\%$ with a 40 second interval vs. $50.74 \pm 9.2\%$ with an 80 second interval. Since doubling the interpulse interval from 40 to 80 seconds did not diminish the current decay amplitude, the observed process is not likely due to incomplete recovery from fast/acute desensitization. This feature of *TadNaC2* thus resembles mASIC1a which undergoes slow desensitization or tachyphylaxis, a unique process not observed for ASIC2 and ASIC3 proposed to involve a prolonged inactivated state that is distinct from acute desensitization (37, 38). To better characterize this property of *TadNaC2*, we employed an experimental paradigm roughly similar to one used previously to study tachyphylaxis of mASIC1a in *Xenopus* oocytes (37). Specifically, we applied six 15 second pulses of pH 4.5 or 5.5 solutions over recorded cells expressing *TadNaC2* or mASIC1a, separated by 55 second interpulse intervals. Consistent with observations made in oocytes, mASIC1a peak currents decayed upon repeated activation at pH 5.5 (Fig. 4c), decreasing to $49.4 \pm 9.5\%$ of their original value after 6 pulses (Fig. 4d). Similarly, peak *TadNaC2* currents declined to $57.9 \pm 20.5\%$ at pH 4.5, and $43.5 \pm 16.4\%$ at pH 5.5, while the late/sustained component of the *TadNaC2* current at pH 4.5 declined to $59.5 \pm 22.9\%$. Analysis of the average data revealed that although the decline in current amplitude for each condition relative to the first pulse was statistically significant, the degree and rate of decline between the different channels and conditions was not.”

Figure 4. General electrophysiological properties of *TadNaC2* compared to mouse ASIC1a. (a) Sample recordings of *TadNaC2* currents (top) and mouse ASIC1a currents (mASIC1a, bottom) elicited by perfusion of solutions with decreasing pH. (b) pH dose response curves for *TadNaC2* and mASIC1a revealing a right shifted pH_{50} for the *Trichoplax* channel relative to mASIC1a, and a smaller Hill coefficient (n_H). The values observed for mASIC1a are consistent with previous reports (26). (c) Sample sequential *TadNaC2* currents exhibiting rundown or tachyphylaxis similar to mASIC1a. (d) Plot of average normalized current amplitude \pm standard deviation through successive sweeps for the *TadNaC2* channel (i.e., early and late currents at pH 4.5, and peak current at pH 5.5) and mASIC1a (peak current at pH 5.5), revealing decaying amplitudes for all conditions that are statistically indistinguishable from each other (i.e., $p > 0.05$ for

one-way ANOVAs comparing raw normalized values for each condition at each pulse). The asterisks indicate statistically significant p values (*i.e.*, < 0.05) for pairwise posthoc Tukey tests after one-way ANOVAs of each set of pulses for each condition (*TadNaC2* pH 4.5 early: $p = 1.0E-4$, $F = 7.6$; *TadNaC2* pH 4.5 late: $p = 2.4E-3$, $F = 4.8$; *TadNaC2* pH 5.5: $p = 4.3E-7$, $F = 14.7$; mASIC1 pH 5.5: $p = 1.4E-14$, $F = 38.2$). (e) Sample current recordings for *TadNaC2* and mASIC1a before (black traces) and after (red traces) perfusion of 3 mM amiloride, revealing a nearly complete block for mASIC1a (at pH 5.5) and only ~50% block for *TadNaC2* (pH 4.5). (f) Plot of average percent block of inward current \pm standard deviation for *TadNaC2* and mASIC1a before and after perfusion of 3 mM amiloride. Individual replicates are included as grey circles. B+T indicates the total decay in average current for a successive sweep, which includes the effects of drug block (B) and tachyphylaxis (T), while B indicates the isolated component of drug block alone, obtained by subtracting the average decline in amplitude caused by tachyphylaxis. Denoted p values are for post hoc Tukey's tests after one-way ANOVA ($p = 1.7E-11$, $F = 56.1$). (g) Sample sequential *TadNaC2* currents elicited by perfusion of pH 4.5 solutions bearing increasing concentrations of amiloride. (h) Amiloride dose response curve revealing a more pronounced decline in normalized peak inward current with increasing concentration of amiloride, compared to the rundown observed in the absence of drug attributable to tachyphylaxis.

The authors mention potentiation of *TadNaC2* by 3 μ M amiloride. It would be helpful to have a complete concentration-response curve to see what happens at higher concentrations of amiloride.

We have conducted a new set of experiments to generate a dose response curve for amiloride (new panels g and h in figure 4). In addition, we would like to point out that we made an error in our previous description of the amiloride concentration described in figure 4. That is, where as in the materials and methods section we correctly indicated a concentration of 3 mM, we incorrectly indicated a concentration of 3 μ M in the figure and main text. We apologise for this error which has been corrected in this version of the manuscript.

Why was there no transient current with Na⁺ as the sole extracellular cation (Fig. 5b)? And why was the transient current in this panel discovered only with K⁺?

We do not know what causes this, but in this revised manuscript we corroborate these findings by showing that replacement of external Na⁺ with Li⁺ produces similar currents lacking a fast/transient component. We also modified the text to address this observation (please see our response to the next comment below). We believe this is an interesting feature that will be interesting to study in the future, but that this is outside of the scope of this current study.

In Fig. 5c, the authors compared early reversal potentials with Na⁺ and with K⁺. But these reversal potentials were measured at different time points and cannot be compared. Early and late Na⁺ report the reversal potential of the same current component. Thus, ion selectivity of the early current cannot be calculated. This part is confusing and sheds some doubt on the electrophysiological characterization, in particular with regard to ion selectivity.

We note that we are not the first to measure separate reversal potentials for different components of a single macroscopic current for a biphasic Deg/ENaC channel (*e.g.*, Springauf and Gründer, *The Journal of Physiology* 2010). In our analysis, we identified a clear difference in reversal potential for the early vs. late components of the macroscopic current, recorded at fixed voltages. This observation indicates that the permeation properties are indeed different throughout the course of the biphasic macroscopic current, since the only factor that could account for differences in reversal potential between the early and late components is differences in permeation/selectivity (given that voltage and ion concentrations/equilibrium potentials remain constant for each trace). However, we agree with the reviewer that this part of the manuscript was confusing, and therefore significantly altered figure 5 and the corresponding text in the results section to improve clarity (lines 254 to 286):

"*TadNaC2* resembles mammalian ASIC3 in conducting biphasic macroscopic currents comprised of an early current that activates and desensitizes quickly, followed by a late current that activates and desensitizes more slowly (41). These two components of the *TadNaC2* current become even more distinguishable at pH 3.5, where two separate peaks can be observed (Fig. 5a). We thus wondered whether these two components of the macroscopic current exhibit differences in their ion selectivity. To test this, we employed the bi-ionic reversal potential technique by perfusing different

monovalent cations over recorded cells (Li^+ , Na^+ , and K^+), while maintaining equimolar Na^+ in the internal recording solution, and measuring changes in current reversal potential (voltage where currents reverse from inward to outward) when external permeating ions are altered (42). This technique allows quantification of permeability ratios of desired cations relative to Na^+ ($pX^+/p\text{Na}^+$, where X^+ is the external cation). Recording $Tad\text{NaC2}$ currents at different fixed voltages at pH 4.5, with 150 mM Na^+ on each side of the cell membrane, produced slowly activating currents that lacked a fast transient component (Fig. 5b). As expected, these currents reversed from inward (negative) to outward (positive) near zero millivolts (*i.e.*, 0.87 ± 0.87 mV; Fig. 5c). Replacement of extracellular Na^+ with an equal concentration of Li^+ , which has a smaller ionic radius than Na^+ , produced similar currents that reversed near 0 mV and lacked a transient component (2.51 ± 0.72 mV), indicating that $Tad\text{NaC2}$ is equally permeable to Na^+ and Li^+ . Notably, all our previous recordings made using standard salines with external Na^+ and internal K^+ or Cs^+ ions produced biphasic currents at pH values below 5.5, unlike currents observed under the bi-ionic conditions of $\text{Na}^+_{\text{in}}/\text{Na}^+_{\text{out}}$ and $\text{Na}^+_{\text{in}}/\text{Li}^+_{\text{out}}$. Thus, it appears that the kinetics of the macroscopic current can differ depending on the types of permeating ions present across the cell membrane, an interesting observation that will require deeper analysis in future studies.

Instead, replacement of extracellular Na^+ with equimolar K^+ (*i.e.*, $\text{Na}^+_{\text{in}}/\text{K}^+_{\text{out}}$) produced canonical biphasic currents with a fast transient component and a late sustained component (Fig. 5b). The occurrence of these two clearly delineated current components allowed us to measure reversal potentials for each, revealing that although both exhibit a negative shift in voltage compared to bi-ionic sodium, the late current exhibited a more marked hyperpolarizing shift compared to the early current (*i.e.*, -60.42 ± 2.30 vs. -49.01 ± 1.62 mV, respectively; Fig. 5c). A box plot of the reversal potential data for the different bi-ionic measurements, coupled with ANOVA and post-hoc tests (Fig. 5d), substantiates the negative shift in reversal potentials for both the late and early currents in the presence of external K^+ , reflecting a general preference of $Tad\text{NaC2}$ for Na^+ over K^+ ions. Furthermore, the more significant shift in reversal potential for the late vs. the early current indicates that ion selectivity changes over the course of the biphasic current, such that the early current is less selective for Na^+ over K^+ compared to the late current ($p\text{Na}^+/p\text{K}^+$ permeability ratios of 7.3 ± 0.5 and 11.0 ± 1.1 , respectively; Fig. 5e)."

Figure 5. *TadNaC2* conducts biphasic currents *in vitro* that are relatively insensitive to Ca^{2+} block. (a) Sample current recordings of the *TadNaC2* channel at pH 4 and 3.5, revealing a biphasic current with a fast transient component (*i.e.*, early current), and a slower, sustained (late current) component. The biphasic current becomes more evident at pH 3.5. (b) Sample proton-activated *TadNaC2* currents recorded at different voltages (voltage protocol on top), under bi-ionic conditions of equimolar intracellular Na^+ and extracellular Na^+ (Na^+_{ext}) or extracellular K^+ (K^+_{ext}). The star and square symbols denote portions of the currents that were that were measured to determine reversal potentials. (c) Plot of average reversal potential data (\pm standard deviation) for the bio-ionic reversal potential

Was the transient current with Ca²⁺ as the sole extracellular cation amiloride-sensitive? I do not see a sustained outward current under this condition. Could Ca²⁺ slowly block this channel? In the Discussion section (line 642), it is mentioned that the early *TadNaC2* current is highly Ca²⁺ permeable. But because Ca²⁺ permeability has only been qualitatively assessed, this statement is not supported by the data.

We agree with the last comment from above, and have decided to remove this set of data from the manuscript. In more recent experiments, we have indications that this component of the current might also permeate protons, which will require a much deeper analysis that is outside of the scope of this initial characterization.

4) Figure 7: It would be helpful to show current traces at different pH values for selected mutants.

Thank you for this suggestion. We have added traces for all mutant channels with significant differences in pH₅₀ and current amplitude relative to wildtype (new panel f in Figure 7). We also include additional recordings of the E105 single mutant, to increase the replicate number for this particular variant. Moreover, we removed the Hill coefficients because these were quite variable and difficult to compare between channel variants.

Figure 7. Unique residues contribute to proton-activation of *TadNaC2*. (a to c) Average pH dose response curves ± standard deviation for wildtype (wt) *TadNaC2* and various variants bearing amino acid substitutions within the wrist (panel a), finger (panel b), and palm (panel c) regions. (d and e) Plot of average pH₅₀ (panel d) and normalized peak current amplitude ± standard deviation for wildtype (wt) and various point mutated *TadNaC2* channels. Cyan-colored asterisks denote p value thresholds for two-sample t-tests comparing wildtype to mutant values. (f) Sample whole-cell currents of wildtype *TadNaC2* and select mutant variants.

5) The authors should mention that there are also bona fide ASICs that are activated by protons and lack the critical histidine corresponding to H73.

To the best of our knowledge, all wildtype ASIC channels that have been characterized to date bear the critical histidine residue, and most (but not all) also possess the K211 residue. Our apologies if we are mistaken, and we would sincerely appreciate an indication as to which manuscript(s) bear this information. Nonetheless, we have modified the text for clarification as follows (lines 305 to 321):

“A protein alignment of several regions bearing these and other determinants for proton-activation of ASIC channels, including the group A ASIC channels from mice (*i.e.*, ASIC1 to 4), selected group A and B channels from *Branchiostoma belcheri* (25), and the singleton group B channel from *Lingula anatina* (10) reveals near complete conservation of the H73 and K211 residues (Fig. 6b). The only exception are the proton-insensitive ASIC2b splice variant which lacks H73 (47), and ASIC4 which is also proton-insensitive and lacks K211 (48). To the best of our knowledge, all functional group A and B ASIC channels that have been experimentally characterized *in vitro* bear a conserved H73 residue, and most bear a K211 equivalent (10). The mouse ASIC5/BASIC channel, which falls in a separate clade from ASIC channels (Fig. 2) and is not activated by protons, lacks both the H73 and K211 residues. These residues are also absent in other Deg/ENaC channels that are sensitive to external protons *in vitro*, including the proton-inhibited *T. adhaerens* channel *TadNaC6* (24), and the proton activated channels *TadNaC2*, the ENaC- δ channel from human (6), the channels ACD-2, DEL-9, and ASIC-1 from *C. elegans* (14), Pickpocket1 from *D. melanogaster* (16), and *NeNaC2* from the sea anemone *Nematostella vectensis* (8). However, *TadNaC2*, as well as the mouse ASIC4 and BASIC channels, possess a cationic residue just one amino acid upstream of the K211 position (*i.e.*, R201 in *TadNaC2*). *TadNaC2* and its *Hoilungia hongkongensis* orthologue *HhoNaC2* also possess a conserved lysine one position downstream (K203 in *TadNaC2*).”

6) The question arises as to whether the relatively low proton sensitivity of *TadNaC2* is relevant to *Trichoplax*. A short discussion of the natural environment of this organism and the pH fluctuations that it might experience would be helpful.

Thank you for this suggestion. We have added the following section to the discussion (lines 634 to 662):

“Potential roles for proton-sensing Deg/ENaC channels in placozoans.

T. adhaerens is simple seawater invertebrate that lacks body symmetry, has only six ultrastructurally distinguishable cell types, and lacks a nervous system and synapses (66). Nonetheless, the animal is able to coordinate its various cell types to conduct directed locomotion including chemotaxis (67), gravitaxis (68), and thermotaxis (69). Locomotion is achieved via the action of asynchronously beating cilia on its ventral epithelium, coupled with secretion of mucous thought to facilitate ciliary gliding (70). Furthermore, various endogenous neuropeptides have been identified that when applied ectopically regulate *T. adhaerens* locomotive behavior (71, 72). Interspersed among the ventral ciliated cells are Lipophil cells which bear large acidic vesicles that are secreted over food algae during feeding (66). Secretion of these substances causes rapid breakdown of algae (*i.e.*, within seconds), after which the animal is conducts churning motions thought to promote nutrient uptake (73). Interestingly, a single cell transcriptome study of *T. adhaerens* revealed that *TadNaC2* is expressed in lipophil cells (74). In agreement, we previously showed through fluorescence *in situ* hybridization that *TadNaC2* mRNA is restricted within a central region of the animal (24), consistent with what is referred to as the lipophil zone where these namesake cells reside (70). Instead, mRNA of the proton-blocked channel *TadNaC6* was detected along the periphery of the flat, plate-like animal (24), in a region that contains neuroendocrine-like gland cells that also express voltage-gated calcium channels (75, 76). The expression of *TadNaC2* within the lipophil zone positions it to detect sudden drops in pH upon secretion of acidic vesicles at the ventral epithelium during digestion. Since the aqueous volume between the epithelium and the underlying substrate is expected to be quite small, lipophil secretion and external digestion could conceivably lower the extracellular pH to a range that effectively activates *TadNaC2*, allowing the channel to provide feedback on external digestion by depolarizing lipophil cells.

Furthermore, we have preliminary evidence that *T. adhaerens* will contract upon acute exposure to acidic extracellular pH, and chemotax away from an acidic pH environment (77). However, whether the proton-sensitive

channels *TadNaC2* and *TadNaC6*, or other yet uncharacterized *TadNaC* channel complexes contribute to these pH-sensing behaviors is unknown. Indeed, various other membrane receptors, including G protein-coupled receptors, can also detect extracellular protons (78). Clearly, future studies in placozoans involving the development of gene ablation techniques will be required if we are to understand the physiological functions of *Deg/ENaC* channels in this important group of animals.”

7) Lines 582-591: I found the section on the relationship of *DEG/ENaCs* to *P2X* receptors and *PAC* channels highly speculative with no experimental support. I recommend deleting this part of the manuscript.

We have removed this part of the discussion.

Minor comments

1) Approximately half of the introduction is a summary of the findings. This part could be substantially shortened to briefly summarize only the most important findings.

We have made substantial deletions to this section of the introduction, leaving only the most important findings.

2) The abstract claims that *TadNaC2* has strikingly similar biophysical features to *ASICs*, but the Results show that this channel has rather some unique features.

We have changed this part of the abstract to highlight how *TadNaC2* shares unique features with each of the three mammalian *ASICs* (lines 33 to 37):

“Here, we report that the non-bilaterian invertebrate *Trichoplax adhaerens* possesses a proton-activated *Deg/ENaC* channel, *TadNaC2*, with a unique combination of biophysical features including tachyphylaxis like *ASIC1a*, reduced proton sensitivity like *ASIC2a*, biphasic macroscopic currents like *ASIC3*, and weak sensitive to block by amiloride and Ca^{2+} ions.”

3) Mention the pH used on Figure 5B.

We have added the following to the results section describing these experiments (lines 264 to 265):

“Recording *TadNaC2* currents at different fixed voltages at pH 4.5, with 150 mM Na^+ on each side of the cell membrane, produced slowly activating currents that lacked a fast transient component (Fig. 5b).”

4) Figures 3-5: I suggest labeling lines in the graphs directly and to delete the legend from the figures (including “AVG \pm SD”, an information that should be provided in the real legend below the figures). This makes it easier for the reader.

Thank you for this suggestion. We made some revisions to the legends which we hope make interpreting the plots easier, but feel these help in the interpretation and hence would like to keep them.

5) Line 410: Should it read “K203” instead of “K205”?

Thank you for noting this mistake. We have corrected it.

6) Line 432: the authors mention that they wanted to determine whether channel membrane expression was reduced. But in this paragraph, they only assessed total protein expression.

We have changed this sentence to the following (lines 438 to 439):

“Next, we wanted to determine whether the noted decrease in current amplitude caused by select mutations was due to reduced functionality or a reduction in channel protein expression.”

7) Lines 630-631: because *ASIC3* develops a sustained current (different from its window current) only at very low pH, it is an outdated concept that this sustained current contributes to nociceptive signaling.

We have removed this statement.

8) Lines 636-640: personally, I found this Discussion to be highly speculative. This is just a note for consideration.

We have removed these two sentences.

9) Lines 679-681: the authors mention results from P71H and T72H mutants, which are not shown. Either the results should be shown or this sentence deleted.

This was also suggested by reviewer 3. We have deleted this part of the discussion.

10) Lines 687-688: “which is notably absent in HhoNaC2”. Because there is no functional data for HhoNaC2, it is not clear why the absence is notable.

We have changed this sentence to the following (lines 589 to 591):

“Alanine mutation of R201 in *TadNaC2*, which is absent in the *H. hongkongensis* orthologue *HhoNaC2*, enhanced proton sensitivity evident in an increased pH_{50} value (Fig. 7d).”

11) Line 696: Please provide a reference for the hydrophobic hub.

We have cited the appropriate Lynagh *et al.*, (2008) paper.

12) Lines 699-702: it was not completely clear to me why H80 in *TadNaC* fulfills functions similar to K211 in ASICs.

We have reworded this section to make our suggestion clearer (lines 596 to 604):

“In ASIC channels, a hub of hydrophobic residues that lies adjacent to K211 (*i.e.*, residues F87, F174, F197, and L207), is thought to functionally couple conformational alterations between the palm and wrist regions during gating (25). These hydrophobic residues are conserved in *TadNaC2*, which has four hydrophobic residues in corresponding amino acid positions (*i.e.*, F86, W165, F188, and L198; Supplementary Figure 2). Moreover, the K211 residue in ASIC1 is thought to form transient inter-subunit interactions with acidic residues in the thumb region of the adjacent subunit which contribute to activation (61). Indeed, future studies will be required to determine whether the non-homologous H80 residue in *TadNaC2*, which contributes to proton activation and resides in a similar structural region as K211 (Fig. 6d), similarly operates through the hydrophobic hub and inter-subunit interactions.”

13) Lines 727-729: The K203Delta mutant may lead to a loss of function by many mechanisms. That it imposes instability on homotrimeric assembly is a speculation.

We have reworded this sentence to make it clear that we are merely proposing possible explanations for our observations (lines 630 to 632):

“This indicates that this particular mutation prevents trafficking to the cell membrane, perhaps via disrupted assembly of the homotrimeric channel, or aggregation of channel subunits.”

14) Line 730: “ability to record current for these variant channels” – there was no current with K203Delta.

We have removed this sentence as we think it was unnecessary.

Reviewer #2 (Remarks to the Author):

In "Independent evolution of an ASIC-like Deg/ENaC sodium channel in the Placozoa" Elkhatab et al., present clustering analysis and a detailed phylogenetic analysis of the ASIC/ENaC channel superfamily, new functional and structural analysis of a proton-activated placozoan ortholog and analysis of behavioral responses to pH changes in placozoans. The authors state their phylogeny resolves true evolutionary relationships between important ASIC/ENaC clades that have been missed in similar recent analysis because of refinements in their phylogenetic methodology. The authors posit that proton-activation is independently evolved in their placozoan ortholog TadNaC2 and vertebrate ASIC channels based on differences in the underlying mechanism and separate branching in the phylogeny and point to several other independent evolutions of proton gating in the channel family that others have observed. The results from behavioral analysis show several acid-dependent behaviors, only one of which appears sensitive to the non-specific ASIC/ENaC blocker amiloride. The manuscript concludes that they have resolved the true evolutionary relationships of ASIC/ENaC channels and point out potential flaws in previous analyses, but they do not discuss major caveats with their own analysis and do not provide all the information required to assess its validity. Results on the proton-gating mechanism of TadNaC2 are similarly overinterpreted, though the conclusion the mechanism differs from ASIC channels appears solid. It also appears premature for the authors to include behavioral analysis on proton-dependent behaviors in the same manuscript as the TadNaC2 work. Since they do not have the ability to provide genetic evidence or even solid pharmacological evidence of a link between these behaviors and TadNaC channels, the behavioral analysis should be presented elsewhere as a separate study. There are numerous examples of proton dependent behaviors that do not involve these channels and including it here biases the field to see a link that might not exist. Specific comments on issues in the manuscript that need to be addressed are listed below.

Major issues:

1. While the cluster analysis is a useful tool for visualization of the relatedness of sequences in this channel family, its use in selecting sequences for phylogenetic analysis as explained is dubious. The key issue seems to be how the cut-off score for displaying linkage was decided. The chosen value makes a nice display in Figure 1, but is there a scientific reason the value was chosen or was it arbitrary? This matters because the cluster analysis was used to select sequences for phylogenetic analysis. Would those results be the same if a higher or lower cut off value was used? Selective elimination of data is sometimes warranted, but it needs to be carefully justified.

Thank you to the reviewer for this comment. The cluster-based analysis of sequences is based on an all-vs-all sequence similarity assessment (based on e-value). Like BLAST, there is not a "one-size-fits-all" cut-off, and this will depend on the type of protein being analyzed. While a threshold of $1e-2$ is suitable for neuropeptides (very short proteins), it is inadequate for receptors or ion channels because it will create unreliable connections. As such, the cut-off values were determined by carefully testing different e-values. In this case, we know "a priori" that Purinergic and Chloride channels are an evolutionarily distinct group of receptors, and they were used as a guide for the clustering. We chose the e-value of $1e-40$ because it allowed the connection of divergent channels, such as the PPKs from *Drosophila* and FLP from *C. elegans*, which are known to be real Deg/ENaC/ASIC-related channels. However, it still discriminates against the non-related P2X and PAC channels. At lower e-values, the PPK receptors were no longer connected, and at higher e-values, the PAC channels connected to the Deg/ENaC channels. We agree that this method may not be the "final solution" to the problem, and we concur with the reviewer that the results would not have been the same if a higher or lower threshold were chosen. Nevertheless, we believe that the chosen threshold was lenient enough to allow the identification of divergent channel sequences and stringent enough to not allow the connection of Purinergic P2X or chloride channels as part of the Deg/ENaC cluster of receptors, which we know are evolutionarily different groups of receptors.

2. The phylogenetic analysis is lacking some key details that are needed to assay how strong the results truly are. Most importantly there are no details on where the phylogeny was trimmed and how many informative positions it contained. File S4 is supposed to have the trimmed alignment but it looks like a full untrimmed alignment and it is impossible to determine the number of positions from it. Because the majority of the alignment is gap, it's hard to visualize how the sequences align to each other and whether the alignment is reasonable. In this gene family the distances between key nodes are very small in comparison to amount of sequence divergence that has occurred afterwards. If the number of phylogenetically informative positions is small (just a couple hundred amino acids for example), then only a very few positions are determining the branching orders the authors state this study resolves, and makes that statement an overreach. The results presented here would be easier to consider as a "resolution" or a significant advance over other published studies if the authors showed results were robust to sequence selection (cut off for inclusion) and phylogenetic method (is it sensitive to algorithm or selection model, for example). Alternatively if they feel they can present a detailed justification as to why their sequence selection cut-offs, alignment trimming and phylogeny methods are the best choices to answer the question in hand, they could approach it in that way. The authors point out valid potential caveats with recent, but they don't point out these caveats of their analysis. It is really difficult to build robust phylogenies from short alignments of large poorly-conserved gene families, and the truth is the detailed evolutionary history may not be fully resolvable. It is relatively easy to produce statistically robust trees by eliminating the right sequences, but does this really get to the true answer?

We agree with the reviewer and apologize for this error. We intended to upload the trimmed alignment but mistakenly uploaded the non-trimmed alignment. Based on the comments from reviewers, we conducted two new phylogenies with two different outgroups. This time, we included the details about the trimming (performed with trimAl) and the final number of informative positions (397) in the alignment used for the phylogeny (added to the methodology section). We also provided supplementary files for each phylogeny, including both aligned and trimmed-aligned sequences. This will allow for an assessment of the robustness of the alignment used for the phylogeny. We agree with the reviewer that the high divergence observed in several channel sequences makes it challenging to definitively resolve the evolutionary history of these receptors. It is not our goal to provide a final answer on the evolutionary patterns of these receptors. As such, we have toned down the section discussing the potential evolutionary patterns of the channels.

3. Despite the caveats above, it is interesting that this is the 2nd recent study to find the two distinct ancient clades of these channels in animals, and most overlapping sequences go to the same clade in both studies. These parts of the results seem most robust, and it may be that the differences between the two studies are less robust. The discussion on these points should be more balanced. It seems that both studies have some potential value to offer in supporting each other. The authors should also not speculate that long branch attraction as the reason for the differences between the studies unless they can point to evidence that this has occurred and is responsible for the differences. They should also consider if any of the branching patterns in their tree could also be influenced by long branch attraction.

In this version of the manuscript, we made efforts to highlight the similarities between our findings and those by Aguilar-Camacho *et al.*, rather than focus on differences. Furthermore, our updated phylogenetic tree is even more in agreement with the previously published work. We also refocused our discussion on the *Trichoplax* channels and various novel findings that our analysis brings forward.

4. The choice of divergent *Drosophila* and *C. elegans* channels as an outgroup is inappropriate, and the tree should be run again either with a true outgroup or it could simply be presented as an unrooted phylogeny. But the selected outgroup channels are neither an outgroup by species or by evolutionary origin. Any phylogeny of the gene family rooted to them by definition does not reflect the true evolutionary relationships within the gene family. They must logically branch from somewhere within the ASIC or ENaC groups, the tree is not able to resolve where because of their high sequence divergence. But presenting them as an evolutionary outgroup is not at all accurate to the evolutionary history of the gene family. It does illustrate a major difficulty with phylogenetic analysis of large gene families – new duplicates are often under lower or divergent selective pressures and thus change faster and sometimes appear basal in phylogenies even though we know they are derived.

We agree with this suggestion which was also made by reviewer 1. In this version of the manuscript we present two new phylogenetic trees using either filasterean Deg/ENaC channel sequences (Figure 2) or a set of metazoan P2X receptor sequences as the root (Supplementary Figure 1).

5. The paper conclusions are based on the premise that the ancestor of the ASIC group was not proton activated. The alternative hypothesis, that the ASIC group ancestor was proton activated and that proton sensitivity has been lost in vertebrate BASICs is not considered, but is actually better supported by the phylogenetic distribution of proton-activated vs. proton-insensitive channels. The best evidence to support proton-insensitivity of the ancestor is that the mechanisms of proton activation in vertebrate ASICs and *TadNaC2* are different. But this may not suggest the absence of low affinity proton activation in the ancestor (as seen in *TadNaCs*), but instead addition of high proton affinity in ASICs. There simply isn't enough functional data to distinguish between these possibilities, and the merits of both should be considered.

Thank you for the valid point. We have added the following to the discussion to address this alternate hypothesis (lines 526 to 543):

“Numerous non-ASIC Deg/ENaC channels have been identified that are also activated by extracellular protons but lack most key residues involved in proton-activation of ASIC channels. This includes *TadNaC2*, the human ENaC- δ channel (6) and the *C. elegans* channels ASIC-1, ACD-2, DEL9, and (14), which form homotrimeric channels *in vitro* that conduct slow onset proton-activated currents with minimal desensitization, the *NeNaC2* and *NeNaC14* channels from *N. vectensis*, which respectively conduct moderately and non-desensitizing currents *in vitro* (8), and the *D. melanogaster* channel PPK1, which conducts transient, fast desensitizing cation currents in sensory neurons (16). Notably, ASICs, *TadNaC2*, *N. vectensis NeNaC2* and *NeNaC14*, chordate ENaC- δ , and *C. elegans* ASIC-1/ACD-2 are all separated by intervening lineages of Deg/ENaC channels that are not known to be proton-activated (Fig. 2 and Supplementary Figure 1). One possible explanation for this is that proton-activation evolved numerous times independently, which is supported by the absence of key molecular determinants for proton-activation of ASIC channels in all non-ASIC proton-activated channels (Fig. 6b). Alternatively, the ancestral channel that gave rise to Clade A and B channels was proton-activated, and this gating feature was then lost in the various lineages that are not proton-activated, including BASIC channels, *HyNaC* channels, *TadNaC6*, *FaNaC* and *WaNaC* channels, and the *C. elegans* channels MEC-4 and MEC-10. Although neither of these scenarios can be discounted with certainty, it is clear that future functional characterization, particularly of channels from lineages with unknown gating properties, will be important for better understanding Deg/ENaC channel evolution.”

6. The data used to show alterations in proton sensitivity for various mutants is a not great, in a large part because the small sequence sizes give measurements with large errors.

We hope that the inclusion of sample recordings for key mutations (Figure 7f) in this version of the manuscript better illustrates how certain mutations significantly impair channel proton-activation.

But more importantly, there is no control for maximal channel opening. Without this, it is impossible to tell if these mutations selectively reduce proton activation or simply make it harder to open the channels that do get to the plasma membrane. This leaves the only good evidence as changes in affinity, and the data quality is not high enough in most cases to give a lot of confidence. It is at least suggestive but not definitive, and leaves some risk it won't pan out. The data that the mechanism differs from ASIC is strongest.

We believe that it would be very difficult to determine maximal channel opening of *TadNaC2* in CHO-K1 cells. Mostly, this is because of the variable expression of the channel and its variants when expressed in CHO cells. Unlike Oocytes where a fixed amount of mRNA can be injected, the transfection efficiency is extremely variable among co-cultured and transfected cells. This is evident for example by the large range of fluorescence intensity observed between adjacent cells after transfection, indicating different levels of EGFP expression from the bicistronic vector pIRES2-EGFP. Furthermore, the levels of recombinant channel proteins in mammalian cells can vary significantly depending on subtle differences in culture duration, media components (*e.g.*, quality/lot number of fetal bovine serum), and temperature.

Secondly, the observed rundown of *TadNaC2* currents indicates that maximal currents are subject to change upon pre-exposure to varying pH, which would happen as soon as any dish bearing multiple cells was subjected to perfusion.

Reviewer 3 raised a related concern, suggesting that perhaps the reduction in amplitudes was influencing the pH dose response curves (*i.e.*, different amplitude currents exhibit different proton sensitivities). However, we do not believe this to be the case as we did not find any correlation (positive or negative) between current amplitude and pH_{50} . To help clarify this, we have added the following to the manuscript (lines 416 to 420):

“We note that this difference in pH sensitivity is not attributable to the decrease in current amplitude, since macroscopic current amplitudes varied significantly for the wildtype channel (Fig. 7e), while the pH_{50} values were much less variable (Fig. 7d), and we found no correlation between amplitude and pH_{50} within the wildtype dataset.”

In contrast, we think there *is* a correlation between recorded current amplitude (in Figure 7E) and membrane expression as measured in our biotinylation assays (Figure 8). Thus, we are confident that the analyzed mutations that impact channel activation and pH sensitivity separately impact current amplitude via reduced expression, both total and at the cell membrane.

7. The animal behavioral data is interesting but does not belong here because it doesn't provide significant evidence for the involvement of *TadNaCs* and there are many other mechanisms that produce proton sensitive behaviors. Amiloride is not specific enough to conclusively establish involvement of *TadNaCs*, and most of the behaviors are insensitive to it as well. Presenting this data here unfairly establishes an expected link to *TadNaCs* over other possible mechanisms that isn't warranted at this point. However, the data is interesting in its own right and could probably stand on its own in a 2nd separate publication.

In light of this suggestion, which was also made by reviewer 1, we have removed the behavioral data from the manuscript with the intention of expanding and publishing it separately.

Very minor point:

There are a number of places where the authors use superlatives like “Remarkably” for phenomena that aren't unusual at all. For example, it isn't remarkable that you can get big currents from non-codon optimized channels, when many of the historically best expressing proteins in vertebrate expression systems are invertebrate orthologs that haven't been codon optimized. Its also not remarkable that *Trichoplax* can respond to pH changes without a nervous system when virtually every living organism across the tree of life responds to pH changes. ON the other hand, it is remarkable how similar P2X and ASIC folds are despite the huge sequence divergence. More reserved use of superlatives would focus attention to the places they are really needed.

We made efforts to remove this word when deemed unnecessary throughout the manuscript.

Reviewer #3 (Remarks to the Author):

The authors report the discovery of a proton-activated Deg/ENaC channel in the non-bilaterian early branching phylum Placozoa, called TadNaC2. Using sequence comparison, structural modeling, and mutation analysis, the authors show that TadNaC2 proton-activation differs substantially from the phylogenetically related Acid Sensing Ion Channels (ASICs), exclusively present in bilaterian animals.

Furthermore, the results of this study suggest that proton activation developed many times independently within the Deg/ENaC superfamily.

The findings in this study will be not only relevant in the field of ion channels, but in the field of protein evolution and the emergence of protein function.

Overall, this study is of good quality, including the methods, results, and data interpretation.

I have only one concern. One of the main points of the manuscript is that the newly discovered TadNaC2 channel and ASICs channels proton-activation are fundamentally distinct from one another, necessitating certain residues in the palm and finger regions (i. e. H80 and H109). However, the authors show that the protein level for those mutants (i.e. H80A and H109A) in transfected cells is lower than WT protein levels. Authors should consider transfecting cells with lower WT vector concentration (with current amplitudes similar to mutants) to assess that the shift in the pH₅₀ is not due to low protein level.

We do not believe that the current amplitudes had a bearing on pH₅₀. We have added the following statement to the results section to indicate our reasoning (lines 416 to 420):

“We note that this difference in pH sensitivity is not attributable to the decrease in current amplitude, since macroscopic current amplitudes varied significantly for the wildtype channel (Fig. 7e), while the pH₅₀ values were much less variable (Fig. 7d), and we found no correlation between amplitude and pH₅₀ within the wildtype dataset.”

Furthermore, authors complement their idea of different residues are needed for proton activation in the different families (i. e. H80 in TadNaC2 and H73 in ASICs), by making TadNaC2 looks like an ASIC (i.e., mutating P71 and T72 to H in a TadNaC2 with H80A background). Because the authors could not record currents from those mutants, they claim that those positions are not equivalent between channels. However, the authors do not show if the construct is expressing, so the lack of function could be due to a lack of protein and not a lack of rescue.

We agree and have removed this statement from the manuscript.

In addition to the point mentioned above, the following minor points should be considered in further versions of the manuscript:

- Authors use the term “chordate” as a synonym of “vertebrate” or “craniate” which is not completely true. “Chordate” refers to a group of animals which include “Cephalochordata”, “Urochordata” and “Craniata/Vertebrata”. In lines 166, 169, 189, and 557 authors should change “chordate” to “vertebrate”. In Figure 2 authors should change “Chordate/Cephalochordate ASICs (Group A)” by “Vertebrate//Cephalochordate ASICs (Group A)”, “Chordate/Ambulacraria/Lophotrochozoan ASICs (Group B)” by “Cephalochordate/Ambulacraria/Lophotrochozoan ASICs (Group B)”, “Chordate/Urochordate BASIC channels” by “Vertebrate/Urochordate BASIC channels”, and “Chordate ENaC channels” by “Vertebrate ENaC channels”.

We apologize for this oversight and thank the reviewer for noting this mistake. We have corrected the nomenclature in the text and figures to not conflate the term chordate with cephalochordate, urochordate, and vertebrate.

-Authors define two superclades among the DEG/ENaC family (i.e., ASIC and ENaC superclades), however, IMHO assigning a superclade name based on the function of a small portion of the channels of that big clade could bring some misinterpretations. I encourage the authors to use a more neutral nomenclature (e. g. Aguilar-Camacho et al., 2023).

We agree with this suggestion. We have changed the names of these clades to Clades A and B, in accordance with the nomenclature put forward by Aguilar-Camacho *et al.*

-Figure 2. Branch names like “Ambulacrarian FaNaC/WaNaC- related channels” can bring misunderstanding. Assigning a function-related name to an untested channel knowing how variable in functions are DEG/ENaC genes is risky. I recommend reconsidering.

We have changed the names of these clades to merely stipulate the groups of organisms included therein.

- Supplementary Table 1. Maybe authors could consider adding a third taxonomic column (e.g. Clade/Phylum/Subphylum) to make the categories more adequate (e.g. Cephalochordata is not a Phylum is a Subphylum).

Thank you. We have revised this table to ensure correct nomenclature.

- Line 307. Full species names (binomial nomenclature) should be employed the first time that an animal is mentioned. (i.e., *Branchiostoma belcheri* and *Lingula anatina*).

We have made this correction.

- Line 415. Which is the mutant that the authors refer to H80A or H109A?

Our apologies, we incorrectly described H109A as H80A. In this version of the manuscript, we made some changes to this paragraph as we incorporated the sample recording data into the results (Figure 7f), and this particular sentence was removed.

- Line 424. As the authors have shown is not that the mutation disrupted the function, it seems that the channels are not expressed in the membrane.

Thank you for noting this. We have changed this sentence to the following (lines 430 to 431):

“However, deletion of the K203 residue (K203Δ) completely prevented us from recording currents for this channel even with very acidic pH.”

- Line 427. K2034E. Can you fix it?

We have made this correction.

- Line 439. Can the authors show the data?

We have added a sample current trace for the EGFP-tagged channel (new Supplementary Figure 3).

Supplementary Figure 3. Sample traces recordings of untagged and EGFP-tagged *TadNaC2* channels in response to activation by a pH 4.5 solution.

- Line 896. Can the authors confirm the concentration was “micro-” and not “mili-” molar?

Our apologies, as noted above, the 3 mM concentration as indicated in the materials and methods was correct, while the 3 μ M described in the results and in the figures was incorrect. We have changed all values throughout the manuscript to 3 mM.

-Figure 6b. Mammalian numbering of the residues. There are two D345, I guess the second is D407. Can the authors fix it?

Thank you for noticing this. We have made the correction to the figure.

Reviewers' comments:

Reviewer #1 (Remarks to the Author):

The author have substantially improved their manuscript and adequately responded to my comments. I congratulate them on this nice paper!

I have two remaining minor comments:

- 1) Line 220: in the cited study on tachyphylaxis, the authors used rat ASIC1a, not mouse ASIC1a.
- 2) Lines 310-312: one exception of a functional ASIC that lacks a H73 residue is zebrafish zASIC1.1 (Paukert et al., 2004, J Biol Chem. 279:18783-18791).

Reviewer #2 (Remarks to the Author):

The revised version of "Independent evolution of an ASIC-like Deg/ENaC sodium channel in the Placozoa" Elkhatab et al., includes extensive edits in response to a wide range of reviewer comments. Many of these have improved the manuscript significantly, including removing the unrelated (but interesting) animal behavior data and correcting fundamental errors in the phylogenetic analysis. The latter reduces the novelty of the manuscript by bringing its evolutionary conclusions more in line with a previous study, but there is still enough novelty here, primarily in the functional characterization of Trichoplax channels, to justify publication. However, there are still a couple of major issues left over from the original manuscript that need to be substantively addressed before the study is suitable for publication.

First, clarifications are still needed to justify inclusion of the Cluster Analysis (Figure 1). As explained in the rebuttal, the cut-off value for clustering was chosen because it separated P2XR and PAC clusters that are known to have different evolutionary origins and left the Deg/ENaC family together. None of this justification is included in the revised manuscript, and it would need to be. But if all the analysis is used for is to exclude channels that could be excluded a priori, what is the scientific purpose?

Furthermore, there are numerous small clusters that are not connected to the main supercluster that are annotated to include channels from the superfamily in question, including many if not all of the PPKs. These are not noted in the rebuttal letter which would lead one to expect a very different looking figure with 3 clusters (Deg/ENaC, PAC, P2XR) Were these smaller DEG/ENaC-containing clusters included or excluded in the phylogenetic analysis? If excluded, this comes back to the original point that the cut-off value is arbitrary and not scientifically useful if it excludes known family members. In this case, cluster analysis should not have influenced the phylogeny. If instead the small isolated clusters were included, then why present the cluster analysis at all? It would essentially have no purpose. My point is it needs to be run and presented in a scientifically rigorous way (and described in complete detail), or simply deleted from the manuscript. But perhaps I am confused because of the incomplete descriptions.

Second, multiple reviewers pointed out that the data on proton sensing mechanism (Fig. 7) were not high quality and made the related conclusions suspect. This problem was not addressed with new data, but it should have been. In short, the data for H109A and H80A is simply not of sufficient quality to support the conclusion that these two residues play a significant role in proton activation. Whether the proton dose response curves are significantly shifted (or not) looks like it depends entirely on which points are viewed as outliers. In H109A for example, the curve is fit as if pH 5 is an outlier and pH 4.5 is accurate (both can't reside near the curve) and the choice appears arbitrary. However, if this was reversed and pH 4.5 was treated as an outlier, then pH sensitivity of the mutant would appear very similar to WT. The same can be said for H80A, and the example traces provided for both mutants do not inspire confidence. It looks like data could be contaminated by desensitization issues or just highly variable due to small current size? There is a real danger here that the conclusions based on this data will not hold up. The authors should either remove these conclusions or provide additional high quality data that convincingly supports the conclusions that these mutants have significantly altered proton sensitivity.

Reviewer #3 (Remarks to the Author):

In the second version of the manuscript "Independent evolution of an ASIC-like Deg/ENaC sodium channel in the Placozoa" by Elkhatib et al., the authors have made major changes in text and figures, content, and structure. As required, the authors have removed the behavioral experiments from the manuscript improving the clarity and message of the manuscript.

After careful consideration, I can say that the current version of the manuscript addresses most of my concerns. However, there are a few areas that require improvement. Regarding mutants R201A and K203A, some statements can be misleading. Authors claim that mutations R201A and K203A "produced different effects" (line 426). However, both mutants significantly increase the proton sensitivity compared to WT (i.e. from a pH50 of around 5.1 in the WT to 5.3 in K203A and 5.5 in R201A, both with a p-value equal to or lower than 0.05). In the same paragraph, the authors say that R201A and K203A mutants "did not alter the current waveform significantly" (line 428). However, the ratio between the sustained component and transient component of the current is clearly different than WT. I suggest the authors calculate the relation between current amplitudes for sustained and transient components as an indicator of the waveform. The authors should, also, show current traces for K203A mutants (I could not find them).

Because of the large number of changes throughout the manuscript some figures and text had become outdated. For example, authors have removed the Hill coefficient from Fig. 7, but they still refer to it in the text (Lines 574 and 583). Line 623, "Fig. f" consider changing to "Fig. e". Line 627, "Fig.7 g to j" should be "Fig. 8". Fig. 8 legend (line 1081) panel "i" should be "c".

Finally, the authors use sources that have been questioned in previous versions of the manuscript to create a theoretical context (line 657). I suggest the authors reconsider if they want to add that reference.

Overall, I believe that with those few modifications, the manuscript presented by Elkhatib et al. will be a valuable contribution to scientific literature.

We would like to thank the reviewers for their thoughtful comments and suggestions. Below we include all relevant revised text and figures that we hope adequately address the remaining concerns.

Reviewer #1 (Remarks to the Author):

The author have substantially improved their manuscript and adequately responded to my comments. I congratulate them on this nice paper!

Thank you very much for this comment and your useful suggestions on both versions of the manuscript.

I have two remaining minor comments:

1) Line 220: in the cited study on tachyphylaxis, the authors used rat ASIC1a, not mouse ASIC1a.

We have corrected this (in the results section) as follows:

“This feature of *TadNaC2* thus resembles the rodent ASIC1a channel which undergoes slow desensitization or tachyphylaxis, a unique process not observed for ASIC2 and ASIC3 proposed to involve a prolonged inactivated state that is distinct from acute desensitization (37, 38). To better characterize this property of *TadNaC2*, we employed an experimental paradigm similar to one used previously to study tachyphylaxis of rat ASIC1a in *Xenopus* oocytes (37). Specifically, we applied six 15-second pulses of pH 4.5 or 5.5 solutions over recorded cells expressing *TadNaC2* or rat ASIC1a, separated by 55-second interpulse intervals. Consistent with observations in oocytes, rat ASIC1a peak currents decayed upon repeated activation at pH 5.5 (Fig. 4c), decreasing to $49.4 \pm 9.5\%$ of their original value after 6 pulses (Fig. 4d).”

2) Lines 310-312: one exception of a functional ASIC that lacks a H73 residue is zebrafish zASIC1.1 (Paukert et al., 2004, J Biol Chem. 279:18783-18791).

We have updated the text accordingly:

“A protein alignment of several regions bearing these and other determinants for proton-activation of ASIC channels, including the group A ASIC channels from mice (*i.e.*, ASIC1 to 4), selected group A and B channels from *Branchiostoma belcheri* (25), and the singleton group B channel from *Lingula anatina* (10) reveals near complete conservation of the H73 and K211 residues (Fig. 6b). The only exception are the proton-insensitive ASIC2b splice variant which lacks H73 (47), and ASIC4 which is also proton-insensitive and lacks K211 (48). Indeed, except for the zebrafish ASIC1 homologue zASIC1.1 (10), all functional group A and B ASIC channels that have been experimentally characterized *in vitro* bear a conserved H73 residue and most bear a K211 equivalent. The mouse ASIC5/BASIC channel, which falls in a separate clade from ASIC channels (Fig. 2) and is not activated by protons, lacks both the H73 and K211 residues. These residues are also absent in other Deg/ENaC channels that are sensitive to external protons *in vitro*, including *TadNaC6*, a proton-inhibited channel from *T. adhaerens* (24), and the proton-activated channels *TadNaC2*, human ENaC- δ (6), *C. elegans* ACD-2, DEL-9, and ASIC-1 (14), *D. melanogaster* Pickpocket1 (16), and *NeNaC2* from the sea anemone *Nematostella vectensis* (8).”

Reviewer #2 (Remarks to the Author):

The revised version of “Independent evolution of an ASIC-like Deg/ENaC sodium channel in the Placozoa” Elkhatib et al., includes extensive edits in response to a wide range of reviewer comments. Many of these have improved the manuscript significantly, including removing the unrelated (but interesting) animal behavior data and correcting fundamental errors in the phylogenetic analysis.

We would like to thank this reviewer for his insightful comments in the previous and this second version of the manuscript. We feel that in seeking to address these concerns, the manuscript has become more streamlined, clearer, and of higher quality.

The latter reduces the novelty of the manuscript by bringing its evolutionary conclusions more in line with a previous study, but there is still enough novelty here, primarily in the functional characterization of Trichoplax channels, to justify publication.

We disagree with the notion that our bioinformatics analyses lack novelty. In addition to providing support for the analysis by Aguilar-Camacho et al, our combined CLANS and phylogenetic analysis provides:

- The first robust phylogenetic evidence for the existence of Deg/ENaC channels outside of Metazoa (having only previously been identified by Poznyakov et al, 2021 via BLAST; cited in manuscript).
- Evidence, through cluster analysis, that ctenophores possess a set of highly divergent Deg/ENaC channels (not previously described in the paper by Schmidt et al, 2018, which also did CLANS analysis on Deg/ENaC channels).
- Phylogenetic evidence for the existence of numerous uncharacterized clades of Deg/ENaC channels in metazoans not described in the paper by Aguilar-Camacho et al.

However, there are still a couple of major issues left over from the original manuscript that need to be substantively addressed before the study is suitable for publication.

First, clarifications are still needed to justify inclusion of the Cluster Analysis (Figure 1). As explained in the rebuttal, the cut-off value for clustering was chosen because it separated P2XR and PAC clusters that are known to have different evolutionary origins and left the Deg/ENaC family together. None of this justification is included in the revised manuscript, and it would need to be. But if all the analysis is used for is to exclude channels that could be excluded a priori, what is the scientific purpose? Furthermore, there are numerous small clusters that are not connected to the main supercluster that are annotated to include channels from the superfamily in question, including many if not all of the PPKs. These are not noted in the rebuttal letter which would lead one to expect a very different looking figure with 3 clusters (Deg/ENaC, PAC, P2XR) Were these smaller DEG/ENaC-containing clusters included or excluded in the phylogenetic analysis? If excluded, this comes back to the original point that the cut-off value is arbitrary and not scientifically useful if it excludes known family members. In this case, cluster analysis should not have influenced the phylogeny. If instead the small isolated clusters were included, then why present the cluster analysis at all? It would essentially have no purpose. My point is it needs to be run and presented in a scientifically rigorous way (and described in complete detail), or simply deleted from the manuscript. But perhaps I am confused because of the incomplete descriptions.

Thank you for this suggestion. We disagree that the CLANS analysis should not be included, and would like to note that utilizing CLANS to pre-filter sequences for phylogenetic inference is not without precedent. This includes a recent study on Deg/ENaC channels which utilized a more stringent P value cut-off of 1E-50 (Schmidt et al., 2018) resulting in the exclusion of all PPK channels and likely many more sequences compared to our more permissive cut-off. However, we acknowledge that clarifications are needed pertaining to our methodology and reasons for selecting a specific cut-off for pre-filtering sequences. Therefore, we have completely repeated (and re-described) the bioinformatic analyses as follows:

- We have removed the PAC and P2X sequences from the analysis, since these are unrelated to Deg/ENaC channels, and in fact detract from our central focus on the phylogenetic relationships of TadNaCs with other Deg/ENaC channels.
- We have re-run the CLANS analyses using a range of P value cut-offs (1E-10 to 1E-50), and provided these as supplementary files for evaluation using the freely available CLANS program.
- We have generated a new CLANS figure and phylogenetic tree (new Figures 1 and 2), using a cluster of sequences selected for having at minimum one connection with the main cluster, using a P value cut-off of 1E-30. In addition, we lowered our P value cut-off from 1E-40 to 1E-30 because this permitted the inclusion of the majority of the PPK channels.
- We added a rationale in the results section as to why we selected a P value cut-off of 1E-30 for pre-filtering sequences prior to phylogenetic inference, and updated the materials and methods section accordingly.

Here is the updated text from the results section referring to our updated CLANS analysis:

“To better understand the relationships of *T. adhaerens* TadNaC channels to other Deg/ENaC channels, including those from the fellow placozoan *Hoiliungia hongkongensis*, we used CLuster ANalysis of Sequences (CLANS) (27) on a set of 1074 Deg/ENaC channel protein sequences extracted from high quality gene datasets of representative species spanning the major animal groupings, followed by phylogenetic inference. We tested a range of P value cut-offs for the CLANS analysis (*i.e.*, 1E-10, 1E-20, 1E-30, 1E-40, and 1E-50), finding in all cases that the sequences formed one major cluster comprised of two inter-connected sub-clusters (Fig. 1, Supplementary Data 3 to 7). One of these sub-clusters contained the chordate ASIC and BASIC channels, along with the *T. adhaerens* channels TadNaC1 to 9 and TadNaC11 (and corresponding *H. hongkongensis* homologues), and the other the chordate ENaC channels along with the singleton placozoan homologues TadNaC10 and HhoNaC10. Our analysis is altogether consistent with a similar recent study (19), both also finding the peptide-gated FaNaC and WaNaC channels from lophotrochozoans to associate with the ENaC sub-cluster, and the peptide-gated HyNaC channels from *Hydra magnipapillata* to associate with the ASIC/BASIC sub-cluster.

Comparing the various CLANS analyses at different thresholds, we found that decreasing the P value from 1E-20 to 1E-30 caused numerous sequences to no longer associate with the main cluster, including a large group of ctenophore sequences (Fig. 1, Supplementary Data 4 and 5). Increasing it further to 1E-40 caused a large group of *D. melanogaster* PPK channels to no longer associate with the ENaC sub-cluster, and a set of *C. elegans* ACD channels to lose their relatively strong connectivity with the ASIC sub-cluster (Fig. 1), instead forming a single connection with the ENaC sub-cluster (Supplementary Data 6). We therefore selected a P value cut-off of 1E-30 to isolate a central cluster of sequences for phylogenetic inference, reasoning that this cut-off struck a balance between strategically

removing divergent and/or truncated sequences that would interfere with phylogenetic analysis, while being permissive enough to include most PPK channels. In agreement, a previous study that employed a similar CLANS pre-filtering approach prior to phylogenetic analysis but with a P value of 1E-50 excluded the *D. melanogaster* PPK channels (19). In our analysis, pre-filtering the sequences at 1E-30 resulted in the removal of 200 sequences, which in addition to the noted cluster of ctenophore channels, included numerous singletons and smaller clusters from platyhelminths and cnidarians (Fig. 1). Lastly, our clustering analysis revealed that several Deg/ENaC homologues present in the gene data for unicellular eukaryotic species from the clades Heterokonta (*i.e.*, from the SAR supergroup, for Stramenopila, Alveolata, and Rhizaria) and Filasterea, clustered the ASIC and ENaC sub-clusters (Fig. 1), corroborating a report that Deg/ENaC channels are present outside of animals, in select unicellular organisms (28).”

And in the discussion section:

“Together, our CLANS and phylogenetic analyses provide evidence for the existence of two major clades of metazoan Deg/ENaC channels (Figs. 1 and 2), A and B, in agreement with several previously published clustering and phylogenetic analyses (8, 14, 19). Most animals possess Clade A and B channels, indicating that these two subfamilies emerged early during animal evolution. The exception are ctenophores, which lack group B channels. Thus, depending on whether poriferans or ctenophores were the first to diverge from other animals (52-55), ctenophores either lost Clade B channels, or these emerged in other animals after ctenophores diverged (8). Alternatively, the ctenophore Clade B channels underwent extreme sequence divergence obscuring their phylogenetic history. Indeed, ctenophores possess two groups of channels that did not associate with other Deg/ENaC channels in our CLANS analysis, and similar divergent clusters were apparent for cnidarians and platyhelminths. Indeed, future studies with increased taxon sampling will perhaps uncover whether these divergent channels are phylogenetically-related to Clade A or B channels.

In this study, we provide the first phylogenetic evidence that Deg/ENaC channels are not unique to animals, being present in unicellular eukaryotic lineages of Heterokonta and the filasterea-related species *T. unikontum*. Although our search was not exhaustive, we were unable to find Deg/ENaC sequences in the intervening opisthokont lineages that separate Metazoa and Filasterea, including choanoflagellates, or between these lineages and Heterokonta. This indicates that there was either extensive gene loss of Deg/ENaC channels in the intervening lineages of single celled eukaryotes, or alternatively, lateral gene transfer between animals, filastereans, and heterokonts. Interestingly, similar phylogenetic gaps are apparent for other major cation channels shared between animals and unicellular eukaryotes. For example, Cav3 voltage-gated calcium channels, conserved between animals and choanoflagellates, are also found in algae but not in the vast eukaryotic lineages that fall between these organisms (56). Similarly, the metazoan Na⁺ leak channel NALCN and its extracellular subunit FAM155A have homologues in fungi but no lineages in between (56, 57). Hence, it is conceivable that Deg/ENaC channels arose in the metazoan ancestor through lateral gene transfer, perhaps giving rise to the ENaC family, and that the ASIC family evolved thereafter. Alternatively, Deg/ENaC channels first evolved in animals and then transferred to select eukaryotes.

Like Aguilar-Camacho et al. (8), we found that most placozoan Deg/ENaC channels form a monophyletic clade within Clade A that has a sister relationship with BASIC channels, and together, these have a sister relationship with ASIC channels. Also similar, the singleton placozoan channels *TadNaC10* and *HhoNaC10* fall into Clade B, among the chordate ENaC sodium leak channels, the *C.*

elegans mechanically-gated channels MEC-10 and MEC-4, the *C. elegans* proton-gated channel ASIC-1, and the lophotrochozoan peptide-gated channels FaNaC and WaNaC.

Our phylogenetic analysis also identified several novel clades of bilaterian Deg/ENaC channels with unknown properties that are phylogenetically proximal to channels with described functional and/or physiological properties. In Clade A subclade I, we identified several groups of chordate (cephalochordate and urochordate), ambulacrarian, and lophotrochozoan channels that form strong phylogenetic relationships with the BASIC channels, the *Tad*NaC channels, and the proton-sensitive channels ACD-1, ACD-5, FLR-1, and ACD-2 from *C. elegans*. In Clade A subclade II, we identified groups of ambulacrarian, protostome, ctenophore, and poriferan channels among the PPK channels from *D. melanogaster*, the peptide-gated HyNaC channels from the cnidarian *H. magnipapillata*, and the NeNaC channels from *N. vectensis*. Similarly in Clade B, we identified numerous groups of uncharacterized channels from a broad range of animals among the ENaC channels, the *C. elegans* channels MEC-10, MEC-4, and ASIC-1, and the lophotrochozoan FaNaC and WaNaC channels. Future efforts aimed at determining the gating and physiological properties of channels within these various uncharacterized clades, especially from the earliest diverging animal lineages and unicellular eukaryotes, will be important as we seek to fill gaps in our understanding of Deg/ENaC channel evolution.”

And here are the updated/new figures:

Figure 1. BLOSUM62 cluster map revealing two major sub-clusters of metazoan Deg/ENaC channels. Nodes correspond to individual channel sequences and are coloured by taxon as indicated by the legend. Edges correspond to BLAST connections with P-values $\geq 1E-30$. The general locations of the chordate ASIC and ENaC channels, the cnidarian HyNaC and NeNaC channels, the lophotrochozoan FaNaC and WaNaC channels, and the *C. elegans* ACD channels are indicated. Singletons and non-connected clusters with less than five sequences are masked but available in the corresponding CLANS file (Supplementary Data 5).

Note that despite making the noted changes to our CLANS/phylogenetic analyses, our resulting phylogenetic tree is very similar to our previous versions:

Figure 2.

Figure 2. Maximum likelihood protein phylogeny delineates two major clades of metazoan Deg/ENaC ion channels. The tree was generated with the program IQ-TREE 2 with the best-fit model WAG+F+G4 and rooted with the filasterean-related Deg/ENaC channel homologues. Node support values are for 100

standard bootstrap replicates (green). The asterisks and labels (pink) indicate single channels or clades bearing Deg/ENaC channels that have been characterized as proton-activated.

Lastly, here are the revised materials and methods:

“Multiple-species sequences of the families Deg/ENaC (PF00858) channels were obtained from the PFAM database (<https://pfam.xfam.org>). These sequences were aligned using MUSCLE, and automatically trimmed with trimAl (74) using the gappy-out mode. The trimmed sequences were used to produce Hidden Markov Models using HMMER3 (75). The subsequent search for receptors in the database obtained as described above was performed using HMMER3 with an expect value cut-off of 1E-10 (76). The identified sequences are provided in Supplementary Data 2.

The relationship between candidate Deg/ENaC sequences was first analyzed using a cluster-based strategy with the CLuster ANalysis of Sequences CLANS algorithm (27). The sequences were clustered using P value cut-offs of 1E-10, 1E-20, 1E-30, 1E-40, and 1E-50 (Supplementary Data 3 to 7, respectively). We selected a P value cut-off of 1E-30 to pre-filter Deg/ENaC channel sequences prior to phylogenetic analysis, permitting the extraction of 874 sequences with at least one connection to the main cluster. These selected sequences were then analyzed using DeepTMHMM (77) to predict the transmembrane helices (TMHs), and those with no predicted TMH were discarded in order to remove sequences that were too fragmented to contribute to the phylogenetic analysis. Several additional sequences identified as duplicates via their sequence identifiers were also manually removed, after confirmation via pairwise sequence alignment. Selected and filtered protein sequences (798 total; provided in Supplementary Data 8) were aligned with MAFFT version 7 with the iterative refinement method L-INS-I (78), and alignments were trimmed with trimAl in gappyout mode (74) producing an alignment with 455 informative positions. The maximum-likelihood tree presented in Fig. 2 was generated using IQ-TREE 2 (79) with the best-fit model WAG+F+G4. To calculate branch support, we ran 100 standard bootstrap replicates (80). Select clades on the trees were collapsed with Figtree v1.4.4 (81), and then colored and the trees annotated using Adobe Illustrator. The nexus file for the tree is available in Supplementary Data 9, while the aligned sequences and aligned-trimmed sequences are available in Supplementary Data 10 and 11, respectively.”

Second, multiple reviewers pointed out that the data on proton sensing mechanism (Fig. 7) were not high quality and made the related conclusions suspect. This problem was not addressed with new data, but it should have been. In short, the data for H109A and H80A is simply not of sufficient quality to support the conclusion that these two residues play a significant role in proton activation.

We realize that plotting the dose response data for multiple mutants in combined graphs made it difficult to interpret the trends and standard deviations of our data, including for the H109A and H80A mutants. We therefore changed these plots to show the connections between data points (new panels A to C in Figure 7), and created a new supplementary figure with just the dose response curves for the H109A and H80A mutants alongside the wild type channel (Supplementary Figure 2A and B). In these new/modified plots, it is evident that the standard deviations for these two channel variants are clearly distinct from those of the wild type channel, and are reasonably tight at most pH values. These plots make it easier to see that mutation of these two residues caused a biphasic sensitivity to protons. This effect was strongest for the H109A variant, apparent in the sample recording provided for this channel (Figure 7f): at pH 5, the macroscopic current bears a combined fast and slow component, then at pH 4.5, the amplitude is severely diminished and the transient current is absent, leaving only a slow onset

current. This slow onset current then increases in amplitude at pH 4.0. For the H80A mutant, we also note a general absence of the transient current, with a minimal fast element apparent only under very acidic pH conditions (see Figure 7f).

Nonetheless, although we are confident in our data, it is clear from the reviewer's comments that we needed to do a better job describing the mutagenesis data (including the biphasic effects), and tame our descriptions of the significance of the H80 and H109 residues for *TadNaC2* proton activation. We therefore modified the title, abstract, introduction, results, and discussion as follows:

Title change:

Function and phylogeny support the independent evolution of an ASIC-like Deg/ENaC channel in the Placozoa

Abstract:

“Structural modeling and mutation analyses reveal that *TadNaC2* proton-activation is inherently different from ASIC channels, lacking key molecular determinants, and involving unique residues within the palm and finger regions (H80 and H109).”

Introduction:

“We find that *TadNaC2* lacks all major determinants for proton activation of ASIC channels, including the critical H73 and K211 residues that are common to ASIC channels (10, 25), and the acid pocket, a region also important for proton activation of ASIC channels (26). Instead, we identify two histidine residues, H80 and H109, within the palm and finger regions respectively, that contribute to *TadNaC2* proton activation as revealed by mutation analysis.”

Results:

“In the finger region, single mutations of E104A and E105A had no effect on pH_{50} or peak current amplitude (Fig. 7b, d, and e). However, mutation of both together caused a moderate decrease in both metrics, and altered the macroscopic current waveform by diminishing the fast/early component (Fig. 7f). A more dramatic effect occurred for the single mutation H109A, which in addition to reducing maximal peak current amplitude (Fig. 7e), produced a biphasic macroscopic current and a complete loss of the early current component at pH 4.5 and 4.0 (Fig. 7b and f; Supplementary Figure 2a). Interestingly, mutation of the acid pocket residue D345 in mouse ASIC1a (Fig. 6b), which is close to the predicted finger region of *TadNaC2* where H109 resides (Fig. 6f), similarly imposes biphasic sensitivity to pH, attributed to the loss of one of two separate proton binding sites involved in channel activation (the other being in the palm domain) (54). However, the biphasic effect is much more severe for the *TadNaC2* H109A mutation, where instead of plateauing between pH 5.0 and 4.0 like the mASIC1a channel, the current amplitude first decreases from pH 5.0 to 4.5, then increases again from pH 4.5 to 4.0 (Fig. 7b and f, Supplementary Figure 2a). This atypical feature precluded accurate fitting of the dose response data with either standard or biphasic dose response curves, since both encapsulate strictly incremental processes (*i.e.*, R^2 values of 0.64 and 0.68, respectively). Nonetheless, fitting a standard dose response curve over the data revealed reduced sensitivity to protons compared to the wildtype channel, with an average pH_{50} of 4.7 ± 0.2 vs. 5.2 ± 0.1 (Fig. 7d). Instead, fitting the data with a bimodal dose response curve produced a pH_{50-1} value of 5.5 ± 0.3 and a pH_{50-2} value of 4.5 ± 0.3 , both of which are

statistically different from wildtype (*i.e.*, P values for two-sample t-Tests ≤ 0.005 and 0.0005 , respectively). However, since the H109A variant shows diminished sensitivity to protons at the threshold pH of 5.5 (Supplementary Figure 2a), this mutant channel is not likely more sensitive to protons at threshold pH values, but rather, has an overestimated the pH_{50-1} value caused by the poor curve fit. Notably, while macroscopic current amplitudes varied significantly for the wildtype channel, the pH_{50} values were less variable (Fig. 7d and e). Furthermore, we found no correlation between current amplitude and pH_{50} for the wildtype channel, indicating that observed differences in pH sensitivity for the H109A mutant and other channels variants is not due to decreased current amplitude. The most severe of all mutations tested was a triple mutation E104A/E105A/H109A, which produced a channel with very weak activation at pH 5.5 and 4.5, completely lacking transient/early currents at all tested pH (Fig. 7f). This resulted in the most significant reduction in proton sensitivity and current amplitude, with an average pH_{50} of 4.4 ± 0.1 (Fig. 7d and e). Altogether, it appears as though the H109 residue, together with E104 and E105, plays an important role in the proton activation of *TadNaC2*.”

Discussion:

“In the palm region, we found that the H80 residue in *TadNaC2* plays an important role in proton activation, wherein its mutation caused a significant reduction in pH_{50} (Fig. 7c and d). An interesting feature of the H80A mutant channel was its plateaued or biphasic activation between pH 5.0 and 4.5 (Fig. 7c and e). In mouse ASIC1a, the protonatable residue E79 sits in a similar position as H80 in *TadNaC2* (Fig. 6b), at the distal end of the first beta strand that projects from transmembrane helix 1 in the wrist domain into the palm domain (54). Notably, like the H80A mutation in *TadNaC2*, mutation of E79 to a lysine, expected to disrupt a putative proton-binding site comprised of this residue and E416 on an adjacent beta strand (62), produced a biphasic proton sensitivity. The authors of this study found that a separate mutation, of residue D345 within the acid pocket (Fig. 6b), also produced biphasic proton sensitivity, and that mutation of E79 and D345 together produced severely diminished proton sensitivity that was not biphasic. This suggests that ASIC1a possesses at least two separate proton binding sites that contribute to channel activation, one within the palm domain and another within the finger-thumb region, and that these sites operate somewhat independently of each other (54).”

“In the finger region we identify the residue H109, E104, and E105 as important for proton activation of *TadNaC2*, with a single mutation of H109 causing a moderate reduction in pH_{50} , and the triple mutation of E104A/E105A/H109A having the most severe effect compared to all other tested mutants (Fig. 7b and d). These residues are fully conserved between *TadNaC2* and *HhoNaC2*, while H109 is also found in several other Deg/ENaC channels including several ASIC channels, and the E104/E105 doublet is found in *NeNaC2* (Fig. 6b). As noted, the finger region is of particular importance for ASIC2 channels where four amino acids differentiate proton-sensitive ASIC2a channel variants from non-functional ASIC2b variants (50). Alanine substitution of the H109 equivalent in ASIC2a (*i.e.*, H109A) produced different effects in separate studies, two describing no effect (45, 50), and another a complete loss of proton-activation perhaps attributable to a reduction in membrane expression (46). To the best of our knowledge, whether this equivalent residue in ASIC1a, or a histidine residue one position upstream in ASIC3 (Fig. 6b), contribute to proton activation has not been explored. However, as noted above, mutation of residue D345 in mouse ASIC1a, which sits adjacent to the finger region where H109 resides in *TadNaC2* (Fig. 7f), has a strong impact on proton activation and similarly imposes a biphasic pH sensitivity (54).

Thus, an interesting prospect emerges that like ASIC1, *TadNaC2* possesses two separate loci for coordinating protons and channel activation, one in the finger-thumb region and the other in the palm domain, although utilizing different molecular determinants.”

New/modified figures:

Supplementary Figure 1. (a and b) pH dose response curves for the *TadNaC2* H109A and H80A channel variants. (c) Plot of percent residual current at 5 seconds after peak for the *TadNaC2* wildtype and R201A channel variants at different pH. The asterisks denote statistically significant p values ≤ 0.05 for two-sample t -tests.

Figure 7. Unique residues contribute to proton-activation of *TadNaC2*. (a to c) Average pH dose response curves \pm standard deviation for wildtype (wt) *TadNaC2* and variants bearing amino acid substitutions within the wrist (panel a), finger (panel b), and palm (panel c) regions. (d and e) Plot of average \pm standard deviation pH_{50} (panel d) and normalized peak current amplitude (panel e) for wildtype (wt) and various point mutated *TadNaC2* channels. Cyan-colored asterisks denote p value thresholds for two-sample t-tests comparing wildtype to mutant values. (f) Sample whole-cell currents of wildtype *TadNaC2* and select mutant variants.

Whether the proton dose response curves are significantly shifted (or not) looks like it depends entirely on which points are viewed as outliers. In H109A for example, the curve is fit as if pH 5 is an outlier and pH 4.5 is accurate (both can't reside near the curve) and the choice appears arbitrary. However, if this was reversed and pH 4.5 was treated as an outlier, then pH sensitivity of the mutant would appear very similar to WT. The same can be said for H80A, and the example traces provided for both mutants do not inspire confidence.

We did not treat any of the data as outliers in the curve fitting; we just fit a standard dose response function over all the data points. In this version of the manuscript, we have fit both standard and biphasic dose response curves over the data, as described in the results text in the section above.

It looks like data could be contaminated by desensitization issues or just highly variable due to small current size? There is a real danger here that the conclusions based on this data will not hold up. The authors should either remove these conclusions or provide additional high quality data that convincingly supports the conclusions that these mutants have significantly altered proton sensitivity.

We are confident in our data and disagree that these two mutants did not affect gating and proton sensitivity. We don't think that the current size had a significant bearing on pH_{50} , because the R201A had similarly diminished current amplitudes but had an increased, rather than decreased pH_{50} value. Furthermore, we hope that our new plot of just the H80A and H109A mutants alongside the wildtype channel illustrates that replicated currents for each mutant variant are adequately consistent with each other, and most importantly, clearly different from those of the wildtype channel.

Reviewer #3 (Remarks to the Author):

In the second version of the manuscript “Independent evolution of an ASIC-like Deg/ENaC sodium channel in the Placozoa” by Elkhatib et al., the authors have made major changes in text and figures, content, and structure. As required, the authors have removed the behavioral experiments from the manuscript improving the clarity and message of the manuscript.

After careful consideration, I can say that the current version of the manuscript addresses most of my concerns. However, there are a few areas that require improvement. Regarding mutants R201A and K203A, some statements can be misleading. Authors claim that mutations R201A and K203A “produced different effects” (line 426). However, both mutants significantly increase the proton sensitivity compared to WT (i.e. from a pH_{50} of around 5.1 in the WT to 5.3 in K203A and 5.5 in R201A, both with a p-value equal to or lower than 0.05).

Thank you for pointing this out. We have modified the corresponding text to the following:

“In *TadNaC2*, mutation of the two cationic residues that flank the K211 position, R201 and K203, produced an increase in proton sensitivity with respective pH_{50} values of 5.3 ± 0.1 and 5.5 ± 0.0 (Fig. 7c and d).”

In the same paragraph, the authors say that R201A and K203A mutants “did not alter the current waveform significantly” (line 428). However, the ratio between the sustained component and transient component of the current is clearly different than WT. I suggest the authors calculate the relation between current amplitudes for sustained and transient components as an indicator of the waveform.

Thank you for making this astute observation. As recommended, we have analyzed the R201A vs. wt data and found that indeed the residual current 5 seconds after peak was diminished relative to wildtype at pH values of 5.0 and 4.5, but not 4.0. We have created a new plot to demonstrate this new analysis, shown in the new Supplementary Figure 2 panel C (please see above). We have added this description to the results section as follows:

“In *TadNaC2*, mutation of the two cationic residues that flank the K211 position, R201 and K203, produced an increase in proton sensitivity with respective pH_{50} values of 5.3 ± 0.1 and 5.5 ± 0.0 (Fig. 7c and d). Notably, the R201A mutation also altered the macroscopic current waveform such that the amplitude difference between the early and late components was greater at pH 5.0 and 4.5 compared to wildtype, but not at pH 4.0 (Supplementary Figure 2c).”

The authors should, also, show current traces for K203A mutants (I could not find them).

We have added a sample current trace for the K203A mutant to panel 7F (please see above). We would also like to address an error we made previously in our description of the currents for this channel variant, in that it lacks a transient current component, much like the H80A variant. We sincerely apologies for this mistake, and have modified the text as follows:

Results section:

“Instead, the K203E mutation disrupted the early current such that it only became evident at very acidic pH values (Fig. 7f). Deletion of this same residue (K203Δ), to emulate K211Δ variants of ASIC channels,

resulted in an inability to detect currents even with very acidic pH. Alanine substitution of the unique protonatable H80 residue in the palm region, which is proximal to R201 and K203 in our predicted structures (Fig. 6), caused the dose response data to become more variable, and the pH sensitivity to become biphasic similar to the H109A mutation (Fig. 7c and f; Supplementary Figure 2b). Furthermore, and like the K203E mutation, the H80A mutation significantly disrupted the transient current, which was only evident under very acidic pH conditions (Fig. 7f).”

Because of the large number of changes throughout the manuscript some figures and text had become outdated. For example, authors have removed the Hill coefficient from Fig. 7, but they still refer to it in the text (Lines 574 and 583). Line 623, “Fig. f” consider changing to “Fig. e”. Line 627, “Fig.7 g to j” should be “Fig. 8”. Fig. 8 legend (line 1081) panel “i” should be “c”.

Thank you. We have made these corrections.

Finally, the authors use sources that have been questioned in previous versions of the manuscript to create a theoretical context (line 657). I suggest the authors reconsider if they want to add that reference.

We have removed this source and the corresponding text from the manuscript.

Overall, I believe that with those few modifications, the manuscript presented by Elkhatib et al. will be a valuable contribution to scientific literature.

Thank you for your insightful comments and suggestions.

Reviewers' comments:

Reviewer #2 (Remarks to the Author):

In this revision the authors addressed questions about 1) the details and methods for using their Cluster Analysis as a pre-filter for phylogenetic analysis and 2) Data for TadNaC2 mutants H80A and H109A which they interpret to mean these residues serve as proton binding sites for proton-dependent activation.

1) The additional explanations and analysis for the cluster analysis and subsequent phylogeny are helpful. Furthermore, it is now clear that the tree is not intended to represent the entire DEG/ENaC superfamily, but instead is intended solely to show the position of the Placozoan channels relative to major groups within the family. The justification of cluster analysis cut-off values is sufficient for this purpose.

2) The authors make a solid case that their data as collected is of sufficient quality for analysis, and the authors are correct that the pH dose response curves of the H80A and H109A mutants are significantly different than WT as measured. However, the interpretation of the H80A and H109A data, at least as it appears to have been collected, is still problematic. The data are not sufficient to favor these residues as proton sensing sites for activation over other possible explanations. First, the authors have the difficult hurdle that the only way they can open the channels is via proton activation. Therefore, there is no clear way to differentiate proton sensing from downstream effects on gating that could also alter observed current size and thus the dose response curve. For example, the mutants have altered desensitization that could dramatically impact the size of currents measured independent of the degree of activation. It is difficult to determine the full extent to which desensitization alters currents measured because the protocols for collection of pH dose response curves are not clearly laid out in the methods. However, from example sweeps shown in Figure 4A, it appears that measurements were made sequentially starting at high pH and then lowering. This collection method would be very likely to contribute to the unusual dose response curves observed that initially made me question data quality. For example, for H109A, the first large current is observed at pH 5, but it desensitizes much more rapidly and completely than WT. Then at pH 4.5 there is almost no current. If these were recorded sequentially, could it just be that the channels are still desensitized? And that they slowly recover at very low pHs? Would pH 4.5 have a large, desensitizing current if it was the first pH step applied to the cell? Similar arguments could apply to H80A for which there isn't a visible desensitizing spike to make it easier to interpret the degree of desensitization (but the authors should note that if desensitization is fast enough relative to activation, it could still be present even in the absence of an initial spike). The point is here is that much more than just pH-dependent activation is contributing to the current size here. The authors need to acknowledge that they haven't deconvoluted the results sufficiently to conclude that these mutations show H80 and H109 are the proton binding sites for pH-dependent activation. They can't (and shouldn't) even favor that explanation over alternatives based on this data. However, they could conclude that these mutants have clearly altered gating demonstrating that these structural regions of the channels also play a central role in gating in the TadNaC2, as they do in ASICs. Pulling back to this level of conclusion and discussing the limitations of the data collection protocol and the difficulties of deconvoluting sensing/gating would circumvent the need for new data sets collected under different protocols. Alternatively, the authors could provide additional data to boost the case that these mutants are directly involved in the proton binding events that lead to channel activation.

Reviewer #2 (Remarks to the Author):

In this revision the authors addressed questions about 1) the details and methods for using their Cluster Analysis as a pre-filter for phylogenetic analysis and 2) Data for TadNaC2 mutants H80A and H109A which they interpret to mean these residues serve as proton binding sites for proton-dependent activation.

1) The additional explanations and analysis for the cluster analysis and subsequent phylogeny are helpful. Furthermore, it is now clear that the tree is not intended to represent the entire DEG/ENaC superfamily, but instead is intended solely to show the position of the Placozoan channels relative to major groups within the family. The justification of cluster analysis cut-off values is sufficient for this purpose.

We are happy that the revisions made this section of the manuscript clearer. Thank you again for your detailed review of the paper and the ideas you have brought to our work.

2) The authors make a solid case that their data as collected is of sufficient quality for analysis, and the authors are correct that the pH dose response curves of the H80A and H109A mutants are significantly different than WT as measured.

Thank you.

However, the interpretation of the H80A and H109A data, at least as it appears to have been collected, is still problematic. The data are not sufficient to favor these residues as proton sensing sites for activation over other possible explanations.

Please see below.

First, the authors have the difficult hurdle that the only way they can open the channels is via proton activation. Therefore, there is no clear way to differentiate proton sensing from downstream effects on gating that could also alter observed current size and thus the dose response curve. For example, the mutants have altered desensitization that could dramatically impact the size of currents measured independent of the degree of activation. It is difficult to determine the full extent to which desensitization alters currents measured because the protocols for collection of pH dose response curves are not clearly laid out in the methods. However, from example sweeps shown in Figure 4A, it appears that measurements were made sequentially starting at high pH and then lowering. This collection method would be very likely to contribute to the unusual dose response curves observed that initially made me question data quality. For example, for H109A, the first large current is observed at pH 5, but it desensitizes much more rapidly and completely than WT. Then at pH 4.5 there is almost no current. If these were recorded sequentially, could it just be that the channels are still desensitized? And that they slowly recover at very low pHs? Would pH 4.5 have a large, desensitizing current if it was the first pH step applied to the cell? Similar arguments could apply to H80A for which there isn't a visible desensitizing spike to make it easier to interpret the degree of desensitization (but the authors should note that if desensitization is fast enough relative to activation, it could still be present even in the absence of an initial spike). The point here is that much more than just pH-dependent activation is contributing to the current size here. The authors need to acknowledge that they haven't deconvoluted the results sufficiently to conclude that these mutations show H80 and H109 are the proton binding sites for pH-dependent activation. They can't (and shouldn't) even favor that explanation over alternatives based on

this data. However, they could conclude that these mutants have clearly altered gating demonstrating that these structural regions of the channels also play a central role in gating in the *TadNaC2*, as they do in ASICs. Pulling back to this level of conclusion and discussing the limitations of the data collection protocol and the difficulties of deconvoluting sensing/gating would circumvent the need for new data sets collected under different protocols. Alternatively, the authors could provide additional data to boost the case that these mutants are directly involved in the proton binding events that lead to channel activation.

Although in our previous version of the manuscript we refrained from directly referring to H80 and H109 as proton binding residues, we realize that certain sections of the text and some of the titles might have implied this as a possibility. We therefore made edits to the text to remove any mention of proton-sensing or proton-binding sites. We also modified all descriptions of specific amino acids as contributing to “proton gating”, rather than “proton activation”. We also agree with the idea that the lack of a transient current observed for some mutants could result from altered gating kinetics, and have thus incorporated this idea into the manuscript, as well as the requirement for future studies aimed at deconvoluting roles for amino acids in proton binding vs. contributions to gating (activation, desensitization, and recovery from desensitization).

In the discussion (starting at line 663), we added the following caveat about our experiments and the requirement for future research:

“An important caveat about the experiments and data presented in this study is that they do not permit direct inferences about the specific contributions of H80, H109, and other tested amino acids towards *TadNaC2* proton gating. Specifically, future functional and structural experiments will be required to determine whether these amino acids are acting as direct proton binding sites or play separate roles in channel gating (*e.g.*, activation, desensitization, and/or recovery from desensitization). This will also require a deeper characterization of the transient and sustained components of the *TadNaC2* macroscopic current, and a determination of how (and if) different structural elements of the channel protein contribute to these currents.”

And here are the additional changes that we introduced into the manuscript to address this concern:

Abstract:

Line 38, sentence changed from

“Structural modeling and mutation analyses reveal that *TadNaC2* proton-activation is inherently different from ASIC channels, lacking key molecular determinants, and involving unique residues within the palm and finger regions (H80 and H109).”

To

“Structural modeling and mutation analyses reveal that *TadNaC2* proton gating is different from ASIC channels, lacking key molecular determinants, and involving unique residues within the palm and finger regions.”

Introduction:

Line 111, sentence changed from

“Instead, we identify two histidine residues, H80 and H109, within the palm and finger regions respectively, that contribute to *TadNaC2* proton activation as revealed by mutation analysis.”

To

“Instead, our mutation analysis revealed that two histidine residues, H80 and H109, within the palm and finger regions respectively, contribute to *TadNaC2* proton gating.

Results:

Line 395, section title changed from

*“Novel protonatable residues in the palm and finger region contribute to *TadNaC2* activation.”*

To

*“Novel residues in the palm and finger region contribute to *TadNaC2* proton gating.”*

Line 396, paragraph changed from

“Despite lacking key deterministic residues for proton activation, the similar predicted structure of *TadNaC2* compared to mASIC1a prompted us to examine whether corresponding structural regions in the placozoan channel bear proton-sensitive elements. Thus, we performed site-directed mutagenesis on selected aromatic or protonatable residues in the wrist region (F70, D75, and E77), protonatable residues in the finger region (E104, E105, and H109), and protonatable or cationic residues in the palm region (H80, R201, and K203) (Fig. 6a). To assess changes in H⁺ sensitivity, we tested each mutant with a series of perfused solutions of various pH to generate dose response curves of recorded macroscopic currents (Fig. 7a to c).”

To

“Despite lacking key deterministic residues for proton activation, the similar predicted structure of *TadNaC2* compared to mASIC1a prompted us to examine whether corresponding structural regions in the placozoan channel bear unique or conserved elements involved in channel gating. Thus, we performed site-directed mutagenesis on selected aromatic or protonatable residues in the wrist region (F70, D75, and E77), protonatable residues in the finger region (E104, E105, and H109), and protonatable or cationic residues in the palm region (H80, R201, and K203) (Fig. 6a). To assess changes in H⁺ sensitivity and gating properties at different pH, we tested each mutant with a series of perfused solutions of various pH to generate dose response curves of recorded macroscopic currents (Fig. 7a to c).”

To address the observation that H109A mutation imposed marked changes in the kinetics of the transient current, we added the following sentence at line 445:

“It is notable that the transient current observed at pH 5.0 desensitized more rapidly compared to the wildtype channel, while at more acidic conditions the transient current was absent leaving only a slowly activating sustained current that increased in amplitude from pH 4.5 to 4.0 (Fig. 7f).”

Line 451, sentence changed from

“Altogether, it appears as though the H109 residue, together with E104 and E105, plays an important role in the proton activation of *TadNaC2*.”

To

“Altogether, it appears as though the H109 residue, together with E104 and E105, plays an important role in the proton gating of *TadNaC2*.”

Line 473, sentence changed (and a new one added) from

“Altogether, these observations indicate that the H80 residue also contributes to the proton activation of *TadNaC2*.”

To

“Altogether, these observations indicate that the H80 residue also plays a role in the proton gating of *TadNaC2*. Furthermore, the R201 and K203 residues also contribute to *TadNaC2* gating, however, their mutation did not produce a rightward shift in the pH dose response curve as it did for the analogous K211 residue in ASIC channels (25), indicating key functional differences.”

Line 1097, Figure legend changed from

“Figure 7. Unique residues contribute to proton-activation of *TadNaC2*.”

To

“Figure 7. Unique residues contribute to proton gating of *TadNaC2*.”

Discussion:

Line 597, title changed from

“Unique molecular determinants underlie proton activation of *TadNaC2*.”

To

“Unique molecular determinants underlie proton gating of *TadNaC2*.”